# Mapping multi-dimensional variability in water stress strategies across temperate forests

Daijun Liu ®[1,2,3] ✉, Adriane Esquivel-Muelbert ®[1,2], Nezha Acil ®[1,2,4,5],
Julen Astigarraga ®[6], Emil Cienciala ®[7,8], Jonas Fridman ®[9],
Georges Kunstler ®[10], Thomas J. Matthews[1,2,11], Paloma Ruiz-Benito[6,12],
Jonathan P. Sadler ®[1,2], Mart-Jan Schelhaas ®[13], Susanne Suvanto[1,2,14],
Andrzej Talarczyk ®[15], Christopher W. Woodall ®[16], Miguel A. Zavala ®[6],
Chao Zhang[17] & Thomas A. M. Pugh ®[1,2,18]

Increasing water stress is emerging as a global phenomenon, and is anticipated to have a marked impact on forest function. The role of tree functional strategies is pivotal in regulating forest fitness and their ability to cope with water stress. However, how the functional strategies found at the tree or species level scale up to characterise forest communities and their variation across regions is not yet well-established. By combining eight water-stress-related functional traits with forest inventory data from the USA and Europe, we investigated the community-level trait coordination and the biogeographic patterns of trait associations for woody plants, and analysed the relationships between the trait associations and climate factors. We find that the trait associations at the community level are consistent with those found at the species level. Traits associated with acquisitive-conservative strategies forms one dimension of variation, while leaf turgor loss point, associated with stomatal water regulation strategy, loads along a second dimension. Surprisingly, spatial patterns of community-level trait association are better explained by temperature than by aridity, suggesting a temperature-driven adaptation. These findings provide a basis to build predictions of forest response under water stress, with particular potential to improve simulations of tree mortality and forest biomass accumulation in a changing climate.

Forests across the world are facing increasing water stress because of climate change[1–3]. Forest responses to water stress will strongly determine their future ability to deliver key ecosystem functions and services such as carbon sequestration, microclimatic amelioration and wood production[4]. These responses to water stress will be largely driven by the ecological strategies of the trees that compose them[5–7]. Ecological strategies describe the set of adaptations and behaviours that allow a species to maintain a population[8–10] and we use this term

here in terms of adaptations to do so in the face of water stress. These water stress adaptation strategies include closing leaf stomata early, investing in stronger water transport structures, dropping leaves, storing water and developing deeper roots[7,11–14] (Fig. 1a). These strategies themselves emerge from anatomical or physiological properties that can be characterised by functional traits[7,15], closely linked to survival, development, growth and reproduction[10,16,17]. The values of these traits are often constrained by trade-offs between them[13,18,19]. Whilst

**Fig. 1 | Conceptual diagram of how tree level adaptations with respect to water stress scale up to biogeographic patterns of trait associations at the level of forest communities. a** Trees have several possible functional pathways that affect their response to water stress[7,9,17]. **b** These pathways combine into different potential species-level strategies, which include structurally conservative, early stomatal closure to avoid water loss, water storage and deep rooting. **c** Species assemble into communities, where the combination of their strategies controls community form and function. Community-level functional traits are calculated as their community-weighted mean of each species within a community. Co-variation of these traits, as assessed using principal component analysis (PCA), can identify the trait associations at the community level, which in most cases are expected to reflect the underlying strategies within each community. **d** Biogeographic patterns of these associations are expected to emerge as a function of broad-scale patterns in climate[65]. This figure is created in BioRender. Zhang, C. (2024) BioRender.com/s74n436.

the knowledge of water stress strategies at the tree or species level is increasingly understood[7,11,12], we lack large-scale assessments of the trait associations at the level of forest communities, which emerge from the strategies across their constituent individuals. To understand how responses to water stress vary across the world's forests, it is first necessary to characterise these trait associations and to identify how they vary in space.

Tree functional strategies have been widely characterised as a continuum between acquisitive species, which take risks in order to win contests for resources, and conservative species, which invest in tolerance of stress in order to thrive in harsh conditions[8–10,17–19]. For example, a tree may grow in a way that indicates light acquisition plays a more important role than the ability to tolerate water stress, hence suggesting carbon allocation strategies focused on height growth relative to the area of conductive sapwood, for a given leaf area. Assuming the same sapwood conductivity and robustness of xylem to cavitation, during a drought, this high leaf area to sapwood ratio may lead to excessively negative pressures in the xylem, causing hydraulic failure and death, whereas species favouring a lower ratio of leaf to sapwood area may survive (Fig. 1b). The combination of structural traits influences whether a tree is structurally more acquisitive or conservative. Other adaptations to water stress (e.g., isohydricity or water regulation) may not map so directly onto this continuum[6,14,20]. For instance, isohydric species, which close their stomata under moderate water stress, may avoid putting stress on their xylem but also impair their ability to photosynthesise. Anisohydric species keep stomata open under greater water potential gradients, but either need to invest in more robust water transport structures or risk hydraulic failure. Furthermore, water storage and deeper rooting may allow trees to endure dry conditions by avoiding water stress[21,22]. The functional strategy space is, therefore, complex and where an individual tree is situated within it has implications for the function and ecological services it provides.

To move from tree-level responses to forest community-level responses, it is necessary to characterise how water stress strategies assemble into communities. When species assemble into communities, the strategies that come to dominate depend on niche complementarity and resource competition[23–25]. As water resources become less limiting, competition may tend to favour more acquisitive strategies, resulting in convergence towards a limited range of water-demanding strategies[9,10,17,26]. In water-limited environments, a wide range of water stress strategies may be presented. These strategies have been shown to be ecologically partitioned according to specific combinations of environmental conditions[11,27,28]. How the trait association at the community level varies as a function of water stress is not well established. Understanding trait associations is, however, fundamental to being able to predict the likely response of the forest community to ongoing environmental changes and to generating accurate predictions by vegetation models, which are used to make projections of future forest function[2]. These models are currently taking big steps forward in their capacity to simulate physiological processes relevant to water stress responses[29–32], but require precise data on the distribution of forest water stress strategies. An exploration of the trait associations along climatic gradients, allowing us to infer the water stress strategy within forest communities, would provide a new perspective on forest functional biogeography and open up opportunities to support large-scale projections of forest function using vegetation models.

Here, we combine a large dataset of functional traits for woody plants with forest inventory plot data across regions of the United States of America and Europe, including Spain, France, Germany, Poland, Czechia and Sweden. We consider eight continuous functional traits of woody plants (shrubs and trees) related to potential functional strategies for dealing with water stress (Table 1). We concentrate on acquisitive-conservative, structural, stomatal and water storage strategies, as the information on rooting traits is very limited. The choice of these traits is driven particularly by their relevance for informing process-based modelling of forest form and function. We aggregate these forest plots using a 0.25° × 0.25° grid (grid-cell level statistics are hereafter referred to as community level) to reduce stochasticity and then calculate community-weighted mean traits for each community separately. The community-weighted means encapsulate the trait associations at the community level. With the exception of communities including two or more equally prevalent but opposite strategies, they can also be expected to be representative of the strategies across the individuals that compose that community. We use data for 12,452 forest communities across the USA and Europe, for which we also extract climate variables relating to water availability and temperature information based on their geographic locations (details of the variables in the Methods). We first assess the axes of variation of community-level trait associations with respect to water stress traits based on principal component analysis (PCA), and then compare them to the axes of variation of plant strategies at the species level. Finally, we relate the community-level associations to the geographic and

**Table 1 | The functional traits considered in this study**

| Functional traits (abbreviation) | Units | Explanation |
|---|---|---|
| Leaf nitrogen content (N) | mg g$^{-1}$ | Proxy for Rubisco content of leaves and thereby photosynthetic capacity and growth[9,10,17]. More acquisitive strategies are expected to have higher N. |
| Maximum xylem conductivity per unit sapwood area (Ks) | kg m$^{-1}$ s$^{-1}$ MPa$^{-1}$ | Key structural determinants of the xylem water potential need to be tolerated to keep the canopy supplied with water[7,13,18]. High Ks and/or LS are expected to be associated with more acquisitive strategies. |
| Leaf area to sapwood area ratio (LS) | mm$^2$ mm$^{-2}$ | |
| Leaf mass per area (LMA) | g m$^{-2}$ | Investment in leaf tissue relates to defensive allocation and helps dissipate energy from high radiation inputs, protecting against high temperature[9,10,17]. More conservative strategies have higher values. |
| Xylem water potential at 50% loss of conductivity (P50) | MPa | A critical determinant of tolerance to large water potential gradients and closely related to tree mortality[7,11,12]. Lower P50 values indicate more tolerance to water stress. |
| Embolism vulnerability curve between P50-P88 (Slope) | % MPa$^{-1}$ | Extent to which xylem cavitation is a gradual or threshold-like response to water potential[11]. All else being equal, a steeper slope indicates more sensitivity to water stress. |
| Leaf turgor loss point (TLP) | MPa | Closely correlated to isohydricity[14,74]. A less negative TLP is associated with more isohydric behaviour. |
| Wood density (WD) | g cm$^{-3}$ | Associated with the investment in defence and water storage[34,75,76] and a fundamental link between carbon investment and size growth. It has been linked to stomatal regulation and water storage strategies[14]. |

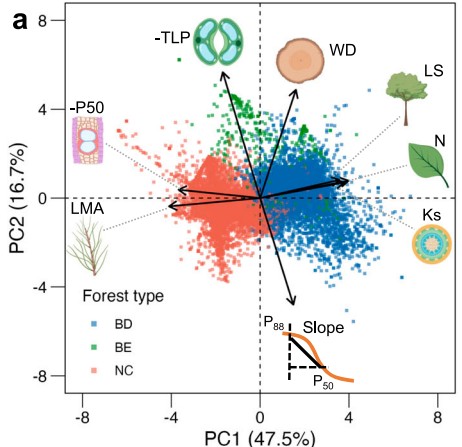
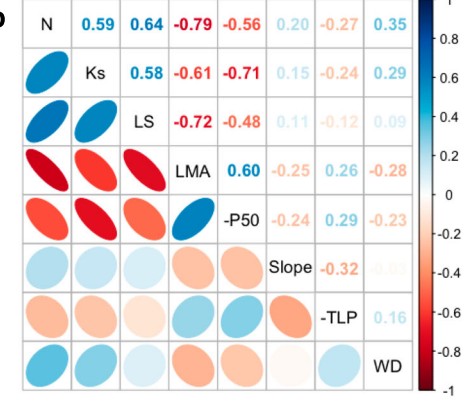

**Fig. 2 | Key trait associations along the two dimensions for the forest communities. a** Biplot resulting from the principal component analysis (PCA) of eight functional traits at the community level. There were 12,452 forest communities used in our analysis. The community weighted-mean trait was calculated based on all species-level trait values weighted by the proportions of the species basal area within a community. The dominant forest type was identified according to the relative proportion of dominant types representing more than 50% of the community basal area: BD (broadleaved deciduous, blue, 5584 communities); BE (broadleaved evergreen, green, 407 communities) and NC (needleleaved conifer, red, 6461 communities). **b** The Pearson correlation coefficients between the functional traits at the community level (all *p* values < 0.001). The Pearson correlation tests are two-sided. Abbreviations of functional traits: N leaf nitrogen content, Ks maximum xylem conductivity per unit sapwood area, LS leaf area to sapwood area ratio, LMA leaf mass per area, P50 xylem water potential at 50% loss of conductivity, Slope for the embolism vulnerability curve between P50–P88, TLP leaf turgor loss point and WD wood density. Traits explanations are provided in Table 1. The elements in panel **a** are created in BioRender. Zhang, C. (2024) BioRender.com/u64o243.

climatic space of temperate forest communities (Fig. 1c, d). We hypothesise that: (1) the strategic variation at the species level will be reflected in the variation of trait associations at the community level; (2) there is a tendency towards communities dominated by more structurally acquisitive strategies (and thus community-level trait associations) as the climate becomes less arid, reflecting a shift towards competition for light rather than water; and (3) there is more divergence in community-level associations across water-limited habitats, reflecting the broad range of options that trees can pursue to survive when water is limiting.

## Results and discussion
### Variation in trait associations at the community level
Clear differences in trait associations along the two main axes were identified. The principal component analysis (PCA) revealed that the traits of leaf nitrogen content and leaf mass per area were primarily loaded along the first principal component (PC1) dimension (Fig. 2a).

These two traits are associated with the acquisitive-conservative trade-off of the leaf economics spectrum (LES), with higher PC1 indicating a tendency towards more acquisitive strategies in the species comprising the community[9,10,17]. Hydraulic traits of xylem embolism resistance, sapwood conductivity and leaf area to sapwood area ratio also loaded along this same axis. Each of these functional traits contributed 15.8–20.8% to the variation in PC1, and they were strongly correlated with each other (Pearson correlation coefficient = 0.46–0.79; all *p* < 0.001; Fig. 2b). These community-level associations are consistent with the link between acquisitive-conservative strategies at the leaf-level and the wider plant structure reported at the species level[13,33].

The other aspects of hydraulic strategy displayed associations largely orthogonal to PC1. Leaf turgor loss point, wood density and xylem cavitation slope primarily loaded onto PC2 (Fig. 2a, b). We relate this axis to the isohydricity spectrum (Table 1), with higher PC2 values associated with a more anisohydric strategy (more negative leaf turgor loss and higher wood density). Communities with a more negative

average leaf turgor loss point and a higher average wood density scored higher on this axis, while communities with a lower average slope of embolism vulnerability curve had lower scores (Fig. 2a). More robust xylem (i.e., more negative P50) was associated with higher wood density, but the correlation was relatively weak and P50 and wood density largely loaded orthogonally in the PCA, consistent with wood density being a complex trait which is associated with many ecological processes[10,34,35]. The cavitation slope had an extremely weak correlation (Pearson correlation coefficient = −0.03) with wood density and also had a relatively low correlation (Pearson correlation coefficient = −0.32) with leaf turgor loss (Fig. 2b). We observed a relatively weak, but significant, relationship between slope and leaf turgor loss, which was consistent with an earlier finding at the species level that stomatal closure is closely coordinated with hydraulic failure due to xylem embolism[36]. Altogether, this indicates less tight coupling of the traits along PC2 than those along PC1.

## Species to community scaling

The community-level PCA displayed similar loadings of traits along two main dimensions of the PCA to that at the species level (Figs. S2a and S2b). This result is consistent with our first hypothesis, H1, that "the strategic variation at the species level will be reflected in the variation of trait associations at the community level" (Figs. S3 and S4). There was, however, 9.1% (64.2% vs. 55.1% for the community level and species level, respectively) more explained variance on the community level (Fig. 2a) when compared to the species level (Fig. S2a), which resulted from a higher loading contribution along the first dimension when scaling up. Leaf nitrogen content was more closely coupled to structural strategy (e.g., xylem conductivity and leaf area to sapwood area ratio) at the community level (Fig. 2a) than at the species level (Fig. S2a). This may reflect the optimisation of resource usage (e.g., competition for light and forest growth) at the species level affecting forest community-level properties[13,37]. That the axis associating leaf economics with wider plant structure and the axis we link to the isohydric spectrum are both found at the species and community levels appears to suggest that the process of community assembly does not allow communities to diverge substantially from the same basic trade-offs that govern individual plant strategy. Therefore, the functional trait space that needs to be taken into account in large-scale assessments of forest properties and ecosystem functions remains consistent at individual and community extents[38,39].

The functional trait space at the community level (Figs. 2a and S4a) showed a clear split among three traditional plant functional groups based on leaf type and habitat (broadleaved evergreen, broadleaved deciduous and needle-leaved conifer), which was less apparent at the species level (Fig. S4). This split was especially clear between conifer and broadleaf trees, with broadleaf deciduous almost universally falling on the acquisitive side of PC1 and broadleaf evergreen on the anisohydric end of PC2. These general patterns were also reflected at the species level but with more overlap. These species-level patterns of more acquisitive trait associations in broadleaved deciduous trees and more conservative ones in conifers are consistent with previous studies[9,17]. However, our results suggest that environmental filtering accentuates this differentiation among the trait associations of functional groups when assessed at the community level.

We tested the gap-filling method used for each trait across species by validating different proportions of missingness (0.2–0.8 for the species that have observed trait values), finding high-coefficients of determination (mean $R^2$ = 0.62–0.94; Fig. S5a) and low root mean squared error (RMSE) (mean RMSE < 1; 0.3–0.86; Fig. S5b) for all comparisons. Generally, we found the differences to be relatively small compared to the trait values from the observations. Additionally, the trait coordination principles of forest communities reported above were robust with respect to the three methods used to gap-fill traits after species matching (Fig. S6) and to the grid cell size

used (details in the "Methods" section; Fig. S7). Furthermore, Procrustes tests demonstrate the loading values of the main axes of the PCA were similar for the comparisons between different grid sizes and different gap-filling methods at the species level (all $p$ values < 0.001; Fig. S8) and at the community level (all $p$ values < 0.001; Fig. S9). They were also robust to excluding forest communities with <3 plots (19% of the 12,452 communities), which led to similar patterns of trait variation (Fig. S10a), contribution to the two principal components, and Pearson correlation coefficients between the traits (Fig. S10b).

## Geographic variation of trait associations

We found distinct geographic patterns of the trait association variation across the regions (Fig. 3). Trait associations consistent with an underlying acquisitive strategy (or higher PC1 scores) were mainly distributed in central and eastern USA and Northern France. Whilst associations linked to a conservative strategy (or lower PC1 scores) were distributed in the western USA, boreal regions, and the Mediterranean Basin (Fig. 3). More extreme values of PC1 were found in the USA than in Europe, which might be a result of the broader variation in environmental conditions (e.g., more water stress and higher temperatures)[40], a homogenising effect of potentially more pervasive forest management in Europe[41] or different evolutionary histories. Higher PC2 scores, linked with anisohydricity, were broadly distributed across southwest-south USA and south-western Spain, the areas where broadleaved evergreen species tended to dominate the forest composition (Fig. S11). The tendency towards more anisohydric strategies identified in these areas differed from a previous study, based on diurnal variations in vegetation optical depth from satellite data (VOD), which found them to be more isohydric in character[42]. This disparity may arise from scale-induced differences in the ecosystems considered, with our results only focusing on tree communities and based on measurements at the tree scale, whilst the 25 km scale of the VOD pixels integrates across all vegetation types. Lower PC2 scores, suggesting a prevalence of more isohydric trait associations, were widely distributed in the forest ecosystems across the two continents, especially in central USA and Sweden, where conifers and broadleaved deciduous trees dominate the forests. This suggests that forest communities in these regions may be more inclined to chronic responses to water stress, characterised by resource limitation following stomatal closure[6,20]. There are, however, large areas in the eastern USA and Central Europe that do not tend towards either strongly anisohydric or isohydric trait associations at the community level. This result may reflect underlying species-level strategies that sit in the middle of the spectrum or may arise from the coexistence of species with different isohydric strategies making up similar proportions of the same community.

## Climatic factors drive variation of trait associations

We found that water demand due to rising temperatures could be more crucial for temperate forest communities than water availability (Fig. 4). We carried out a relative importance analysis to assess the contribution of five climate variables related to water availability and demand to the two dimensions of the PC scores (Fig. 4a, d). Contrary to our expectations (Hypothesis 2), aridity, as quantified by the aridity index (AI), was not a strong determinant of the community-level trait averages (Fig. 4b, e). Instead, we found that higher mean summer temperatures were positively associated with PC1 scores (conservative to acquisitive strategy) (Fig. 4c), whilst higher mean annual temperatures were positively associated with PC2 scores (water storage capacity to anisohydricity) (Fig. 4f). These positive associations with temperature variables may indicate that the adaptation of temporal forest communities relies more on water demand than water availability (or aridity index). However,

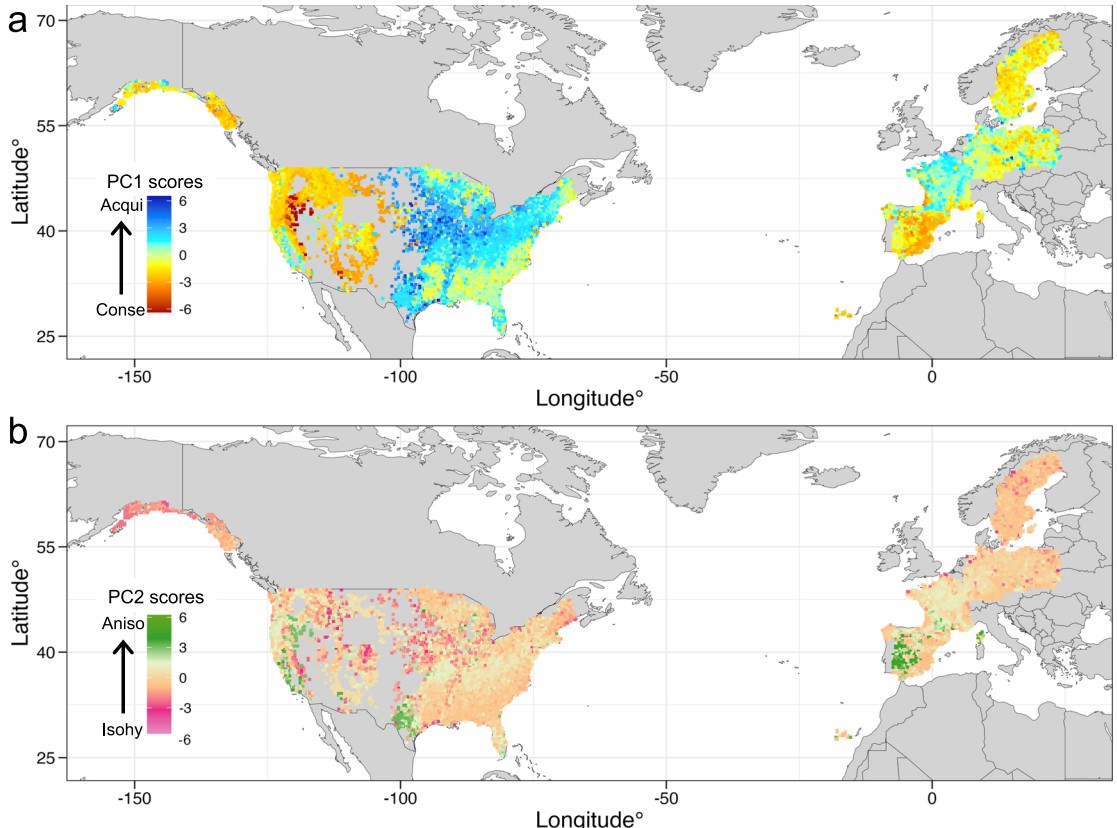

**Fig. 3 | Geographic distribution of the trait associations of forest communities.** There were 12,452 forest communities used in our analysis. The arrows indicate the trait associations of **a** conservative (Conse) to acquisitive (Acqui) and **b** Isohydric (Isohy) to Anisohydric (Aniso) separately. The principal component (PC) scores were calculated using the PCA coordinates in the main two components (Fig. 2a) to obtain the eigenvalue for each community.

the relationships between PC scores and climate factors were also relatively weak, aligning with the patterns observed in global trait–climate relationships[43]. These weak relationships with climate variables suggest that the trait associations of temperate forest communities that emerge may result from functional adaptations to other aspects of environmental conditions. One such candidate variable would be nutrient availability, as recently reported for the Amazon forest[44].

Although aridity was not a strong predictor of PC1 or PC2, we observed a divergence in trait variation of both PC1 and PC2 in more arid habitats (low aridity index, Fig. 4b, e), while there was a tendency to converge on a relatively narrow functional trait space with increasing water availability (higher aridity index, Fig. 4b, e). This supports Hypothesis 3 that the trait associations of forest communities diverge when water is limiting, reflecting an underlying broad range of species-specific water stress strategies to survive in these conditions. Local conditions of temperature, elevation and soil nutrient availability have been previously shown to explain the community-weighted variance of functional traits[45,46] and trade-offs related to these may be influencing the trait combinations emerging at the community level[19]. Higher spatial resolution work will be necessary to identify and distinguish which factors cause one water stress strategy to prevail over another, leading to these community-level patterns. Additionally, a larger coverage of species with observations of functional traits, in particular hydraulic traits, will be beneficial for evaluating the adaptations of forest communities under water stress. The same applies to niches within communities; we found a similar or higher variance of individual traits within communities as opposed to across them (Fig. S12). This, in turn, will be essential for understanding how forest function is likely to change under increasing water stress in the future.

## Future challenges

Although the dataset here is one of the broadest-reaching for water stress-relevant traits yet assembled, we still found data gaps (ca. 3–28%; detailed trait missing in Table S1) after trait imputation by genus and family for the forest tree species, even in these well-studied North American and European forests. Although our sensitivity tests showed robust results with respect to the trait imputations based on phylogenetic signal, further collection and compilation of field measurements will reduce uncertainty in upscaling assessments such as those presented here. Such measurements would not only help to better quantify intraspecific or interspecific trait variation across biogeographic regions[47,48] but also allow better parameterisation of process-based models to explore linkages with forest function. Our selection of traits, being driven by considerations of modelling tree form and function from first principles, did not include those that clearly map onto the height dimension that has been identified by other large-scale trait analyses[17,49]. However, the height dimension is one that is hypothesised to emerge from the application of traits related to resource acquisition, resource allocation and tissue turnover and mortality (Table 1) within a process-based vegetation model. Exploration of the structural and behavioural outcomes of functional strategies by coupling large-scale trait analyses with suitable process-based models[50–52] could facilitate a more complete picture of plant form and function.

Identifying how functional traits respond to water stress is crucial, which may lead to apparent discrepancies in the exploration of trait trade-offs and main functional strategies. The eight traits in our study were selected on the basis of potential mechanisms outlined the previous literature[7,17], and data availability[53]. Here we have examined the strategies of woody plants, however, herbaceous species were not taken into account. Additionally, we excluded some other traits that

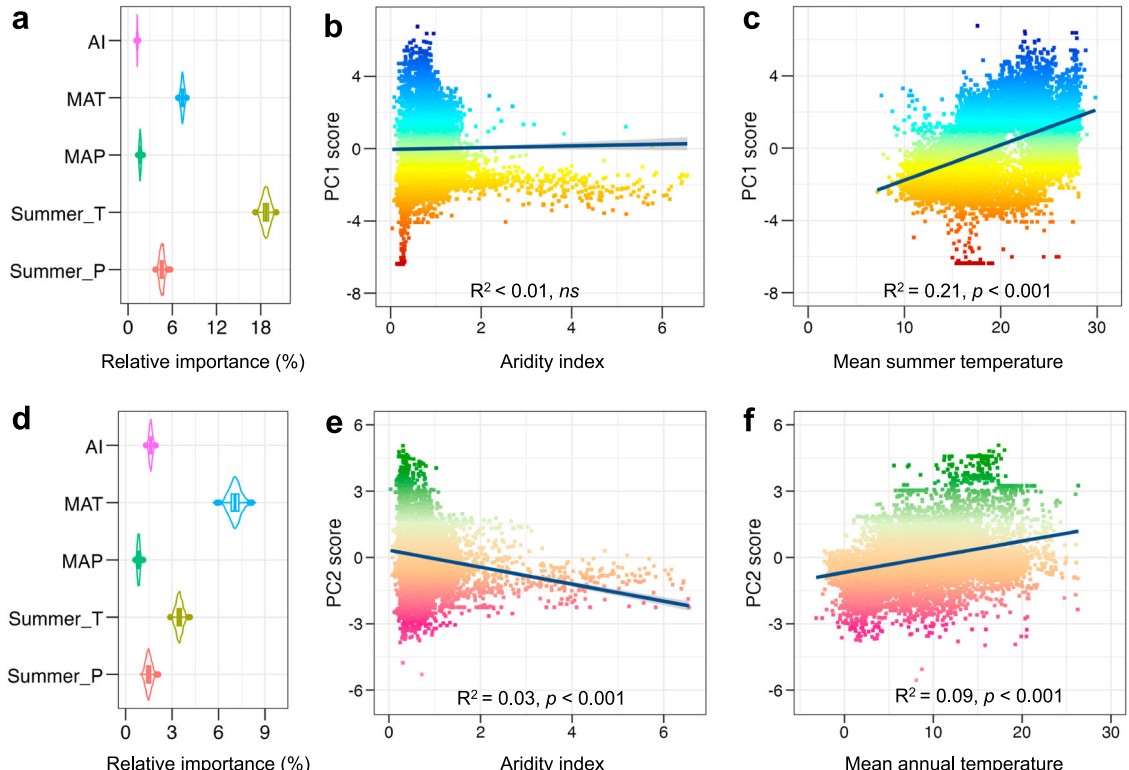

**Fig. 4 | Effect of climatic factors on the trait associations of forest communities (Principal component (PC) scores: PC1 in upper panels and PC2 in lower panels). a** and **d** Result of the relative importance analysis of the individual climate variable for the trait associations along PC1 or PC2. A total of 12,413 forest communities are selected and the number of bootstrap samples is 1000 in the relative importance analysis. Violin plots show the probability density of the bootstrapped importance values. The boxplots indicate the interquartile range (the first quartile (Q1) and the third quartile (Q3)), which includes the median (50th percentile) of the bootstrapped estimates of the relative importance values for the five climatic variables. The whiskers represent the minimum and maximum values within 1.5 times the interquartile range from Q1 and Q3, respectively. **b** and **e** The linear regressions with the aridity index, as hypothesised in the introduction. A total of 12,433 forest communities are used in the analysis. **c** and **f** The regressions with the most important climatic variable for the PC1 (mean summer temperature: Summer_T) and PC2 (mean annual temperature: MAT) separately. The regression tests are two-sided and the regression lines and shading are presented as the means and standard errors, respectively. A total of 12,434 forest communities are used in the analysis. The shading in panels **b**, **c**, **e** and **f** follows the same colour legends as in Fig. 3 for PC1 and PC2, respectively.

we anticipated to be important for water stress strategies, such as rooting depth[54], root-to-shoot ratio[55] and minimum leaf conductance, the latter influencing the rate of plant desiccation[56]. The difficulty of collecting these traits is reflected in the low sample size available in the literature. While wood density is widely studied, it is only a weak indicator of potential capacitance[35], because it is still not known how much stored water is accessible. Likewise, we were unable to use direct measurements of plant capacitance due to the limited availability of suitable observations. Information on rooting depth, minimum leaf conductance and capacitance may bring a new angle to the study of the functional trait space in response to water stress, as may trade-offs with other aspects of plant function[44].

Scaling up individual or species-level traits to the community level may only capture the average of trait values, which approximate local strategies. The intraspecific trait variation in response to environmental conditions was not included within the method of community-weighted mean traits. New approaches (e.g., normalising plant traits per unit land area) to integrate community traits and their contextual information are necessary[38]. Finally, it is possible that the results in some regions may be strongly influenced by forest management. For instance, the south-eastern US, which is an area with intensive plantations and regular cutting, might be expected to alter the natural selection of strategies[57]. Identifying a broad gradient of plots with a long history of being unmanaged would be valuable to separate natural biogeography from human management decisions. Further research is required to address these above-mentioned challenges and

to improve the understanding of functional strategies with respect to global-scale climate change in the future.

While the mechanisms of tree- or species-level strategies to water stress have been increasingly clarified in recent research[7], upscaling these strategies into associations at the level of forest communities and mapping these across regional scale has been missing to date. Our study integrated species-level traits into forest communities across the regions to characterise the emerging functional trait associations and to explore the variability across two continents. Our results demonstrate consistency in trait coordination at both the species level and community level, which highlights the potential of applying these functional traits to explore forest biodiversity patterns and ecosystem functions at the community level or ecosystem level[38,39]. The identified geographical patterns of trait associations could be used as a basis for parameterising forest hydraulic function in process-based vegetation models, allowing us to make predictions of forest functions that are tightly grounded in observed community trait associations. This could include, for instance, assessing how water stress affects the current and future forest carbon sink.

We found that traits relating to xylem embolism resistance, sapwood conductivity and tree structural strategies collapsed into a single axis of variation, loading into the widely reported acquisitive-conservative spectrum (or fast-slow continuum). However, the xylem cavitation slope, leaf turgor loss point and wood density loaded onto an orthogonal axis, are associated with the spectrum of isohydricity and, tentatively, water storage. This complexity in functional strategy

implies a similar complexity in the response of forest functions to water stress. Our results suggest that trait associations at the community level are more closely related to water demand than water availability. However, the relationships with climate are relatively weak, and the proliferation of different strategies in more arid environments, along with the limited explanatory power of climate variables, raises the question of what factors exert the primary filter governing which water stress strategy comes to dominate a forest community. Whilst, to some extent, this filter may be determined by small-scale heterogeneity in edaphic and climatic conditions, it also indicates that the water stress strategy may be influenced by the trade-offs among multiple functional traits across a much wider range of environmental conditions. Unpacking this unexpectedly complex environmental filter will be a major challenge for functional biogeography, but one that is necessary to overcome if we are to understand how forest composition and functions are likely to adapt to novel environmental stress.

## Methods

### Forest inventory data and plot aggregation

We used forest inventory data from the USA and six countries in Europe (Spain, France, Germany, Czechia, Poland and Sweden) (Table S1). These forest inventory data included the approximate geographic locations of the forest plots and, for each individual living tree, the species and diameter at breast height (1.3 m). The large spatial-scale data covers the temperate, Mediterranean (Spain and western USA) and boreal biomes (Sweden and Alaska) across their climate gradients. To keep the forest inventory data temporally consistent between countries, censuses closest to the year 2010 ( ± 3 years) were selected since the majority of the forest inventory data were available at that period. We used a total of 219,787 forest inventory plots across the USA and Europe in our analyses. We selected the living trees in each plot and included only those with a diameter at breast height larger than 12.7 cm across all the datasets. We aggregated inventory plots into 0.25° grid cells to dampen variation induced by the small sizes of plots and to provide a consistent spatial unit across all the countries. We term the sample of trees in a grid cell as a community (mean plot number per community = 17.3; Fig. S1). We selected all the woody plants (trees and shrubs) in each community based on the woodiness information from TRY and the Woody Plant Database (http://woodyplants.cals.cornell.edu). Species scientific names were standardised to the Plant List (http://www.theplantlist.org/) according to the *plantlist* package in R (version 0.8.0)[58]. The cactus species *Pilosocereus royenii* was removed. In total, 643 woody species were retained for the analysis. To further distinguish the different strategies of woody plants, broad leaves were split into two groups according to their leaf phenological types (evergreen and deciduous). For the conifers, the species count is not as rich as for the broadleaf trees, with only eight conifer species being deciduous. We thus classified woody species into 3 functional groups: (1) Broadleaved Deciduous (BD); (2) Broadleaved Evergreen (BE) and (3) needleleaved Conifer (NC). Of these 643 species, there are 307, 215 and 121 for BE, BD and NC, respectively.

### Calculating functional traits at the species level and community level

Eight key functional traits of woody species at the species level were selected for this analysis (Table 1). The trait data at the species level were requested from the TRY dataset[53] and supplemented by a literature review of 75 research publications searched in Web of Science and Google Scholar until May 2020 using the search terms "Xylem water potential at 50% loss of conductivity", "Turgor loss point", "slope of the vulnerability curve", "Xylem conductivity" and "Leaf area to sapwood area ratio" separately (Supplementary Data 1). Similar to the woody species in the forest inventory data, species scientific names were standardised to the Plant List according to the *plantlist* package

(version 0.8.0) in R[58]. We then combined the traits with the species composition in each community, and median values for each species-level trait were matched to the species present in the forest inventory data. After this initial species matching, the amount of missing trait data differed greatly across the eight selected traits (Table S2). For well-studied traits such as leaf nitrogen content and wood density, about 40% of the woody species were missing, rising to 60%-80% for the hydraulic traits. For the species-level functional traits that were missing, we used the median trait values of the same genus instead. And if the trait was still missing, we used the median values of the family to fill the gaps[59]. However, there were still some species for which the trait data were missing (3%-25%). We imputed the remaining trait values for the species using the *funspace* package in R (version 0.1.1)[60], which is coupled with phylogenetic information to impute missing trait data[61]. We carried out sensitivity analyses to explore the consequences of this imputation for our results (see the "Statistical analysis" section below).

We calculated mean values for each functional trait for each community, weighted by the relative basal area of the species in that community[25]. The functional traits, except wood density, were log-transformed to increase the symmetry of their distribution before the calculation of weighted mean values. For turgor loss point (TLP) and xylem embolism vulnerability (P50), we first multiplied trait values by −1 to obtain positive values before log-transformation. Furthermore, we calculated the basal area for each functional group in a community and compared this to the total basal area to obtain the relative proportion for each functional group. Based on this, forest functional types at the community level could be identified according to the relative proportion of dominant groups representing more than 50% of the grid basal area[62]. 215 forest communities were removed from the analyses because the dominant functional type was <50%. In total, we used 12,452 communities in our analyses.

**Climate factors across regions.** Water availability and temperature are crucial in determining forest function, diversity and productivity[63]. Here we considered the pattern of trait associations that are strongly related to the water availability based on annual precipitation amounts and summer precipitation, as well as to the aridity index (ratio between precipitation and evapotranspiration) and to water demand caused by evapotranspiration based on mean annual temperatures and summer temperatures. The climatic variables of mean annual precipitation (MAP), precipitation of driest quarter (Summer_P), mean annual temperature (MAT) and mean temperature of warmest quarter (Summer_T) were derived from the WorldClim dataset (https://www.worldclim.org/data/bioclim.html)[64]. The aridity index (AI) was obtained from the Global Aridity and PET Dataset (https://cgiarcsi.community/data/global-aridity-and-pet-database/), which provides local water availability (ca. 1 km resolution) considering both evapotranspiration processes and rainfall deficits (version 3)[65]. Higher AI values indicate more water availability (less water stress), while lower AI values indicate less water availability (more water stress). Because the accuracy of forest plot locations is fuzzed, typically within ca. 1–2 km, we used the climate factors from WorldClim with the spatial resolution of 2.5 arc minutes (~5 km at the equator). For the aridity index, we aggregated the resolution from 30 arc second to 2.5 arc minute resolution. We extracted the climate factors for each plot individually and then aggregated them into a grid cell (0.25° × 0.25°) by taking the mean of their values to study the patterns of trait association variation across the regions.

### Statistical analysis

**Principal component analysis (PCA).** All eight functional traits were selected a priori before the PCA test based on these traits being strongly linked to acquisitive-conservative, structural, stomatal and water storage strategies in the literature (Table 1). We also took into

account the importance of these traits, or aspects of function that these traits act as proxies for, in process-based vegetation modelling[50]. The selected traits were scaled and centred before running PCA tests. The PCA was conducted to explore the main axes of trait variation at the species level and community level using the *FactoMineR* package in R (version 1.34)[66]. This procedure helps to identify the main plant strategy spectrum among the functional traits. After this, we checked the consistency of the functional trait variation between the species level and community level based on the percentage of explained variance. For the PCA test at the community level, we also calculated the contribution values of each trait loading to the two main axes. In addition, we also calculated the correlations between community-level functional traits using the *corrplot* package in R (version 0.92)[67] to visualise the correlations. To further quantify the trait associations of forest communities along the main axes, we calculated the PCA coordinates in the main components to obtain the eigenvalue for each grid according to the score function. The contribution of each community-level trait to these two dimensions was separately calculated (PC1 and PC2 scores).

**Relative importance of the climate variables for the trait associations.** The relative importance analysis (RIA) was used to evaluate the relative importance of climatic variables (e.g., AI, MAP, MAT, Summer_T and Summer_P) in relation to the two forest trait associations at the community level using the *relaimpo* package in R (version 2.2-6)[68]. We removed the forest communities that did not have climate information, leaving 12,409 communities in the relative importance analysis. To determine the relative weights of these five climatic factors in a linear regression model, we used the functions boot.relimp and booteval.relimp from the *relaimpo* package to compute the bootstrapped estimates of the relative importance of each climate factor and the confidence interval separately. Here, the type parameter was set to the "lmg" method[69], which is a commonly suggested approach for determining relative importance. The number of bootstrap samples was 1000 in our analysis.

**Sensitivity analyses.** In order to test the effectiveness of the gap-filling method, for species for which we had observations for a particular trait, we randomly selected seven different proportions (from 0.2 to 0.8, increasing each time by 0.1) of the species for which we had observations of trait values (missingness proportions) and predicted their values from the remaining species for which we had data for their trait values, following the gap-filling method described above—i.e. the median of genus and family and phylogenetic relationship (default, as described above). Then, we randomly ran this 100 times to obtain the coefficient of determination ($R^2$) and root mean squared error (RMSE) values of z-transformed predicted versus observation trait values as an indicator of overall prediction accuracy for the trait gap-filling[70,71]. Lastly, we evaluated the mean $R^2$ and RMSE values. Here, we did not use very low proportions of missingness (<0.2) in order to avoid introducing bias resulting from a low number of observations. We also tested gap-filling the missing trait values using three different methods: (a) the median of the genus and family and phylogenetic relationship, (b) median of the genus and phylogenetic relationship and (c) only the phylogenetic relationship. We used Procrustes analysis using the *vegan* package (version 2.6-4)[71] to check the similarity of PCA tests by comparing the main PC loading values (1) among the different gap-filling methods described above and (2) between community level (0.25° grid) and species level. Moreover, we ran sensitivity tests to assess the influence of different grid sizes (0.1° and 0.5°) at the community-level trait means. Furthermore, we conducted sensitivity tests to evaluate whether a low number of forest plots in some grid cells influenced the trait variation along the

first two dimensions and trait correlations. For this, we re-ran the PCA tests excluding 2,375 forest communities that had <3 forest plots.

All analyses in this study were conducted in R (version 4.2.3) (R Core Team 2023) and figures were produced using the *ggplot2* R package (version 3.4.3)[72,73].

**Reporting summary**
Further information on research design is available in the Nature Portfolio Reporting Summary linked to this article.

## Data availability

The raw NFI data underlying the analyses are available for open access for the United States of America (https://research.fs.usda.gov/products/dataandtools/tools/fia-datamart), France (https://inventaire-forestier.ign.fr/dataifn/?lang=en), Germany (https://bwi.info/Download/de/) and Spain (https://www.miteco.gob.es/es/biodiversidad/temas/inventarios-nacionales/inventario-forestal-nacional/cuarto_inventario.html). Raw NFI data for the other countries are available by concluding data access agreements with the agencies owning the data; Sweden (https://www.slu.se/en/Collaborative-Centres-and-Projects/the-swedish-national-forest-inventory/), Poland (https://buligl.pl/web/buligl-en/w/national-forest-inventory) and Czechia (https://www.czechterra.cz/). The community-level traits, PC scores (e.g., PC1 and PC2) and climate variables for this analysis can be found in the Zenodo repository: https://zenodo.org/records/13757078.

## Code availability

The R code for the statistical analyses and generating the figures is available in the Zenodo repository: https://zenodo.org/records/13757078.

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

## Acknowledgements

We thank all the people who conducted field measurements and data collection for the forest inventory dataset and those who made their trait data available through the TRY dataset (Request IDs 5395 and 8814). We thank the team behind the Global Forest Dynamics database, initiated by the TreeMort project, on which this study is based. T.A.M.P., N.A, A.E.-M., M.-J.S. and D.L. were financially supported by the European Research Council (ERC) under the European Union's Horizon 2020 research and innovation programme (grant agreement No. 758873, TreeMort). D.L. was also financially supported by the FWF Austrian Science Fund (Lise Meitner Programme M2714-B29). A.E.-M. was further funded by the Royal Society Standard Grant RGS/R1/21115 'MegaFlora', the UKRI/NERC TreeScapes NE/V021346/1 'MEMBRA', the NERC/NSF Gigante NE/Y003942/1 and the FRB/CESAB 'Syntreesys'. N.A. was also funded by the Natural Environment Research Council (NERC, NE/R016518/1) and the European Space Agency BIOMASS Climate Change Initiative+. E.C. acknowledges funding from the project AdAgriF—Advanced methods of greenhouse gases emission reduction and sequestration in agriculture and forest landscape for climate change mitigation (CZ.02.01.01/00/22_008/0004635). This paper is a contribution to the strategic research areas BECC and MERGE funded by the Swedish government and the Nature-Based Future Solutions profile area at Lund University. J.A., P.R.B., M.A.Z. and T.A.M.P. acknowledge funding from the CLIMB-FOREST Horizon Europe Project (No. 101059888) that was funded by the European Union and the Science and Innovation Ministry (Agencia Estatal de Investigación subproject LARGE, No. PID2021-123675OB-C41). C.Z. acknowledges the financial support from the Academy of Finland (340744). S.S. acknowledges funding from the European Union's Horizon 2020 research and innovation programme under the Marie Skłodowska-Curie grant agreement No 895158 (ForMMI). This research was supported in part by the U.S. Department of Agriculture, Forest Service. The findings and conclusions in this publication are those of the authors and should not be construed to represent any official USDA or U.S. Government determination or policy.

## Author contributions

D.L. and T.A.M.P. designed the research with inputs from J.P.S., T.J.M., N.A., A.E.M. D.L. and T.A.M.P. led the writing of the manuscript. A.E.M., J.A. and M.J.-S. standardised the forest inventory data and assisted in data preparation and cleaning, with contributions from S.S., J.A., C.Z., M.A.Z., and P.R.B. D.L. led the data combination and statistical analysis. C.W.W., S.S., J.A., P.R.B., M.A.Z., E.C., J.F., G.K., A.T. and M.J.S. curated the forest inventory data. All the co-authors commented on analytical approaches and edited the draft and final manuscripts.

## Competing interests

The authors declare no competing interests.

## Additional information

[1]School of Geography, Earth and Environmental Sciences, University of Birmingham, B15 2TT Birmingham, UK. [2]Birmingham Institute of Forest Research, University of Birmingham, B15 2TT Birmingham, UK. [3]Department of Botany and Biodiversity Research, University of Vienna, Rennweg 14, 1030 Vienna, Austria. [4]National Centre for Earth Observation, University of Leicester, LE4 5SP Leicester, UK. [5]Institute for Environmental Futures, School of Geography, Geology and the Environment, University of Leicester, LE1 7RH Leicester, UK. [6]Universidad de Alcalá, Departamento de Ciencias de la Vida, Grupo de Ecología y Restauración Forestal (FORECO), 28805 Alcalá de Henares, Spain. [7]IFER - Institute of Forest Ecosystem Research, Cs. Armady 655, 254 01, Jilove u Prahy, Czech Republic. [8]Global Change Research Institute of the Czech Academy of Sciences, Bělidla 986/4b, 603 00, Brno, Czech Republic. [9]Department of Forest Resource Management, Swedish University of Agricultural Sciences, SE901-83 Umeå, Sweden. [10]Univ. Grenoble Alpes, INRAE, LESSEM, F-38402 St-Martin-d'Hères, France. [11]Centre for Ecology, Evolution and Environmental Changes/Azorean Biodiversity Group/CHANGE—Global Change and Sustainability Institute and Universidade dos Açores—Faculty of Agricultural Sciences and Environment, PT-9700-042 Angra do Heroísmo Azores, Portugal. [12]Universidad de Alcalá, Departamento de Geología, Geografía y Medio Ambiente, Grupo de Investigación en Teledetección Ambiental, 28801 Alcalá de Henares Madrid, Spain. [13]Wageningen University and Research, Wageningen Environmental Research (WENR), Droevendaalsesteeg 3, 6708PB Wageningen, The Netherlands. [14]Natural Resources Institute Finland (Luke), Latokartanonkaari 9, 00790 Helsinki, Finland. [15]Forest and Natural Resources Research Centre/Taxus IT, ul. Płomyka 56A, 02-491 Warszawa, Poland. [16]The United States Department of Agriculture (USDA) Forest Service, Northern Research Station, NH 03824 Durham, USA. [17]Optics of Photosynthesis Laboratory, Institute for Atmospheric and Earth System Research (INAR)/Forest Sciences, Viikki Plant Science Centre, University of Helsinki, Helsinki 00014, Finland. [18]Department of Physical Geography and Ecosystem Science, Lund University, Sölvegatan 12, 22362 Lund, Sweden. ✉e-mail: cqliudaijun@gmail.com

