## [Peer Review File · Nature Communications]

Mapping multi-dimensional variability in water stress strategies across temperate forestsREVIEWER COMMENTS

Reviewer #1 (Remarks to the Author):

Review of the manuscript entitled Mapping multi-dimensional variability in water stress strategies across temperate forests by Liu et al.

Overall statement and summary: Overall I enjoyed reading the manuscript.

The authors discuss a highly relevant problem of forests' reactions to water stress in times of its increasing frequency and severity.

Their approach is to use in-situ measured functional traits. I would consider this approach to be appropriate, since plant traits are known to reflect their resilience to water stress.

I found targeting a specific type of stress of particularly high utility. This allows for a more specific interpretability.

More so, the authors address a central problem of trait-environment interaction, which is the scale mismatch of in-situ measured plant traits and the ecosystem or forest scale at which ecosystem functions are usually measured.

They do so by attributing eight water stress-related traits on a tree scale to those species growing in various forests in the USA and Europe for having forest-wise plant strategies of water stress.

Subsequently these strategies were related to climate factors.

Specifically, the authors show:

1. Woody plants water stress traits vary in two dimensions that explain 64% of their variation. They vary along an axis onto which leaf nitrogen, leaf area to sapwood area ratio (LS), Maximum xylem conductivity per unit sapwood area (Ks), leaf mass per area (LMA) and xylem water potential at 50% loss of conductivity (-p50); and onto a second axis representing mainly wood density (WD), how abruptly embolisms occurs (Slope), as well as leaf turgor loss point (TLP), reflecting leaf water regulation, i.e. stomatal closure. The authors interpret the first axis as reflecting how acquisitive or conservative plants use water. The second axis is condensed to isohydricity.

2. These axes are consistent independently of the aggregation scale: may it be species level or forest community.

3. Forest communities' water stress traits are distributed unevenly in space, but their patterns cannot be well explained by climatic water stress factors.

4. Mean summer temperature correlates slightly with the acquisitiveness, while the mean annual temperature correlates weakly with the isohydricity.

Overall, the findings, methods, and language fit the scope of Nature Communications

I found particularly powerful to add a abundance measure to the in-situ measured traits by weighing by woody plant diameter. The scale of study is with 0.25° small thus climatic heterogeneity within these grids may be considered sufficiently small for merging trait data. The data basis per grid appears to be sufficiently large for containing a representative trait and species subset, and the availability of indirect abundance metrics (diameter) allows for further ecological information. Further the range of two continents allows inferences for large climatic gradients with high resolution and allows interpretable results. One can compare not only the climatic variability, but also make assumptions about the taxonomic diversity in the two distinct land masses.

Having said this, I think this article could profit from a number of changes.

Major comments

[1] Clarity. Although this is a well written manuscript, I recommend a couple of changes.

1.) The whole paragraph in lines 125 – 141, can and should be communicated better. The language as well as the logic should be reconsidered. I gave some suggestions in my comments below.

- 2.) Language. In parts I was missing clarity, I made suggestions where applicable. Particularly, I think you can make the method more accessible by sticking to either the term "forest" or "community" of this aggregation level. Since you are working with different aggregation levels, make it easier for the reader to understand which aggregation you are talking about. For instance, in Fig 2 you talk about "forests across regions" (i.e. species aggregated to cells, to forests), Forests = communities (?). Please use one term only. Fig.3 "forest communities". Fig 4 "forest".
- 3.) ll. 152 and 155. I think you could fuse the expectations as well as the hypothesis, as these should, if I am not mistaken, be the same thing.
- ll. 154. Add for comprehension somewhere before the hypothesis that you also relate these data to environmental data ("and (b) gradients in these strategies across the geographic and climatic space of temperate forests can be identified (Fig. 1c, d)»)
- 4.) Please add at an earlier time than now that species included are woody species. To me it was not clear for a long time what data sit behind the species level. Make this clear earlier and particularly in the figure captions.

[2] Two additional analyses that are simple to implement could shed a light onto discussed aspects.

1.) The authors describe the similarity of the PCA based on species and forest community scale. (in line 210: "The community-level PCA displayed similar loadings of traits along two main dimensions of the PCA to that at species-level (Fig. 2a; Fig. S2a)».)

The within vs. across community variation of plant traits could give further insights into how similar the species' traits are within one forest community vs. across. For example, you could calculate the coefficient of variation per trait across the communities. For example, Joswig et al. 2022, found a relationship of variance explained by environment and coefficient of variation (Figure 42, Supplementary Information).

The authors also describe the weak PC1 and PC2 relationships with climate. This could be due to a high spatial heterogeneity (e.g. geodiversity). This could be indicated with high internal variability. You point to this in ll. 353: "But surprisingly the relationships with climate are relatively weak and the proliferation of different strategies in more arid environments and the limited explanatory power of climate variables poses the question of what factors exert the primary filter governing which water stress strategy comes to dominate a forest community.» ll. 190.

2.) The attribution of trait to the first and second principle component is based on loadings. Other studies attribute some traits you use to other syndromes/axes. For example leaf N and LMA were attributed to leaf economics, e.g. by Wright et al. 2004 (Nature) , Díaz et al. 2015 (Nature). I think the article would benefit from discussing these "double" attributions. This brings me to another point, which are quantifiable trait attributions. I thus suggest adding an analysis, which allows a quantified attribution of these traits to trait groups. I presume the results will be the same as you estimated visually. For example, Joswig et al. 2022 (NEE) show a hierarchical clustering analysis of all trait groups, based on (absolute) Pearson correlation coefficients of trait-trait relationships (figure 1a) that is also adopted in Maynard et al. 2022, (Nature; figure 4a). If you wanted to include this trait group attribution, you could simply transform figure 2c for this purpose (but I would not ask you to do so).

[3] Implications

- 1.) I think the authors could have stressed more the following interpretation, suggesting (ll. 352.) "that community-level water stress strategies are more closely related to water demand than water availability». Possibly this could be added to the abstract.
- 2.) In the discussion, I was missing the picture put together again. Could you «walk» through the different environments and describe the forest communities' adaptations?
- 3.) To which extent do you think your findings from single woody plants are representative of forest

communities? You could discuss possible conflicts and open up important discussions on this matter (non-woody species, temporal variability). Which other traits would be candidates reflecting plant response well, I am sure the authors can do a great job at imagining these (e.g. root to shoot ratio?)

Further comments

Abstract

II. 72: "climate drivers" because of complex biosphere – environment interactions, I would suggest calling these climate "factors".

II. 72: I had difficulties understanding the following sentence. You might want to consider simplifying, else this finding could possibly be put after the description of the trait dimensions, so it might be easier to follow the train of thoughts. "We find a high degree of consistency between community-level water stress strategies and those at species-level». I am sure the authors will find a good way. My idea that I do not ask you to use would be: We find water stress strategies to be consistent across the scales of communities and species.

LI. 75: The term "isohydric» may not be familiar to all readers of Nature Communications. You might want to replace, define it briefly or simply add a simpler term in brackets (e.g. water regulation).

II. 77: To my taste the conclusions could be condensed. A suggestion, which I do not expect you to adapt would be: "These findings have the potential to improve predictions of ecosystem models, particularly for patterns of tree mortality and forest biomass accumulation in a changing climate.»

Introduction

LI. 111: I do not understand the logic in this sentence, and I strongly suggest reformulating it, since one can understand the opposite: "For example, a tree may be structurally acquisitive, investing in a limited water transport capacity for its leaf area, allowing it to instead invest resources in height growth to win the competition for light." Accordingly, an acquisitive, thus "risk-taking" tree could be one of thin or few water vessels, transporting little water to the leaves, which are consequently of smaller size. This would entail, less photosynthetic activity and thus fewer resources that can be invested. Acquisitive strategies should be the ones with large water transport capacity to match the need for water for its large leaf area.

Why not write something along the lines of: "For example, a tree may be structurally acquisitive, investing in a *large* water transport capacity for its leaf area, allowing it to invest *more* resources in height growth to win the competition for light."

I would further add the mismatch with past studies' trait attribution. Traits you put into the acquisitive-conservative axis - often termed plant economics - were attributed to the "size" axis according to Díaz et al. 2015, Joswig et al. 2022. The strong relationships of water transport traits, i.e. leaf area as well as water transport capacity (e.g. conduit diameter) were explained to hydraulic scaling. This should be addressed somewhere

II. 116. Add simpler term of isohydricity. «Other adaptations to water stress (e.g., isohydricity*, or water regulation*)»

II. 126. For better understandability, you might want to consider rephrasing: When» *[object] are* assembled into communities, the*ir* dominant strategy depends on the balance between niche complementarity and resource competition»

II. 128. Language & logic. This sentence appears to me flipped. Wouldn't competition be one interpreted process from the evidence of domineering acquisitive strategies? "Competition between species suggests that as water resources become less limiting, more acquisitive strategies will be

favoured, resulting in a convergence towards a limited range of water-demanding strategies^{15,16,24}.»
Something along the lines of: "Dominant acquisitive strategies can be evidence for competition among species when water is not a limiting factor, resulting in a convergence towards a limited range of water-demanding strategies^{15,16,24}.»

II. I would suggest a different phrasing, since this evokes the idea of temporal change. "In water limited environments, a wide range of water stress strategies, and they are partitioned environmentally according to particular combinations of conditions that favour certain strategies^{25,26}"«

II. 132 I think this has been said earlier, and you could leave this away: How the range of community-level functional strategy changes as a function of water stress is not well established. You could integrate into the next sentence: "In order to study how the range of community level functional strategy changes with water stress, it is fundamental the likely response of the forest to ongoing environmental changes, both in a qualitative sense and to underlie quantitative predictions by vegetation models.»

LI. 136 Same here, I would suggest working on the language and logic. These models are increasingly technically capable to simulate water stress responses^{27–30}, but require clear evidence on the distribution of forest strategies.

II. 138 . Also this sentence is too lengthy.

"An exploration of water-stress strategies of forests along climatic gradients promises a new perspective on forest functional biogeography, as well as it may bear implications for forest management using vegetation models."

LI. 146. Could go into the discussion: " , as the information on rooting traits is very limited»

II. 147. Did you consider using a selection criterion for the number of species present?

Figure 1d. The map seems to be distorted, at least it is not a projection that is common. You might want to change that.

How much overlap in terms of species do you have in the different forest stands?

Did the species used, belong to all growth forms?

II. 185 You might want to mention further associations of traits, e.g. carbon and nutrient cycling/stock related to leaf N, wood density etc.

Results and Discussion:

II. 185: An introductory sentence e.g. relating the findings to the question might be useful.

"Community-level strategies.»

II. 185. Please be more specific and describe the differences (but this may be a matter of taste). "Clear differences in forest functional strategies were identified.»

LI. 199. I don't think a trait can "drive" an axis. Maybe rephrase. "The axis was positively driven by these two traits, while it was negatively driven by a steeper slope of the embolism vulnerability curve (or high sensitivity of the xylem to water stress) (Fig 2c).»

For the discussion: Which aspects of this analysis could have biased your research

II. 202. I would suggest being cautious about claiming "links" based on correlations. With the analysis

(Pearson+hierarchical partitioning) you could refer to the correlation (see major comments above). «Contrary to the general expectation that more robust xylem was linked to higher wood density³⁶, the slope had extremely weak correlation (Pearson correlation coefficient = -0.04) with wood density and had also relatively low correlation (Pearson correlation coefficient = -0.33) with leaf turgor loss (Fig. 2c).»

Figure 2. For an easier link between figure 2 and figure 3, you could try to make this graph a quadrat. The reason I suggest this are the identical PC1 and PC2 scores (from -6 to 6) in figure 3, in the original PCA in figure 2, however, PC1 seems to have a greater variation than PC2.

Caption: Please add the number of species/samples/forest sites to the figure. There should be an indication here or better before in the text about which type of species are integrated. E.g. which growth forms.

Please add the long version of the trait abbreviations.

Two suggestions to allow a quadrat shape of 2a:

(1) Including hierarchical partitioning: top row quadratic fig2a, next to 2b, bottom row hierarchically clustered (optional) 2c

(2) Alternatively, you could probably leave out 2b (put into the appendix) without much loss, since the loadings are displayed in 2a.

II.212 possible rephrase the hypothesis quickly, so the reader does not have to go back

Figure S5. Size of grid cell – this allows a thorough methodology.

II. 219 This is interesting; possibly I misunderstand this. Yet: how would it be possible for a community to have a larger PC1 value than an “extreme” PC1 species? Once aggregated, these communities, consisting of single species/plots should have consequently a smaller trait range. Do you think this might be because of the trait combinations projected? Maybe one of the other reviewers might give advice of the possibility to compare absolute numbers (PC loadings, range).

My suggestions would be to look at the trait-wise ranges and compare species vs. community and check if this pattern reoccurring (but I would be surprised).

You could exclude the possibility of this pattern being a necessary mathematical consequence of combining different species/plots with a randomization analysis (e.g. randomly attribute species/plots to communities then perform PCA at community scale).

Please add reference to the figures you are referring to.

«Additionally, the full range of PC1 score was wider at community-level* (Fig. 2a)* than that of species-level *(Fig. S2a)*, suggesting this spectrum at community-level has larger trait variation under environmental variability^{38–40}.»

II. 221 It is a nice result to see the different leaf types separated. Wouldn't it be enough in the future to reduce the trait information to leaf type only to assess the water stress strategy? How much do you additionally to the leaf type explain by trait variation?

«The strategy space at community-level (Fig. 2a; Fig. S4a, b) showed a clear split among three traditional plant functional groups based on leaf type and habit (broadleaved evergreen, broadleaved deciduous and needle-leaved conifer), which was less apparent at species level (Fig. S2a; Fig. S4c, d).»

LI 250: You could add very briefly how you came about with this result. *Spatial variation in functional strategies.*

II. 254. Nice observation. A further explanation could be taxonomic peculiarities in the North America.

“More extreme values of PC1 were found in the US than Europe, which could be linked to the broader variation in environmental conditions (e.g., more water stress and higher temperatures)⁴³ or possibly a homogenising effect of the more pervasive forest management in Europe⁴⁴.»

II. 262: This is a very important comparison. Another reason for these different findings may consist of the different traits measured by remote sensing (fewer “size” traits, more economics CHECK xxx), as well as a number of factors like understory vegetation, abundance, temporal variation, or seasonality. “ This disparity may arise from differences in the ecosystems considered, with our results only representing tree communities, whilst the 25 km scale of the VOD pixels integrates across all vegetation types.»

LI. 340 “potential of applying these functional traits to explore forest biodiversity patterns and ecosystem functions at community-level or ecosystem-level^{41,42}” What does the choice of traits influence, which traits should be added in the future to better detect? Where to measure next?

LI. 342: I agree: “The identified geographical patterns of functional strategy could be thus used as a basis for parameterising forest hydraulic strategy in process-based vegetation models, allowing us to make predictions of forest function that are tightly grounded in observed community strategies. This will further enable, for instance, to assess how functional strategy affects the current and future forest carbon sink.»

LI. 359: Please rephrase: “Whilst to some extent this filter may be exerted by small-scale heterogeneity in edaphic and climatic conditions, there are also indications here that water stress strategy may be influenced by functional trade-offs across a much wider range of environmental conditions.»

Methods:

II. 366: *Forest inventory data.* Be very explicit about what inventory data is in one sentence. Something like: “We used forest inventory data*, which are same-sized plot data including their GPS - location and woody species’ presence and tree or shrub trunk basal area. These data *are from ...”

II. 366 ff. Please add the size per plot.

II. 400. Not a sentence. Change. “While the missing gaps were especially high (70%-90%) for these hydraulic traits.»

II. 407. Be more specific about “see below”, add the paragraph name, and add results to the appendix. “We carried out sensitivity analyses to explore the consequences of this imputation for our results (see *paragraph Statistical analysis*).*»

II. 399 and II. 458. The gap-filling procedure as well as the approach of the sensitivity analysis applied appear valid to me. I would like to see the figures, or at least the specific test results of Procrustes tests as described below. To have the approach solidified, a further analysis could be made for data observed: Compare the error of (observed) species-level traits when filled by genus mean. You could further compare Fig.2a with (1) (originally observed) species filled genus-level data as well as (2) observed species level, and your original Figure 2a with the Procrustes test.

LI. 375 It is a impressive resolution (0.25°)

LI 447 I am not familiar with the relative importance analysis, and hope for my co-referees to judge

this method.

II. 277 I would suggest to add the figure references directly. "We found that water demand due to rising temperatures could be more crucial for the temperate forests than water availability *(figure 4)*.»

II. 280 Did you consider to filter to the 20% most arid places? For habitats with low aridity there may be other limitations than water. "Contrary to our expectations, aridity, as quantified by the aridity index (AI), was not a strong determinant of community-level hydraulic strategy (Fig. 4b, e).»

II. 282. Could you describe the different habitat types (or difference) these climatic variables represent?

«Instead, we found that higher mean summer temperatures were positively associated with PC1 scores (Fig. 4c), whilst higher mean annual temperatures were positively associated with PC2 scores (Fig. 4f).»

Reviewer #2 (Remarks to the Author):

General comments

The manuscript by Liu et al. is well written and addresses a very timely issue with an impressive dataset. Based on data from 219,518 forest inventory plots comprising 12,445 communities, they studied trait associations among a set of eight functional traits linked to the conservative-acquisitive spectrum and to drought response. A main focus is the comparison of trait associations on species and on community level. In addition, the paper discusses relationships between axes of shared trait variance on community level and the environment, as well as differences in traits between communities dominated by different plant functional types. The authors find the associations between traits to be largely consistent across different organizational scales, and robust to the mode of data imputation and grain size used to aggregate community data. They report moderate but significant associations between axes of trait variation and climate predictors, especially temperature. Moreover, they find both different plant functional types and communities dominated by different plant functional types to differ in the PCA scores. Finally, they map the community PCA scores to show regional patterns in the prevalence of different prevailing functional strategies. Especially the latter may be useful to calibrate and improve large-scale dynamic vegetation models.

In general, the manuscript is well thought-through and pleasant to read, however there are certain conceptual issues that urgently have to be addressed before it can be considered for publication, most importantly with respect to the concept of ecological strategies. For this reason, I am afraid I have to reject the paper in its current state. However, I strongly encourage it for resubmission after addressing the concerns that I will list in the following.

Specific comments

Major comments

The most important issue I had when reading the paper was its somewhat loose use of ecological concepts and theory, specifically regarding the concept of functional/ecological strategies. While the term "strateg*" occurs a total 119 times in the manuscript, it is not defined anywhere in the text. This is problematic, because first there is more than one definition of strategies, and second because I am not sure that the authors actually use it in a correct way. A central tenet of the present paper is that

the ecological strategies of the plants present in a plant community can be scaled up to a “community-level strategy”. This idea seems dangerous to me for reasons I will detail in the following. To do so, it is important to set straight what I mean when I talk about strategies. In the broadest sense, ecological strategies can be defined as “how a species sustains a population” (Westoby et al., 1998). This definition is prominently cited in the first paragraph of Reich et al. (2014), an excellent article which showcases the importance of being clear about concepts. When talking about functional strategies, it is moreover important to be clear about the definition of functional traits. Here, I follow the definition of Violle et al. (2007), who understand functional traits as “morpho-physiophenological traits which impact fitness indirectly via their effects on growth, reproduction and survival, the three components of individual performance”. Due to this link to individual performance, functional traits are evolutionarily selected for. Hereby, certain combinations of traits are favored due to environmental pressures and a number of trade-offs reflecting allocation constraints. In the context of functional traits, a strategy can be viewed as this whole “set of traits” / “collection[...] of traits in syndromes” (Reich et al., 2003) that is jointly selected for. Evolutionary selection acts on the individual level (or more precisely, on genes that are manifested in individuals), and ultimately drives the differentiation of species as separate entities with different sets of functional traits. The sets of average functional traits of different species therefore reflect different functional strategies that were favored by selection; thus, it makes sense to look at strategies at the species level. However, there is no evolutionary selection acting at the community level. While working on my review, I spoke about the issue to three different ecologists with experience in functional trait theory – all agreed that the concept of “community-level strategies” is fundamentally flawed as it implies the presence of group selection. The same logic holds true for the concept of “trade-offs”, which are associated with evolutionary constraints acting at the individual level.

My most central recommendation to the authors would therefore be to work through the entire manuscript and replace all mentions of “community-level strategies” and “community-level trade-offs” with a more careful wording that is not at odds with basic biological theory. This should ideally be combined with clear definitions of the most central concepts in the introduction. Patterns in community-weighted means (or community functional parameters sensu Violle et al. 2007) are often thought to reflect trait combinations that are advantageous at a given site and hence a “locally ‘optimal’ trait strategy” (though this does not always have to be the case, see Muscarella & Uriarte, 2016). Already using the term ‘locally optimal strategies’ instead of ‘community-level strategies’ would resolve many of the conceptual issues, because it reflects that a community does not have a strategy, but the individuals that belong to it (and some of these strategies are more optimal than others). Functional strategies advantageous under the local conditions (i.e. trait combinations that confer an evolutionary advantage to the individuals that possess them) are likely to be more prevalent among the individuals in a community, which will be reflected in the patterns in community weighted means. For those reasons, the trait associations on community level will likely be similar to the associations on individual or species level. However, these patterns plainly emerge from the prevailing strategies among the individuals in a community, and are not themselves subject to evolutionary trade-offs.

Muscarella, R. & Uriarte, M. Do community-weighted mean functional traits reflect optimal strategies? *Proceedings of the Royal Society B: Biological Sciences* 283, 20152434 (2016).

Reich, P. B. et al. The Evolution of Plant Functional Variation: Traits, Spectra, and Strategies. *International Journal of Plant Sciences* 164, S143–S164 (2003).

Reich, P. B. The world-wide ‘fast–slow’ plant economics spectrum: a traits manifesto. *Journal of Ecology* 102, 275–301 (2014).

Violle, C. et al. Let the concept of trait be functional! *Oikos* 116, 882–892 (2007).

Westoby, M. A leaf-height-seed (LHS) plant ecology strategy scheme. *Plant and Soil* 199, 213–227 (1998).

Minor comments

Abstract:

L75: "...while those associated with water storage-isohydricity loaded along a second." – I would be careful with the wording here. There are neither data on capacitance nor any metrics of isohydry or stomatal control in the paper, so referring to water storage and isohydry in the abstract seems misleading.

Introduction:

L132ff: "How the range of community-level functional strategy changes as a function of water stress is not well established." – I would recommend to reword this sentence a bit. How would you define (and quantify) the "range of ... [a] functional strategy"?

L135f: "to underlie quantitative predictions" – this sounds weird – do you mean something like "to lend mechanistic support to quantitative predictions"?

L137f: "but require clear evidence on the distribution of forest strategies" – do you mean "require precise data"? Because you need data to parameterize a model, evidence seems a slightly incorrect term here.

L138ff: "An exploration ... would provide ..., as well as opening up ..." – inconsistent verb tense – please check grammar!

L143f: "regions of the USA (United States of America) and Europe (Spain, France, Germany, Poland, Czech Republic and Sweden)" – this apparent parallelism is confusing: the first set of parentheses defines the acronym, the second one lists the regions. Maybe get rid of these parentheses entirely?

L147: "We aggregated these forest plots into a grid level" – not sure where the level comes from and what you mean. "into a grid?"

L152f: "We expected to see strong trade-offs related to the structural strategy" – do you really expect trade-offs on community level, or do you expect that existing trade-offs acting on the individual/species level are reflected in community level data? See major comments!

L159: "community-level strategies" – this concept urgently has to be defined (see major comments!). I did a brief google scholar search and the only hit I got for the terms ["community-level strategies" "functional traits"] that was relevant was a single pre-print:

Neyret, M. et al. A fast-slow trait continuum at the level of entire communities. 2023.07.12.548516
Preprint at <https://doi.org/10.1101/2023.07.12.548516> (2023).

Methods and Material:

[[--I will address the issues in the Methods and Materials before I move on to the Results and Discussion, because I think that this is the way people should read papers, and that Nature should get rid of the methods-last style as it encourages uncritical reading--]]

L373f: "We selected the living trees in each plot and included only those with a diameter at breast height larger than 10 cm (12.7 cm in USA, 12 cm for Czech Republic)." -- I take that this difference is due to different census methods (and the use of "freedom units" in the US) but wouldn't it make more sense to filter out data of all trees below 12.7 cm in the European plots as well? This may decrease the total sample size, but I imagine it is worth it too avoid age-driven differences between regions.

L377ff: "We selected all the woody plants (trees and shrubs) ... based on the information from TRY ... and the Woody Plant Database..." – I am not sure if I understand this correctly. Based on information about what? Information about woodiness? Or presence of information in the databases?

L378: "community" – "community"

L381f: "In total, 749 woody species were retained for the analysis." – it would be interesting to know how many species were removed, why they were removed, and if the fraction of removed species/trees differed between ecosystems.

L382ff: "To further distinguish the different strategies of woody plants, broadleaves were split into two groups according to their leaf phenological types (evergreen and deciduous). We thus classified..." – were there no deciduous conifers like *Larix* spp. in the dataset? Sounds unlikely given the studied range.

L392f: "turgor losing point" – are you sure that was the term you searched? I just checked and found not a single hit for "turgor losing point vs. ca 3670 for "turgor loss point". Equally I only got 35 hits for "embolism vulnerability curve". Maybe you did not use quotes? Anyway, to me this seems uncommon terminology for the field.

L400f: "While the missing gaps were especially high (70%-90%) for these hydraulic traits." – While is a conjunction that introduces a subordinate clause. Normally, such a clause cannot stand on its own. Was this clause supposed to be connected to the previous sentence?

L401f: "For the species-level functional traits that were missing, we used the median trait values of the same genus instead. And if the trait was still missing, we used the median values of the family to fill the gaps⁵⁵." – am I understanding it correctly – the mechanistically most proximate traits for plant drought responses are based on imputation for 70-90 % of the species?

L409: "...using the formula described by 57" – this looks weird with the Nature citation style (an in-text reference would be needed here in my opinion). By the way, the equation is really just a weighted average so I am not sure if this merits a reference (especially since in that paper it is weighted by biomass, not basal area).

L409ff: "The functional traits, except wood density, were log-transformed to increase the symmetry of their distribution." – before or after calculating the weighted mean (hint: before makes more sense).

L432f: "According to the geographic information (e.g., coordinates) of each community, we extracted the climate factors to study the patterns of functional strategies across the regions." – the environmental data are on a different spatial scale than the gridded community data. One of the $0.25^\circ \times 0.25^\circ$ community grid cells (cf L375) covers about 400 of the $1 \times 1 \text{ km}^2$ climate grid cells (at a latitude of 44°). It makes sense to explain how this was accounted for when extracting the climate data for the communities. Based on a cursory reading of Moudrý et al (2023) I guess your best bet may be to extract climate data from the unaggregated locations, then aggregate, but I guess it is worth thinking about it in more detail.

Moudrý, V. et al. Scale mismatches between predictor and response variables in species distribution modelling: A review of practices for appropriate grain selection. *Progress in Physical Geography: Earth and Environment* 47, 467–482 (2023).

L435ff: "Principal component analysis (PCA)" – I am missing information about transformation, scaling/centering of the raw data before calculating the PCA. You mention above that all traits besides wood density were log-transformed, but did you also scale them to unit variance? If not, the standard deviation of each variable will influence their scores – if so, it is not easily possible to compare scores

between Fig 2 and Fig S2. `factominer::pca()` scales to unit variance by default, so this is critical information.

Moreover, I believe it may be useful to explain what was the rationale to choose the specific set of traits you included in your PCA, because PCAs are extremely sensitive to the choice of included variables. If you want some variables to turn out on the first "most important" axis, just make sure that you put in a bunch of closely correlated variables, and you are set. High R^2 , and just the results you want. In this case, high leaf N, Ks and LS are all associated with more acquisitive strategies, while LMA is associated with more conservative strategies, so it is not surprising that all these score high on the first axis. To make such an analysis credible, it makes sense to clearly clarify whether this set of variables was selected a priori and not modified after seeing the data, and why each variable was included. For instance, there are only few papers in the vulnerability curve literature that identify systematic patterns in VC slope, so I found it interesting that this was included instead of e.g. hydraulic capacitance or `gmin`, which are mechanistically likely much more relevant (some rascals might even think that maybe not including Slope would have messed up the nice first LES axis? 😊).

L454ff: ""Lindeman, Merenda, and Gold," which is a commonly suggested approach for determining relative importance 65. " – reference 65 is the paper by Lindeman, Merenda and Gold, so I doubt that this is unbiased evidence for the claim that the approach by Lindeman, Merenda and Gold is a good one...

L470: "ggplot2 R package (version 3.4.3)." – if you explicitly mention package and version, I would also add a citation – if anyone deserves it, then these authors.

Results and Discussion:

L202ff: "Contrary to the general expectation that more robust xylem was linked to higher wood density³⁶, the slope had extremely weak correlation (Pearson correlation coefficient = -0.04) with wood density..." – the slope of a vulnerability curve itself does not contain information about embolism resistance, just about the change in conductance per unit change in water potential (you can have curves with drastically different P50 and the same slope). Therefore, I would not believe a priori that the VC slope should be related to wood density at all, and would require a strong theoretical justification as to why to assume this (the same holds for the slope ~ turgor loss point link). By the way, if you were referring to the slope – WD relationship in Fig. 6b of reference [36] (Hoffmann et al., 2011): this plot shows the slope of a stomatal response curve, not a vulnerability curve.

L205f: "This result suggests that water storage choices are more complex than a simple function of either structural acquisitiveness or stomatal strategy" – why? There are no data about water storage. Wood density is indirectly linked to water storage capacitance due to a trade-off driven by spatial constraints, but I would be careful to interpret it as exchangeable with water storage.

L213ff: "There was, however, 11.8% ... more explained variance in the community-level when compared to species-level, indicating some simplification of the strategy space when scaling up." – I am not sure that many readers will understand the biological interpretation of "simplification of the strategy space". Is there a less convoluted way to say this? Could this have to do with averaging out a sizeable fraction of the variance?

L218f: "This may reflect optimization of resource usage (e.g., competition for light and forest growth) within forest communities" – are you sure? Selection does not act on community level.

L219f: "Additionally, the full range of PC1 score was wider at community-level than that of species-level" – see comment to L435ff: if the standard arguments of `factominer::pca()` were used (i.e. "scale.unit = TRUE"), I would be very careful when comparing axis scores and linking them to "larger trait variation" – the variances in both PCAs are associated with different ranges in traits!

L228f: "...illustrates the importance of taking account of the full range of hydraulic strategies..." – would you really say you did that if 70-90% of the hydraulic trait values are imputed?

L269ff: "There are, however, large areas in eastern USA and Central Europe that do not tend towards either strongly anisohydric or isohydric, indicating that tree species are ordered on a continuum rather than a dichotomy⁴⁶." – yep and also because more and less isohydric and anisohydric species coexist in the same ecosystems – as well as acquisitive and conservative species. Aggregating on the community level discards information about this very obvious fact.

L284: "The link of PC1 with mean summer temperature was not reported before" – if no-one else did a PCA with the exact same variables, it is not surprising that it was not reported before. While it seems tempting to see PC1 as a metric of the spectrum of conservative to acquisitive strategies, in fact a PCA axis is a linear combination of all variables in the analysis (literally just $X_{i1} \cdot \lambda_1$ <corresponding eigenvector>), so comparing the PCA results from a PCA with LES traits and hydraulic traits to PCA results from a PCA using for example the classical LES traits (LMA, Amass, N, P, Rmass and LL, as in Wright et al., 2004) is pretty difficult.

L305ff: "Higher spatial resolution work will be necessary to identify and distinguish which factor causes the one water stress strategy to prevail over another." – while I agree that high-resolution data never hurt, I believe that in this case a better coverage of species with actual measurements of hydraulic traits would be much more beneficial.

L317ff: "Moreover, identifying the main functional traits in response to water stress is crucial, which may lead to apparent discrepancies in exploration of trait trade-offs and main functional strategies⁷." – This sentence seems off – why would identifying the main functional traits lead to discrepancies?

L321: "availability⁵¹ The strategies" – a full stop is missing before "The".

L329: "regular cutting might be expected" – I believe a comma is missing before "might".

L345: "surprisingly the relationships with climate are relatively weak" – is this really so surprising? To me, it seems in line with general patterns in trait-environment relationships (for instance see Anderegg 2023).

Anderegg, L. D. L. Why can't we predict traits from the environment? *New Phytologist* 237, 1998–2004 (2023).

Tables:

Table 1:

1. I would reformat the units in the second column using negative exponents rather than fraction lines (division is left-associative unless stated otherwise, so technically it is correct, but almost all formatting guidelines ask for parentheses to avoid ambiguities in such cases).
2. For K_s , MPa got auto-corrected to Mpa – please fix.
3. "Leaf mass area" – some people seem to call LMA this way but I find "leaf mass per area" much more logical because "leaf mass area" leaves the impression that the area ends up in the
4. numerator.
5. "Slope for the embolism vulnerability curve between P50-P88" – this is a very niche definition for a vulnerability curve slope – normally, the used models estimate the local slope (i.e. first derivative) of the curve at one of the critical water potentials (mostly commonly P50). Is there a rationale behind using such an uncommon definition? Doesn't this complicate comparison with other studies? Btw. if you really calculate slope like this (and also in the other case) the unit of the slope is $\% \text{ MPa}^{-1}$ (assuming the curve to be described in terms of PLC).
6. ". A less negative TLP indicates more isohydric." – "more isohydric behaviour"?

Table S2: check for consistent capitalization (see e.g. No. 5 blackman et al.).

Figures: Figure 1 and Figure 2 differ in naming from Fig. 3 and Fig. 4 – please be consistent.

Figure 2: This figure seems a bit stretched horizontally – does it have 1:1 aspect ratio or did it get messed up in Word? (The same is true for Fig. S6, but not so much for S2).

Figure 4: I feel that you would get a better linearity in Fig. 4 b and e if log-transforming the aridity index – this transformation would be well justified as the AI is a ratio, which is more easily interpretable on a logarithmic scale. In either case it makes sense to reflect about the biological significance of a statistically significant relationship that only explains 1% of the variance in a PCA axis.

Figure S7: This figure seems stretched vertically – or is this a weird projection? In either case the maps in the main text look better.

Dear reviewers,

Thank you very much for giving us the opportunity to revise our manuscript entitled “Mapping multi-dimensional variability in water stress strategies across temperate forests” for Nature Communications. We are grateful for your suggestions and comments, which helped us to significantly improve our manuscript. We have now fully addressed these questions. The point-by-point responses are below.

Best wishes,

Daijun Liu on behalf of all authors.

#####

Response to the reviewers’ comment

REVIEWER COMMENTS

Reviewer #1 (Remarks to the Author):

Review of the manuscript entitled Mapping multi-dimensional variability in water stress strategies across temperate forests by Liu et al.

Overall statement and summary: Overall I enjoyed reading the manuscript. The authors discuss a highly relevant problem of forests’ reactions to water stress in times of its increasing frequency and severity. Their approach is to use in-situ measured functional traits. I would consider this approach to be appropriate, since plant traits are known to reflect their resilience to water stress. I found targeting a specific type of stress of particularly high utility. This allows for a more specific interpretability.

More so, the authors address a central problem of trait-environment interaction, which is the scale mismatch of in-situ measured plant traits and the ecosystem or forest scale at which ecosystem functions are usually measured. They do so by attributing eight water stress-related traits on a tree scale to those species growing in various forests in the USA and Europe for having forest-wise plant strategies of water stress. Subsequently these strategies were related to climate factors.

Specifically, the authors show:

1. Woody plants water stress traits vary in two dimensions that explain 64% of their variation. They vary along an axis onto which leaf nitrogen, leaf area to sapwood area ratio (LS), Maximum xylem conductivity per unit sapwood area (Ks), leaf mass per area (LMA) and xylem water potential at 50% loss of conductivity (-p50); and onto a second axis representing mainly wood density (WD), how abruptly embolisms occurs (Slope), as well as leaf turgor loss point (TLP), reflecting leaf water regulation, i.e. stomatal closure. The authors interpret the first axis as reflecting how acquisitive or conservative plants use water. The second axis is condensed to isohydrocity.
2. These axes are consistent independently of the aggregation scale: may it be species level or forest community.
3. Forest communities’ water stress traits are distributed unevenly in space, but their patterns cannot be well explained by climatic water stress factors.
4. Mean summer temperature correlates slightly with the acquisitiveness, while the mean annual temperature correlates weakly with the isohydricity.

Overall, the findings, methods, and language fit the scope of Nature Communications.

I found particularly powerful to add a abundance measure to the in-situ measured traits by weighing by woody plant diameter. The scale of study is with 0.25° small thus climatic heterogeneity within these grids may be considered sufficiently small for merging trait data. The data basis per grid appears to be sufficiently large for containing a representative trait and species subset, and the availability of indirect abundance metrics (diameter) allows for further ecological information. Further the range of two continents allows inferences for large climatic gradients with high resolution and allows interpretable results.

Thank you very much for your very positive summary.

One can compare not only the climatic variability, but also make assumptions about the taxonomic diversity in the two distinct land masses.

Thank you very much for your suggestion. We agree that functional trait variation could potentially be associated with the taxonomic diversity. However, our study focuses on the water stress traits and strategies, and exploring the relationships between traits and climate variability for which we have direct hypotheses. We accept your suggestion and have added some discussions in Result and Discussion section. Please see the lines 316-321.

“However, the relationships between PC scores and climate factors were also relatively weak, aligning with the patterns observed in global trait-climate relationships⁴⁸. These weak relationships with climate variables suggest that the dominant water stress strategies which emerge may result from functional adaptations to other aspects of environmental conditions. One such candidate variable would be nutrient availability, as recently reported for the Amazon forest⁴⁹.”

Having said this, I think this article could profit from a number of changes.

Major comments

[1] Clarity. Although this is a well written manuscript, I recommend a couple of changes.

1.) The whole paragraph in lines 125 – 141, can and should be communicated better. The language as well as the logic should be reconsidered. I gave some suggestions in my comments below.

We thank you for your comment. We have accepted your suggestions and revised them (Please see lines 131-148). Now they read as:

“To move from tree-level responses to forest community-level responses it is necessary to characterise how water stress strategies assemble into communities. When functional traits at the species-level are assembled into communities, the strategy that emerges as locally dominant depends on the balance between niche complementarity and resource competition^{23–25}. As water resources become less limiting, competition may tend to favour more acquisitive strategies, resulting in a convergence towards a limited range of water-demanding strategies^{16,19,26}. In water-limited environments, a wide range of water stress strategies can develop, and they are partitioned environmentally according to particular combinations of conditions that favour certain strategies^{27,28}. How the locally dominant strategy varies as a function of water stress is not well established. In order to study the range of locally dominant strategy changes with water stress, it is fundamental to understand the likely response of the forest community to ongoing environmental changes, both in a qualitative sense and to underlie quantitative predictions by vegetation models. These models are currently taking big steps forward in their capacity to simulate physiological processes relevant to water stress

responses^{29–32}, but require clear evidence on the distribution of forest water stress strategies. An exploration of dominant water stress strategy of forest communities along climatic gradients would promise a new perspective on forest functional biogeography and open up opportunities to support large-scale projections of forest functions using vegetation models.”

2.) Language. In parts I was missing clarity, I made suggestions where applicable. Particularly, I think you can make the method more accessible by sticking to either the term “forest” or “community” of this aggregation level. Since you are working with different aggregation levels, make it easier for the reader to understand which aggregation you are talking about. For instance, in Fig 2 you talk about “forests across regions” (i.e. species aggregated to cells, to forests), Forests = communities (?). Please use one term only. Fig.3 “forest communities”. Fig 4 “forest”.

We have changed it to “forest communities” as you suggested.

3.) ll. 152 and 155. I think you could fuse the expectations as well as the hypothesis, as these should, if I am not mistaken, be the same thing.

We have deleted the description on “the expectation” (lines 162-166). Now it reads as:
“We assessed: (a) the axes of dominant strategic variation with respect to water stress traits that can be identified at the community-level based on principal component analysis (PCA), (b) whether these differed from axes of variation at the species level and (c) whether gradients in these strategies differ across the geographic and climatic space of temperate forest communities (Fig. 1c, d).”

ll. 154. Add for comprehension somewhere before the hypothesis that you also relate these data to environmental data (“and (b) gradients in these strategies across the geographic and climatic space of temperate forests can be identified (Fig. 1c, d)»)

We have added a sentence on the climate information of forest communities (lines 158-162).
“We thus obtained 12,446 forest communities across the USA and Europe, for which we also extracted climate variables relating to water availability and temperature information based on geographic locations of forest communities (details of the variables in Methods and Materials).”

4.) Please add at an earlier time than now that species included are woody species. To me it was not clear for a long time what data sit behind the species level. Make this clear earlier and particularly in the figure captions.

We have added a mention of this to the abstract (line 71).

[2] Two additional analyses that are simple to implement could shed a light onto discussed aspects.

1.) The authors describe the similarity of the PCA based on species and forest community scale. (in line 210: “The community-level PCA displayed similar loadings of traits along two main dimensions of the PCA to that at species-level (Fig. 2a; Fig. S2a)».) The within vs. across community variation of plant traits could give further insights into how similar the species’ traits are within one forest community vs. across. For example, you could calculate the coefficient of variation per trait across the communities. For example, Joswig et

al. 2022, found a relationship of variance explained by environment and coefficient of variation (Figure 42, Supplementary Information).

We thank you for this suggestion. We have used the Procrustes tests to check the similarity of PCA results for different trait gap-filling methods and grid sizes. Please see the details below.

As you suggested, we have conducted additional analysis to understand how the traits vary within and across forest communities. We compared the mean variance of each trait within forest communities and the variance of CWM-trait across forest communities. We can see that the mean variance of P50 and slope within a community is larger than the variance across all communities (see the plot below). We have briefly commented it in the Result and Discussion section (lines 342 -344).

“The same applies to niches within communities; we found a similar or higher variance of individual traits within communities as opposed to across them (Fig. S11).”

The authors also describe the weak PC1 and PC2 relationships with climate. This could be due to a high spatial heterogeneity (e.g. geodiversity). This could be indicated with high internal variability. You point to this in ll. 353: “But surprisingly the relationships with climate are relatively weak and the proliferation of different strategies in more arid environments and the limited explanatory power of climate variables poses the question of what factors exert the primary filter governing which water stress strategy comes to dominate a forest community.» ll. 190.

We agree that this could indeed be the case, which is why we highlighted it in the text. The position data of the individual NFI plots that we are working with are fuzzed, such that the highest resolution climate data that we can associate plots with is 2.5 arc minutes. We have extracted the climate factors of each plot and then averaged the values of each plot to the grid level ($0.25^\circ \times 0.25^\circ$).

We thus added a description in the Result and Discussion section (line 316-321). It reads as:

“However, the relationships between PC scores and climate factors were also relatively weak, aligning with the patterns observed in global trait-climate relationship⁴⁸. These weak relationships with climate variables suggest that the dominant water stress strategies which emerge may result from functional adaptations to other aspects of environmental conditions. One such candidate variable would be nutrient availability, as recently reported for the Amazon forest⁴⁹.”

The point about the importance of high-resolution information to unpick this is also highlighted (lines 335-338):

“Local conditions of temperature, elevation and soil nutrient availability have been previously shown to explain the community-weighted variance of functional traits^{50,51} and trade-offs related to these may be playing into the dominant strategy emerging at community level¹⁸.”

2.) The attribution of trait to the first and second principle component is based on loadings. Other studies attribute some traits you use to other syndromes/axes. For example leaf N and

LMA were attributed to leaf economics, e.g. by Wright et al. 2004 (Nature) , Díaz et al. 2015 (Nature). I think the article would benefit from discussing these “double” attributions.

We do not consider these to be so much “double attributions” as reflective of that a plant’s functional strategy has to constitute a coherent whole across all its organs. We have added the following interpretative discussion on lines 199-204.

“This consistency reflects a whole-plant functional coherence whereby a plant with, for example, high leaf area to sapwood area ratio (which can support greater relative height growth to win the race for light or more investment in canopy to capture light - both resource acquisitive strategies) will tend to have higher conductivity sapwood to supply those leaves with water, whilst the leaves themselves will tend to be relatively thin, reflecting the inherent resource trade-off involved in prioritising height growth or photosynthetic area.”

We have also added the following to the Future Challenges section (Please see lines 354-362).

“Our selection of traits, being driven by considerations of modelling tree form and function from first principles, did not include those that clearly map onto the height dimension that has been identified by other large-scale trait analyses^{15,54}. However, the height dimension is one that in principle should emerge from application of traits related to resource acquisition, resource allocation and tissue turnover and mortality (Table 1) within a process-based vegetation model. Exploration of the structural and behavioural outcomes of functional strategies by coupling large-scale trait analyses with suitable process-based models⁵⁵⁻⁵⁷, could facilitate a more complete picture of plant form and function.”

This brings me to another point, which are quantifiable trait attributions. I thus suggest adding an analysis, which allows a quantified attribution of these traits to trait groups. I presume the results will be the same as you estimated visually. For example, Joswig et al. 2022 (NEE) show a hierarchical clustering analysis of all trait groups, based on (absolute) Pearson correlation coefficients of trait-trait relationships (figure 1a) that is also adopted in Maynard et al. 2022, (Nature; figure 4a). If you wanted to include this trait group attribution, you could simply transform figure 2c for this purpose (but I would not ask you to do so).

We very much appreciate your suggestion on adding the trait group attribution, which we agree is a valuable technique. However, our line of argumentation is not based on trait groupings, but rather their association with particular axes of variation. A clustering based on Pearson’s correlation coefficients integrates over all axes of variation and not just the dominant ones. This would not be conceptually consistent with our approach. For instance, loadings along PC1 indicate that Leaf N and sapwood conductivity both vary strongly along this axis. But that does not imply that the residual variation in Leaf N and sapwood conductivity after accounting for variation associated with PC1 need be associated with the same traits. It could well be, for example, that LMA is more closely related to the leaf N residuals and wood density more closely related to the stem conductivity residuals. In a case like this, the clustering approach might put Leaf N and sapwood conductivity in separate groups, despite the fact that from the perspective of the main strategic variation, they are both strongly associated with the same dimension. On this basis, we conclude that including the clustering approach of Joswig et al. 2022 would introduce a conceptual inconsistency and potentially require detailed discussion which would dilute the line of argumentation in the manuscript.

[3] Implications

1.) I think the authors could have stressed more the following interpretation, suggesting (ll. 352.) “that community-level water stress strategies are more closely related to water demand than water availability». Possibly this could be added to the abstract.

We have added this sentence into the abstract (lines 76-78). It reads as:

“Surprisingly, spatial patterns of local water stress strategies were better explained by temperature than by aridity, suggesting a temperature-driven adaptation.”

2.) In the discussion, I was missing the picture put together again. Could you «walk» through the different environments and describe the forest communities’ adaptations?

We think that the suggestion is to restructure the Results and Discussion section by the different environments. We have instead chosen to structure the section according to the main aims set out at the end of the introduction and we prefer to retain this structure as we think it keeps a clearer thread throughout the manuscript. However, following your suggestion, the section on “Geographic variation of functional strategies” does go through the spatial variation in the strategies.

3.) To which extent do you think your findings from single woody plants are representative of forest communities? You could discuss possible conflicts and open up important discussions on this matter (non-woody species, temporal variability). Which other traits would be candidates reflecting plant response well, I am sure the authors can do a great job at imagining these (e.g. root to shoot ratio?)

We are not sure what is referred to with “single woody plants”, as each of our data points is based on a forest community. However, we agree that it is important to bring up other important traits which need to be investigated in this context.

We have added some discussion in the main text in the “Future challenges” section (lines 366-372 and lines 376-380).

“Here we have examined the strategies of woody plants, however, herbaceous species were not taken into account. Additionally, we excluded some other traits that we anticipated to be important for water stress strategies, such as rooting depth ⁵⁹, root-to-shoot ratio ⁶⁰ and minimum leaf conductance (gmin) for the rate of plant desiccation ⁶¹. The difficulty of collecting these traits is reflected in the low sample size available in the literature. While wood density is widely studied, it is only a weak indicator of potential capacitance ³⁹, because it is still not known how much stored water is accessible.”

“Furthermore, scaling up individual or species-level traits to the community level may only capture the dominant strategies. The intraspecific trait variation in response to environmental conditions was not included based on the method of community-weighted mean traits. New approaches (e.g., normalising plant traits per unit land area) to integrate community traits and their contextual information are necessary ⁴².”

Further comments

Abstract

ll. 72: “climate drivers” because of complex biosphere – environment interactions, I would suggest calling these climate “factors”.

Changed “climate drivers” to “climate factors” in the whole manuscript.

ll. 72: I had difficulties understanding the following sentence. You might want to consider simplifying, else this finding could possibly be put after the description of the trait dimensions, so it might be easier to follow the train of thoughts. “We find a high degree of consistency between community-level water stress strategies and those at species-level». I am sure the authors will find a good way. My idea that I do not ask you to use would be: We find water stress strategies to be consistent across the scales of communities and species.

We have revised the description to be clearer (Lines 72-74). It reads as:

“We found that the range of water stress strategies which dominated at community-level were consistent with those available at species-level.”

ll. 75: The term “isohydric» may not be familiar to all readers of Nature Communications. You might want to replace, define it briefly or simply add a simpler term in brackets (e.g. water regulation).

Added.

ll. 77: To my taste the conclusions could be condensed. A suggestion, which I do not expect you to adapt would be: “These findings have the potential to improve predictions of ecosystem models, particularly for patterns of tree mortality and forest biomass accumulation in a changing climate.»

We have changed it as (lines 77-80):

“These findings provide a basis to build predictions of forest response under water stress, with particular potential to improve simulations of tree mortality and forest biomass accumulation in a changing climate.”

Introduction

ll. 111: I do not understand the logic in this sentence, and I strongly suggest reformulating it, since one can understand the opposite: “For example, a tree may be structurally acquisitive, investing in a limited water transport capacity for its leaf area, allowing it to instead invest resources in height growth to win the competition for light.” Accordingly, an acquisitive, thus “risk-taking” tree could be one of thin or few water vessels, transporting little water to the leaves, which are consequently of smaller size. This would entail, less photosynthetic activity and thus fewer resources that can be invested. Acquisitive strategies should be the ones with large water transport capacity to match the need for water for its large leaf area.

Why not write something along the lines of: “For example, a tree may be structurally acquisitive, investing in a *large* water transport capacity for its leaf area, allowing it to invest *more* resources in height growth to win the competition for light.”

The basis of our argumentation is that resources invested in larger sapwood area are resources that cannot be invested in height growth, reducing the competitiveness of the tree in the race for light. However, we agree that this argument was not well formulated as our sentence

conflated sapwood area with sapwood conductivity (a separate trait which we examine). It could indeed be the case that there is relatively little investment in sapwood area, but the sapwood that exists is more conductive. We have reformulated as follows (please see the lines 114-121).

“For example, a tree may prioritise light acquisition over being robust to water stress, investing relatively more in height growth than in the area of conductive sapwood for a given leaf area. Assuming the same sapwood conductivity and robustness of xylem to cavitation, during a drought, this high leaf area to sapwood ratio may lead to excessively negative pressures in the xylem, causing hydraulic failure and death, whereas species favouring a lower ratio of leaf to sapwood area may survive (Fig. 1b). The combination of structural traits influences whether a tree is structurally more acquisitive or conservative.”

I would further add the mismatch with past studies’ trait attribution. Traits you put into the acquisitive-conservative axis - often termed plant economics - were attributed to the “size” axis according to Díaz et al. 2015, Joswig et al. 2022. The strong relationships of water transport traits, i.e. leaf area as well as water transport capacity (e.g. conduit diameter) were explained to hydraulic scaling. This should be addressed somewhere

We greatly thank you for this suggestion by adding the previous studies related to “plant size”. The trait such as leaf area to sapwood ratio (LS) is highly related to plant size (Liu et al., 2019). High LS points towards quicker growth and thus potentially larger size. Since our study is mainly focusing on the main trade-off between acquisitive-conservative strategies (e.g., Fig. 2a), we do not include the plant size trait, i.e., height into our analysis. But we have added the discussion in Future challenges (please see lines 354-362).

“Our selection of traits, being driven by considerations of modelling tree form and function from first principles, did not include those that clearly map onto the height dimension that has been identified by other large-scale trait analyses^{15,54}. However, the height dimension is one that in principle should emerge from application of traits related to resource acquisition, resource allocation and tissue turnover and mortality (Table 1) within a process-based vegetation model. Exploration of the structural and behavioural outcomes of functional strategies by coupling large-scale trait analyses with suitable process-based models⁵⁵⁻⁵⁷, could facilitate a more complete picture of plant form and function.”

Il. 116. Add simpler term of isohydricity. «Other adaptations to water stress (e.g., isohydricity*, or water regulation*)»

Added.

Il. 126. For better understandability, you might want to consider rephrasing: When» *[object] are* assembled into communities, the*ir* dominant strategy depends on the balance between niche complementarity and resource competition»

We have revised this sentence (lines 132-134). It reads as:

“When functional traits at the species-level are assembled into communities, the dominant strategy depends on the balance between niche complementarity and resource competition²³⁻²⁵.”

ll. 128. Language & logic. This sentence appears to me flipped. Wouldn't competition be one interpreted process from the evidence of domineering acquisitive strategies? "Competition between species suggests that as water resources become less limiting, more acquisitive strategies will be favoured, resulting in a convergence towards a limited range of water-demanding strategies^{15,16,24}."»
Something along the lines of: "Dominant acquisitive strategies can be evidence for competition among species when water is not a limiting factor, resulting in a convergence towards a limited range of water-demanding strategies^{15,16,24}."»

We have changed the description (lines 134-136). Now it reads as:

"As water resources become less limiting, competition may tend to favour more acquisitive strategies, resulting in a convergence towards a limited range of water-demanding strategies^{15,16,26}."

ll. I would suggest a different phrasing, since this evokes the idea of temporal change. "In water limited environments, a wide range of* water stress strategies, and they are partitioned environmentally according to particular combinations of conditions that favour certain strategies^{25,26}"«

We have changed the description as suggested (lines 136-138). Now it reads as:

"In water-limited environments, a wide range of water stress strategies can develop partitioned environmentally according to particular combinations of conditions that favour certain strategies^{7,27,28}."

ll. 132 I think this has been said earlier, and you could leave this away: How the range of community-level functional strategy changes as a function of water stress is not well established. You could integrate into the next sentence: "In order to study how the range of community level functional strategy changes with water stress, it is* fundamental the likely response of the forest to ongoing environmental changes, both in a qualitative sense and to underlie quantitative predictions by vegetation models.»

We have changed it as suggested (lines 139-143):

"In order to study the range of locally dominant strategy changes with water stress, it is fundamental to understand the likely response of the forest community to ongoing environmental changes, both in a qualitative sense and to underlie quantitative predictions by vegetation models."

Ll. 136 Same here, I would suggest working on the language and logic. These models are increasingly technically capable to simulate water stress responses^{27–30}, but require clear evidence on the distribution of forest strategies.

We have revised this sentence (please see the lines 143-145):

"These models are currently taking big steps forward in their capacity to simulate physiological processes relevant to water stress responses^{29–32}, but require precise data on the distribution of forest water stress strategies."

ll. 138 . Also this sentence is too lengthy.

“ An exploration of *water-stress strategies of forests along climatic gradients promises* a new perspective on forest functional biogeography, as well as *it may bear implications for forest management* using vegetation models.”

We have revised it accordingly (lines 145-148). Now it reads as:

“An exploration of the dominant water stress strategy of forest communities along climatic gradients would promise a new perspective on forest functional biogeography and open up opportunities to support large-scale projections of forest functions using vegetation models.”

Ll. 146. Could go into the discussion:” , as the information on rooting traits is very limited»

We thank you for your suggestion. This sentence summarized why we select these traits based on Fig. 1 a, b. We did not include the rooting traits in our study due to limited data. We thus prefer to shortly mention it here. We have also given more details on the reasons why not include root traits into our analysis in the Result and Discussion (please see the lines 367-370).

“Additionally, we excluded some other traits that we anticipated to be important for water stress strategies, such as rooting depth⁵⁹, root-to-shoot ratio⁶⁰ and minimum leaf conductance (gmin) for the rate of plant desiccation⁶¹.”

ll. 147. Did you consider using a selection criterion for the number of species present?

We selected all species with full species name from the forest inventory data (see Methods and Materials). Our aim is to characterise the forest as it is, rather than subsample by number of species.

Figure 1d. The map seems to be distorted, at least it is not a projection that is common. You might want to change that.

We have modified the maps to make it nicer. Please see the figure 1.

How much overlap in terms of species do you have in the different forest stands?

In total we consider 628 woody species distributed across 12,446 forest communities. Out of the total 643 species, 81.1% are distributed among more than two forest communities.

Did the species used, belong to all growth forms?

Yes, all the species have been assigned to one of the growth forms.

ll. 185 You might want to mention further associations of traits, e.g. carbon and nutrient cycling/stock related to leaf N, wood density etc.

Here we focus on interpreting how the traits relate to each other. We have given associations of the traits in Table 1 and for brevity prefer not to duplicate in the main text.

Results and Discussion:

Il. 185: An introductory sentence e.g. relating the findings to the question might be useful. “Community-level strategies.»

We have changed the subtitle to, “Variations in strategy at the community level”, and broken up the questions at the end of the introduction to more clearly link questions to the results section.

Il. 185. Please be more specific and describe the differences (but this may be a matter of taste). “Clear differences in forest functional strategies were identified.»

We have revised this sentence (lines 188-189). It reads as:

“Clear differences in dominant functional strategies along the two main axes were identified.”

Ll. 199. I don’t think a trait can “drive” an axis. Maybe rephrase. “The axis was positively driven by these two traits, while it was negatively driven by a steeper slope of the embolism vulnerability curve (or high sensitivity of the xylem to water stress) (Fig 2c).»

We have rephrased this sentence (lines 224-227). Now it reads as:

“A more negative leaf turgor loss point and higher wood density contributed to the positive direction of this axis, while a steeper embolism vulnerability curve (indicating high sensitivity of the xylem to water stress) contributed to the negative direction (Fig. 2a).”

For the discussion: Which aspects of this analysis could have biased your research

We have gathered the potential uncertainties such as trait selection and trait variation in the “Future Challenges” section (please see the lines 366-372 and 376-380).

“Here we have examined the strategies of woody plants, however, herbaceous species were not taken into account. Additionally, we excluded some other traits that we anticipated to be important for water stress strategies, such as rooting depth ⁵⁹, root-to-shoot ratio ⁶⁰ and minimum leaf conductance (gmin) for the rate of plant desiccation ⁶¹. The difficulty of collecting these traits is reflected in the low sample size available in the literature. While wood density is widely studied, it is only a weak indicator of potential capacitance ³⁹, because it is still not known how much stored water is accessible.”

“Furthermore, scaling up individual or species-level traits to the community level may only capture the dominant strategies. The intraspecific trait variation in response to environmental conditions was not include based on the method of community-weighted mean traits. New approaches (e.g., normalising plant traits per unit land area) to integrate community traits and their contextual information are necessary ⁴².”

Il. 202. I would suggest being cautious about claiming “links” based on correlations. With the analysis (Pearson+hierarchical partitioning) you could refer to the correlation (see major comments above).

«Contrary to the general expectation that more robust xylem was linked to higher wood density³⁶, the slope had extremely weak correlation (Pearson correlation coefficient = -0.04)

with wood density and had also relatively low correlation (Pearson correlation coefficient = -0.33) with leaf turgor loss (Fig. 2c).»

We have changed it to “associated with”.

Figure 2. For an easier link between figure 2 and figure 3, you could try to make this graph a quadrat. The reason I suggest this are the identical PC1 and PC2 scores (from -6 to 6) in figure 3, in the original PCA in figure 2, however, PC1 seems to have a greater variation than PC2.

We have modified Fig 2 as you suggested. Please see the new Fig. 2 in the main text.

Caption: Please add the number of species/samples/forest sites to the figure. There should be an indication here or better before in the text about which type of species are integrated. E.g. which growth forms.

We have added the number of forest communities in the caption (please see the lines 210-214).

“The dominant forest type was identified according to the relative proportion of dominant types representing more than 50% of grid basal area: BD (broadleaved deciduous, blue, 5584 communities); BE (broadleaved evergreen, green, 407 communities) and NC (needle-leaved conifer, red, 6461 communities).”

Please add the long version of the trait abbreviations.

We have added the trait abbreviations in Figure caption (lines 215-219)

“Abbreviations of functional traits: N (leaf nitrogen content); Ks (maximum xylem conductivity per unit sapwood area); LS (leaf area to sapwood area ratio); LMA (leaf mass per area); P50 (xylem water potential at 50% loss of conductivity); Slope (slope for the embolism vulnerability curve between P50-P88); TLP (leaf turgor loss point) and WD (Wood density).”

Two suggestions to allow a quadrat shape of 2a:

(1) Including hierarchical partitioning: top row quadratic fig2a, next to 2b, bottom row hierarchically clustered (optional) 2c

(2) Alternatively, you could probably leave out 2b (put into the appendix) without much loss, since the loadings are displayed in 2a.

We have accepted the second suggestion. Now the new Fig 2 is shown in the main text.

ll.212 possible rephrase the hypothesis quickly, so the reader does not have to go back

This sentence was already a reminder of our hypothesis, but we missed out the word “that”, which made this unclear. We have remedied this (lines 241-244).

“This result is consistent with our first hypothesis (H1: the large strategic variation at species level will be reflected in the variation of dominant strategies at community level), a similar range of strategies as found at species level was present when considering the dominant strategy at community-level (Fig. S3 and S4).”

Figure S5. Size of grid cell – this allows a thorough methodology.

We have added more details on the methods used in the sensitivity test in Figure S8 (lines 136-139).

“The forest plots have been aggregated into the sizes of 0.1° and 0.5° separately. And then eight key functional traits of woody species at species-level have been assigned to the species in each community. The missing trait gaps have been filled by three methods.”

ll. 219 This is interesting; possibly I misunderstand this. Yet: how would it be possible for a community to have a larger PC1 value than an “extreme” PC1 species? Once aggregated, these communities, consisting of single species/plots should have consequently a smaller trait range. Do you think this might be because of the trait combinations projected? Maybe one of the other reviewers might give advice of the possibility to compare absolute numbers (PC loadings, range).

Yes, larger PC1 values are found at community-level than species-level. We calculated the loading values of eight functional traits along the main PC dimensions. Once the traits at species level assemble into community level, the PC1 explains more of the trait variance at the community level. Thus, larger loading values at community level have loaded in the first dimension of PCA compared to species level. Indeed, the ranges of eight functional traits at species level are larger than that of community level. We also presented the result for readers (Fig. S4).

My suggestions would be to look at the trait-wise ranges and compare species vs. community and check if this pattern reoccurring (but I would be surprised).

As we replied above, the PC1 explains more of the trait variance at community level than at species level even though the ranges of all eight functional traits at species level are larger than that of community level (Fig. S3).

You could exclude the possibility of this pattern being a necessary mathematical consequence of combining different species/plots with a randomization analysis (e.g. randomly attribute species/plots to communities then perform PCA at community scale).

We thank you for this suggestion. As we replied above, the larger PC1 ranges are attributed to the fact that PC1 explained more of the trait variance at community level than at species level. The species distribution is not random, so randomly combining the species into forest communities may lead to a misunderstanding of the PCA result.

Please add reference to the figures you are referring to.

«Additionally, the full range of PC1 score was wider at community-level* (Fig. 2a)* than that of species-level *(Fig. S2a)*, suggesting this spectrum at community-level has larger trait variation under environmental variability38–40.»

We have deleted this sentence since the range can be not used to compare the loading values. This issue does not exist anymore.

Il. 221 It is a nice result to see the different leaf types separated. Wouldn't it be enough in the future to reduce the trait information to leaf type only to assess the water stress strategy?

We agree that it is interesting that the leaf types are so clearly separated, however there remains a large amount of variation within leaf types, and therefore, we do not think that simplifying to leaf type would be an appropriate approach to assess water stress strategy. For instance, both needleleaved and deciduous broadleaved dominated communities span a wide range of positive and negative loadings of PC2.

How much do you additionally to the leaf type explain by trait variation?
«The strategy space at community-level (Fig. 2a; Fig. S4a, b) showed a clear split among three traditional plant functional groups based on leaf type and habit (broadleaved evergreen, broadleaved deciduous and needle-leaved conifer), which was less apparent at species level (Fig. S2a; Fig. S4c, d).»

We thank you for this question. We observed this by taking the mean PC loading values for each of the three groups at community and species levels (Fig. S6). For the community-level, we found that both mean PC1 and PC2 values among the groups were significantly different from each other. For the species-level, we did not observe the difference of PC1 values between BD and BE groups.

LI 250: You could add very briefly how you came about with this result. *Spatial variation in functional strategies.*

This is not a result, but a sub-title relating back to the expectations at the end of the introduction. We have changed the sub-title (and also the one of the next sections) to make this linkage clearer (line 272).

“Geographic variation of functional strategies”

Il. 254. Nice observation. A further explanation could be taxonomic peculiarities in the North America. “More extreme values of PC1 were found in the US than Europe, which could be linked to the broader variation in environmental conditions (e.g., more water stress and higher temperatures)⁴³ or possibly a homogenising effect of the more pervasive forest management in Europe⁴⁴.»

We thank you for this suggestion. We have added this information into the sentence (lines 276-279).

“More extreme values of PC1 were found in the USA than Europe, which might be a result of the broader variation in environmental conditions (e.g., more water stress and higher temperatures)⁴⁴, a homogenising effect of the more pervasive forest management in Europe⁴⁵ or different evolutionary histories.”

Il. 262: This is a very important comparison. Another reason for these different findings may consist of the different traits measured by remote sensing (fewer “size” traits, more economics CHECK xxx), as well as a number of factors like understorey vegetation, abundance, temporal variation, or seasonality.

“ This disparity may arise from differences in the ecosystems considered, with our results

only representing tree communities, whilst the 25 km scale of the VOD pixels integrates across all vegetation types.»

We agree that this is an important point. We have modified the sentence to read (lines 285-288):

“This disparity may arise from scale-induced differences in the ecosystems considered, with our results only focusing only on tree communities and based on measurements at the tree scale, whilst the 25 km scale of the VOD pixels integrates across all vegetation types.”

It now incorporates the points related to scale and other vegetation types. The trait extracted from remote sensing is believed to be directly linked to leaf water potential and therefore conceptually linked to the TLP trait that we use as an indicator for isohydricity. We therefore elected against bringing in the point related to size traits as they are incidental to the line of argumentation.

Ll. 340 “potential of applying these functional traits to explore forest biodiversity patterns and ecosystem functions at community-level or ecosystem-level^{41,42}–“ What does the choice of traits influence, which traits should be added in the future to better detect? Where to measure next?

In the previous paragraph on “Future challenges” we have identified the likely importance of expanding the underlying data to include traits related to rooting and capacitance. We have further expanded this discussion to specifically highlight the importance of separating managed and unmanaged forests. Given that this has been directly addressed at length immediately previous to the conclusion, we prefer not to repeat it here.

Ll. 342: I agree: “The identified geographical patterns of functional strategy could be thus used as a basis for parameterising forest hydraulic strategy in process-based vegetation models, allowing us to make predictions of forest function that are tightly grounded in observed community strategies. This will further enable, for instance, to assess how functional strategy affects the current and future forest carbon sink.»

Thank you very much for your agreement.

Ll. 359: Please rephrase: “Whilst to some extent this filter may be exerted by small-scale heterogeneity in edaphic and climatic conditions, there are also indications here that water stress strategy may be influenced by functional trade-offs across a much wider range of environmental conditions.»

We have rephrased the sentence (lines 412-415). Now it reads as:

“Whilst to some extent, this filter may be determined by small-scale heterogeneity in edaphic and climatic conditions, it also indicates that dominant water stress strategy may be influenced by the trade-offs among multiple functional traits across a much wider range of environmental conditions.”

Methods:

Il. 366: *Forest inventory data.* Be very explicit about what inventory data is in one sentence. Something like: “We used forest inventory data*, which are same-sized plot data including

their GPS -location and woody species' presence and tree or shrub trunk basal area. These data *are from ...”

We have added a sentence as suggested (lines 421-424).

“We used forest inventory data from the USA and six countries in Europe (Spain, France, Germany, Czechia, Poland and Sweden) (Table. S1). These forest inventory data included the geographic locations for the forest plots and for each individual living tree, the species and diameter at breast height (1.3 m).”

ll. 366 ff. Please add the size per plot.

The plot size varies between the forest inventory datasets. We have added the size information in the supporting file (Table S1).

ll. 400. Not a sentence. Change. “While the missing gaps were especially high (70%-90%) for these hydraulic traits.»

We have merged this with the previous sentence (lines 458-460).

“For well-studied traits such as leaf nitrogen content and wood density, about 40% of the woody species were missing, rising to 60%-80% for the hydraulic traits.”

ll. 407. Be more specific about “see below”, add the paragraph name, and add results to the appendix. “We carried out sensitivity analyses to explore the consequences of this imputation for our results (see *paragraph Statistical analysis*).*»

We have added it as suggested (line 466-467).
“(see the Statistical analysis section below).”

ll. 399 and ll. 458. The gap-filling procedure as well as the approach of the sensitivity analysis applied appear valid to me. I would like to see the figures, or at least the specific test results of Procrustes tests as described below. To have the approach solidified, a further analysis could be made for data observed: Compare the error of (observed) species-level traits when filled by genus mean. You could further compare Fig.2a with (1) (originally observed) species filled genus-level data as well as (2) observed species level, and your original Figure 2a with the Procrustes test.

We thank you for your suggestion to test the similarity of two datasets. For the original functional traits at species level, only a few species have originally observed values across all of the eight traits after initial species matching (Table S3). It is thus not feasible to compare the similarity between the original traits and traits filled by genus using this test.

Additionally, we test the effect of the different gap-filling methods in sensitivity analysis at species level and at community level of different grid sizes. At species level, we found that

there were no differences among the comparisons in Procrustes analysis (Fig. S7). At community level, for the forest communities at 0.1 and 0.5 grid cells, we did not observe the differences for the comparisons between the gap-filling methods (Fig. S8).

Ll. 375 It is a impressive resolution (0.25°)

Ll 447 I am not familiar with the relative importance analysis, and hope for my co-referees to judge this method.

Il. 277 I would suggest to add the figure references directly. “We found that water demand due to rising temperatures could be more crucial for the temperate forests than water availability *(figure 4)*.»

We have added it (line 307).

Il. 280 Did you consider to filter to the 20% most arid places? For habitats with low aridity there may be other limitations than water. “Contrary to our expectations, aridity, as quantified by the aridity index (AI), was not a strong determinant of community-level hydraulic strategy (Fig. 4b, e).»

We have filtered forest communities which are located into the arid habitats (aridity index<0.5). The relative importance analysis (RIA) was used to evaluate the relative importance of climatic variables (e.g., AI, MAP, MAT, Summer_T and Summer_P) (see the figure below). We found that higher mean summer temperatures were positively associated with PC1 scores (conservative to acquisitive strategies), whilst higher mean annual temperatures were positively associated with PC2 scores. These associations were consistent with the analysis of all forest communities.

Il. 282. Could you describe the different habitat types (or difference) these climatic variables represent? «Instead, we found that higher mean summer temperatures were positively associated with PC1 scores (Fig. 4c), whilst higher mean annual temperatures were positively associated with PC2 scores (Fig. 4f).»

We are not quite sure what you are referring to with “habitat types”. The climate variables can indeed, for instance, be separated into different biomes following e.g., Olson et al. (2001). However, our intention was to test whether there is an overall link between the hydraulic strategies that we have identified and climate. To separate by biome would break up the

biogeographical gradients that we are aiming to study here. We agree that local analysis of community strategy could be interesting, as suggested in our conclusions, but this may even require different methods, or at least much higher intensity sampling locally.

#####

Reviewer #2 (Remarks to the Author):

General comments

The manuscript by Liu et al. is well written and addresses a very timely issue with an impressive dataset. Based on data from 219,518 forest inventory plots comprising 12,445 communities, they studied trait associations among a set of eight functional traits linked to the conservative-acquisitive spectrum and to drought response. A main focus is the comparison of trait associations on species and on community level. In addition, the paper discusses relationships between axes of shared trait variance on community level and the environment, as well as differences in traits between communities dominated by different plant functional types. The authors find the associations between traits to be largely consistent across different organizational scales, and robust to the mode of data imputation and grain size used to aggregate community data. They report moderate but significant associations between axes of trait variation and climate predictors, especially temperature. Moreover, they find both different plant functional types and communities dominated by different plant functional types to differ in the PCA scores. Finally, they map the community PCA scores to show regional patterns in the prevalence of different prevailing functional strategies. Especially the latter may be useful to calibrate and improve large-scale dynamic vegetation models.

We thank you very much for your summary.

In general, the manuscript is well thought-through and pleasant to read, however there are certain conceptual issues that urgently have to be addressed before it can be considered for publication, most importantly with respect to the concept of ecological strategies. For this reason, I am afraid I have to reject the paper in its current state. However, I strongly encourage it for resubmission after addressing the concerns that I will list in the following.

We have modified the description based on your suggestion. Please see the responses below.

Specific comments

Major comments

The most important issue I had when reading the paper was its somewhat loose use of ecological concepts and theory, specifically regarding the concept of functional/ecological strategies. While the term “strateg*” occurs a total 119 times in the manuscript, it is not defined anywhere in the text. This is problematic, because first there is more than one definition of strategies, and second because I am not sure that the authors actually use it in a correct way. A central tenet of the present paper is that the ecological strategies of the plants present in a plant community can be scaled up to a “community-level strategy”. This idea seems dangerous to me for reasons I will detail in the following. To do so, it is important to set straight what I mean when I talk about strategies. In the broadest sense, ecological strategies can be defined as “how a species sustains a population” (Westoby et al., 1998). This definition is prominently cited in the first paragraph of Reich et al. (2014), an excellent

article which showcases the importance of being clear about concepts. When talking about functional strategies, it is moreover important to be clear about the definition of functional traits. Here, I follow the definition of Violle et al. (2007), who understand functional traits as “morpho-physiophenological traits which impact fitness indirectly via their effects on growth, reproduction and survival, the three components of individual performance”. Due to this link to individual performance, functional traits are evolutionarily selected for. Hereby, certain combinations of traits are favored due to environmental pressures and a number of trade-offs reflecting allocation constraints. In the context of functional traits, a strategy can be viewed as this whole “set of traits” / “collection[...] of traits in syndromes” (Reich et al., 2003) that is jointly selected for. Evolutionary selection acts on the individual level (or more precisely, on genes that are manifested in individuals), and ultimately drives the differentiation of species as separate entities with different sets of functional traits. The sets of average functional traits of different species therefore reflect different functional strategies that were favored by selection; thus, it makes sense to look at strategies at the species level. However, there is no evolutionary selection acting at the community level. While working on my review, I spoke about the issue to three different ecologists with experience in functional trait theory – all agreed that the concept of "community-level strategies" is fundamentally flawed as it implies the presence of group selection. The same logic holds true for the concept of “trade-offs”, which are associated with evolutionary constraints acting at the individual level.

We thank you for the discussion on the application of functional strategies of forest communities and listing the references to support your suggestions. After carefully reading the references, we have accepted your suggestions to change them as “locally dominant strategy” throughout the manuscript.

My most central recommendation to the authors would therefore be to work through the entire manuscript and replace all mentions of “community-level strategies” and “community-level trade-offs” with a more careful wording that is not at odds with basic biological theory. This should ideally be combined with clear definitions of the most central concepts in the introduction. Patterns in community-weighted means (or community functional parameters sensu Violle et al. 2007) are often thought to reflect trait combinations that are advantageous at a given site and hence a “locally ‘optimal’ trait strategy” (though this does not always have to be the case, see Muscarella & Uriarte, 2016). Already using the term ‘locally optimal strategies’ instead of ‘community-level strategies’ would resolve many of the conceptual issues, because it reflects that a community does not have a strategy, but the individuals that belong to it (and some of these strategies are more optimal than others). Functional strategies advantageous under the local conditions (i.e. trait combinations that confer an evolutionary advantage to the individuals that possess them) are likely to be more prevalent among the individuals in a community, which will be reflected in the patterns in community weighted means. For those reasons, the trait associations on community level will likely be similar to the associations on individual or species level. However, these patterns plainly emerge from the prevailing strategies among the individuals in a community, and are not themselves subject to evolutionary trade-offs.

We thank the reviewer for this clearly explained and well-grounded critique. We appreciate that the terminology we used was not appropriate. We have now changed the manuscript to refer to the “locally dominant strategy”. We decided not to go with “optimal”, because with forest management it does not necessarily follow that species are in their optimum locations with respect to the strategy-environment interface. We have removed all instances of

“community-level trade-off”. We have also added a definition of strategy and of functional traits to the first paragraph.

Muscarella, R. & Uriarte, M. Do community-weighted mean functional traits reflect optimal strategies? *Proceedings of the Royal Society B: Biological Sciences* 283, 20152434 (2016).
Reich, P. B. et al. The Evolution of Plant Functional Variation: Traits, Spectra, and Strategies. *International Journal of Plant Sciences* 164, S143–S164 (2003).
Reich, P. B. The world-wide ‘fast–slow’ plant economics spectrum: a traits manifesto. *Journal of Ecology* 102, 275–301 (2014).
Violle, C. et al. Let the concept of trait be functional! *Oikos* 116, 882–892 (2007).
Westoby, M. A leaf-height-seed (LHS) plant ecology strategy scheme. *Plant and Soil* 199, 213–227 (1998).

Minor comments

Abstract:

L75: “...while those associated with water storage-isohydricity loaded along a second.” – I would be careful with the wording here. There are neither data on capacitance nor any metrics of isohydry or stomatal control in the paper, so referring to water storage and isohydry in the abstract seems misleading.

We associate TLP with isohydricity following e.g., Fu & Meinzer (2019) indicate wood density as a possible indicator of capacitance. However, we take the point that the abstract should be more reflective of this (lines 74-76). It now reads:

“Traits associated with acquisitive-conservative strategies formed one dimension of variation, while leaf turgor loss point, associated with stomatal water regulation strategy, loaded along a second.”

Introduction:

L132ff: “How the range of community-level functional strategy changes as a function of water stress is not well established.” – I would recommend to reword this sentence a bit. How would you define (and quantify) the “range of ... [a] functional strategy”?

We have revised this sentence (line 138-139). Now it reads as:

“How the locally dominant strategy varies as a function of water stress is not well established.”

L135f: “to underlie quantitative predictions” – this sounds weird – do you mean something like “to lend mechanistic support to quantitative predictions”?

We have modified this sentence (lines 139-143). Now it reads as:

“In order to study the range of locally dominant strategy changes with water stress, it is fundamental to understand the likely response of the forest community to ongoing environmental changes, both in a qualitative sense and to underlie quantitative predictions by vegetation models.”

L137f: “but require clear evidence on the distribution of forest strategies” – do you mean “require precise data”? Because you need data to parameterize a model, evidence seems a slightly incorrect term here.

We have changed this sentence (lines 143-145). It reads as:

“These models are currently taking big steps forward in their capacity to simulate physiological processes relevant to water stress responses²⁹⁻³², but require precise data on the distribution of forest water stress strategies.”

L138ff: “An exploration ... would provide, as well as opening up ...” – inconsistent verb tense – please check grammar!

We have revised this sentence (lines 145-148). Now it reads as:

“An exploration of the dominant water stress strategy of forest communities along climatic gradients would promise a new perspective on forest functional biogeography and open up opportunities to support large-scale projections of forest functions using vegetation models.”

L143f: “regions of the USA (United States of America) and Europe (Spain, France, Germany, Poland, Czech Republic and Sweden)” – this apparent parallelism is confusing: the first set of parentheses defines the acronym, the second one lists the regions. Maybe get rid of these parentheses entirely?

We have changed the description (lines 149-151). Now it reads as:

“Here we combined a large dataset of functional traits for woody plants with forest inventory plot data across regions of the United States of America and Europe, including Spain, France, Germany, Poland, Czech Republic and Sweden.”

L147: “We aggregated these forest plots into a grid level” – not sure where the level comes from and what you mean. “into a grid?”

We have changed it (lines 156-158). Now it reads as:

“We aggregated these forest plots according to 0.25° x 0.25° grid (grid-cell level statistics are here-after referred to as community-level) to reduce stochasticity and then calculated community-weighted mean traits for each community separately.”

L152f: “We expected to see strong trade-offs related to the structural strategy” – do you really expect trade-offs on community level, or do you expect that existing trade-offs acting on the individual/species level are reflected in community level data? See major comments!

We have removed this sentence - we agree that it was neither appropriately expressed or necessary.

L159: “community-level strategies” – this concept urgently has to be defined (see major

comments!). I did a brief google scholar search and the only hit I got for the terms ["community-level strategies" "functional traits"] that was relevant was a single pre-print: Neyret, M. et al. A fast-slow trait continuum at the level of entire communities. 2023.07.12.548516 Preprint at <https://doi.org/10.1101/2023.07.12.548516> (2023).

Please see our response to the major comment above. We have worked to resolve the problems with terminology.

Methods and Material:

[[--I will address the issues in the Methods and Materials before I move on to the Results and Discussion, because I think that this is the way people should read papers, and that Nature should get rid of the methods-last style as it encourages uncritical reading--]]

L373f: "We selected the living trees in each plot and included only those with a diameter at breast height larger than 10 cm (12.7 cm in USA, 12 cm for Czech Republic)." -- I take that this difference is due to different census methods (and the use of "freedom units" in the US) but wouldn't it make more sense to filter out data of all trees below 12.7 cm in the European plots as well? This may decrease the total sample size, but I imagine it is worth it too avoid age-driven differences between regions.

We thank you for this suggestion to use the same tree diameter threshold across different forests inventory datasets. The results are all based on the from the use trees of the 12.7 cm threshold (lines 429-431).

"We selected the living trees in each plot and included only those with a diameter at breast height larger than 12.7 cm across all the datasets."

L377ff: "We selected all the woody plants (trees and shrubs) ... based on the information from TRY ... and the Woody Plant Database..."-- I am not sure if I understand this correctly. Based on information about what? Information about woodiness? Or presence of information in the databases?

We have added the "woodiness information".

L378: "community" – "community"

Revised it.

L381f: "In total, 749 woody species were retained for the analysis." – it would be interesting to know how many species were removed, why they were removed, and if the fraction of removed species/trees differed between ecosystems.

We have removed one cactus species. Here I added the information (line 439).

"The cactus species - Pilosocereus royenii was removed."

L382ff: "To further distinguish the different strategies of woody plants, broadleaves were split into two groups according to their leaf phenological types (evergreen and deciduous). We thus classified..." – were there no deciduous conifers like Larix spp. in the dataset? Sounds unlikely given the studied range.

Yes, there are some deciduous conifers. But the conifer group does not have such as rich species number (121 species) as other two groups and only eight conifer species were deciduous. Hence, here we only simplified three functional groups. We have added this in the lines 442-443.

“For conifer group, the species count is not as rich as in other two groups, with only eight conifer species being deciduous.”

L392f: “turgor losing point” – are you sure that was the term you searched? I just checked and found not a single hit for “turgor losing point vs. ca 3670 for “turgor loss point”. Equally I only got 35 hits for “embolism vulnerability curve”. Maybe you did not use quotes? Anyway, to me this seems uncommon terminology for the field.

Apologies, this was a typo. Here it should be turgor loss point. We have changed in the whole text. Moreover, we have added the consistent term of “slope of the vulnerability curve”.

L400f: “While the missing gaps were especially high (70%-90%) for these hydraulic traits.” – While is a conjunction that introduces a subordinate clause. Normally, such a clause cannot stand on its own. Was this clause supposed to be connected to the previous sentence?

We have revised this sentence (lines 458-460). Now it reads as:

“For well-studied traits such as leaf nitrogen content and wood density, about 40% of the woody species were missing, rising to 60%-80% for the hydraulic traits.”

L401f: “For the species-level functional traits that were missing, we used the median trait values of the same genus instead. And if the trait was still missing, we used the median values of the family to fill the gaps⁵⁵.” – am I understanding it correctly – the mechanistically most proximate traits for plant drought responses are based on imputation for 70-90 % of the species?

Yes, the trait missing gaps of hydraulic traits were 60-80% at species level after initial compilation. Due to the large trait missing gaps, we imputed the trait values at species level with three different methods and their sensitivity tests indicate that the results were robust (Fig. S5, S7).

L409: “...using the formula described by 57” – this looks weird with the Nature citation style (an in-text reference would be needed here in my opinion). By the way, the equation is really just a weighted average so I am not sure if this merits a reference (especially since in that paper it is weighted by biomass, not basal area).

We have revised the reference and citation style (lines 468-469). Now it reads as:

“We calculated mean values for each functional trait for each community-level, weighted by the relative basal area of the species in that community²⁵.”

L409ff: “The functional traits, except wood density, were log-transformed to increase the symmetry of their distribution.” – before or after calculating the weighted mean (hint: before makes more sense).

Yes, they were log-transformed before calculating the weighted mean. We have clarified this in the text (lines 469-471).

“The functional traits, except wood density, were log-transformed to increase the symmetry of their distribution before the calculation of weighted mean values.”

L432f: “According to the geographic information (e.g., coordinates) of each community, we extracted the climate factors to study the patterns of functional strategies across the regions.” – the environmental data are on a different spatial scale than the gridded community data. One of the $0.25^\circ \times 0.25^\circ$ community grid cells (cf L375) covers about 400 of the $1 \times 1 \text{ km}^2$ climate grid cells (at a latitude of 44°). It makes sense to explain how this was accounted for when extracting the climate data for the communities. Based on a cursory reading of Moudrý et al (2023) I guess your best bet may be to extract climate data from the unaggregated locations, then aggregate, but I guess it is worth thinking about it in more detail. Moudrý, V. et al. Scale mismatches between predictor and response variables in species distribution modelling: A review of practices for appropriate grain selection. *Progress in Physical Geography: Earth and Environment* 47, 467–482 (2023).

We have followed your suggestion to extract the climate information based on plot coordinates and then aggregated them into the grid cell ($0.25^\circ \times 0.25^\circ$) by averaging their values. We have added more details in the Method and Materials (lines 493-499). Now it reads as:

“Because the resolution of forest plots differs among the forest inventory datasets, we used the climate factors from WorldClim with the spatial resolution of 2.5 arc minute ($\sim 5 \times 5 \text{ km}$). For the aridity index, we aggregated resolution from 30 arc second to 2.5 arc minute resolution. We extracted the climate factors for each plot individually and then aggregated them into a grid cell ($0.25^\circ \times 0.25^\circ$) by taking the mean of their values to study the patterns of locally dominant strategy across the regions.”

L435ff: “Principal component analysis (PCA)” – I am missing information about transformation, scaling/centering of the raw data before calculating the PCA. You mention above that all traits besides wood density were log-transformed, but did you also scale them to unit variance? If not, the standard deviation of each variable will influence their scores – if so, it is not easily possible to compare scores between Fig 2 and Fig S2. `factominer::pca()` scales to unit variance by default, so this is critical information.

We have rescaled the data to unit variance using the `scale=TRUE` and `center = TRUE` when analysing the PCA tests. We have clarified this in the methods (lines 505-507). It reads as:

“The selected traits were scaled and centered before running PCA tests, which is beneficial to the principal components that accurately capture the underlying structure and variance of the data.”

Moreover, I believe it may be useful to explain what was the rationale to choose the specific set of traits you included in your PCA, because PCAs are extremely sensitive to the choice of included variables. If you want some variables to turn out on the first “most important” axis, just make sure that you put in a bunch of closely correlated variables, and you are set. High R^2 , and just the results you want. In this case, high leaf N, Ks and LS are all associated with more acquisitive strategies, while LMA is associated with more conservative strategies, so it

is not surprising that all these score high on the first axis. To make such an analysis credible, it makes sense to clearly clarify whether this set of variables was selected a priori and not modified after seeing the data, and why each variable was included.

We fully agree with your point. The selection of functional traits is sensitive in PCA, and it should be clarified clearly why the traits are selected into PCA test. We selected the eight traits in our analysis that are based on the knowledge of tree-level adaptation (Fig.1a) and main representative strategies at species level (Fig.1b). The selection was strongly influenced by the availability of observations, but also filtered through the perspective of traits that are mechanistically informative for process-based modelling. It is this latter point that has led us to choose slope for example, because we expect that the rate of xylem cavitation as a function of xylem water potential is a powerful parameter in determining mortality rate under drought stress (e.g., Choat et al., 2012). Similarly, we included several structural traits such as LMA and wood density, because without this information a process-based model simply cannot be expected to grow a realistic tree (central as they are to decisions around plant carbon allocation) (see, e.g., Sitch et al., 2003).

We have added a sentence to clarify it (lines 501-505). It reads as:

“All eight functional traits were selected a priori before the PCA test, based on these traits being strongly linked to acquisitive-conservative, structural, stomatal and water storage strategies in the literature (Table 1). We also took into account the importance of these traits, or aspects of function that these traits act as proxy for, in process-based vegetation modelling 55.”

Ultimately, however, the choice of traits included will always have an arbitrary component. There is no single objective way to make such decisions - even if one defines a statistical test, an appropriate threshold for that test must still be decided on. The important aspect of our analysis is that two different dimensions emerge which allows us to explore the dominant strategy of forest communities more concisely than analysing and presenting values for each trait individually.

For instance, there are only few papers in the vulnerability curve literature that identify systematic patterns in VC slope, so I found it interesting that this was included instead of e.g. hydraulic capacitance or g_{min} , which are mechanistically likely much more relevant (some rascals might even think that maybe not including Slope would have messed up the nice first LES axis? 😊).

Please see our response to the previous comment. We have included the rate of xylem cavitation (Slope here) because this trait as a function of xylem water potential is expected to be a powerful parameter in determining mortality rate under drought stress (Choat et al., 2012; 2018).

We would have indeed been happy to include hydraulic capacitance, and we had conducted searches for this trait, but did not find such a rich dataset as for the other traits and thus decided not to carry it forward into this analysis.

Additionally, in order to test whether excluding the slope will shift trait variation, we have re-run the PCA test without the slope (as Figure 2 in the main text). We found that excluding slope did not affect the trait variation along the two dimensions (see the figure below).

L454ff: ““Lindeman, Merenda, and Gold,” which is a commonly suggested approach for determining relative importance 65. “ – reference 65 is the paper by Lindeman, Merenda and Gold, so I doubt that this is unbiased evidence for the claim that the approach by Lindeman, Merenda and Gold is a good one...”

We have revised this sentence (lines 527-528). It reads as:

“Here, the type parameter is set to “lmg” method⁷⁴, which is a commonly suggested approach for determining relative importance.”

L470: “ggplot2 R package (version 3.4.3).” – if you explicitly mention package and version, I would also add a citation – if anyone deserves it, then these authors.

We have added the citation for the packages.

Results and Discussion:

L202ff: “Contrary to the general expectation that more robust xylem was linked to higher wood density³⁶, the slope had extremely weak correlation (Pearson correlation coefficient = -0.04) with wood density...” – the slope of a vulnerability curve itself does not contain information about embolism resistance, just about the change in conductance per unit change in water potential (you can have curves with drastically different P50 and the same slope). Therefore, I would not believe a priori that the VC slope should be related to wood density at all, and would require a strong theoretical justification as to why to assume this (the same holds for the slope ~ turgor loss point link). By the way, if you were referring to the slope – WD relationship in Fig. 6b of reference [36] (Hoffmann et al., 2011): this plot shows the slope of a stomatal response curve, not a vulnerability curve.

We appreciated your suggestion. We have revised this sentence since it is not well linked to the work we introduced here (lines 227-234).

“More robust xylem (i.e., more negative P50) was associated with higher wood density, which is likely driven by the predominance of conifers at more negative P50. But the correlation was relatively weak and P50 and wood density largely loaded orthogonally on the PCA, consistent with wood density being a complex trait which is associated with many ecological processes 16,35,39. The cavitation slope had an extremely weak correlation (Pearson correlation coefficient = -0.03) with wood density and had also relatively low correlation (Pearson correlation coefficient = -0.32) with leaf turgor loss (Fig. 2b).”

L205f: “This result suggests that water storage choices are more complex than a simple function of either structural acquisitiveness or stomatal strategy” – why? There are no data about water storage. Wood density is indirectly linked to water storage capacitance due to a trade-off driven by spatial constraints, but I would be careful to interpret it as exchangeable with water storage.

We thank you for this suggestion. We agree that wood density is a weak indicator of water storage. We have revised the description (lines 228-230) and also added a sentence in the Future challenges (line 371-372).

“But the correlation was relatively weak and P50 and wood density largely loaded orthogonally on the PCA, consistent with wood density being a complex trait which is associated with many ecological processes 16,35,39.”

“While wood density is widely studied, it is only a weak indicator of potential capacitance³⁹, because it is still not known how much stored water is accessible.”

L213ff: “There was, however, 11.8% ... more explained variance in the community-level when compared to species-level, indicating some simplification of the strategy space when scaling up.” – I am not sure that many readers will understand the biological interpretation of “simplification of the strategy space”. Is there a less convoluted way to say this? Could this have to do with averaging out a sizeable fraction of the variance?

We have revised this sentence to make it clearer (lines 246-247). Now it reads as:

“..., which resulted from a higher loading contribution along the first dimension when scaling up.”

L218f: “This may reflect optimization of resource usage (e.g., competition for light and forest growth) within forest communities” – are you sure? Selection does not act on community level.

We have revised this sentence (lines 249-251). Now it reads as:

“This may reflect optimization of resource usage (e.g., competition for light and forest growth) at species level affecting forest community-level properties 11,41.”

L219f: “Additionally, the full range of PC1 score was wider at community-level than that of species-level” – see comment to L435ff: if the standard arguments of `factminer::pca()` were used (i.e. “scale.unit = TRUE”), I would be very careful when comparing axis scores and linking them to “larger trait variation” – the variances in both PCAs are associated with different ranges in traits!

We have deleted this sentence since it no longer fits.

L228f: "...illustrates the importance of taking account of the full range of hydraulic strategies..." – would you really say you did that if 70-90% of the hydraulic trait values are imputed?

The point that we are trying to make here is that the range of strategies seen at species level also appear at the level of locally dominant strategies. We have adjusted the sentence to read (lines 257-261):

"The consistency of the strategy space across both species and community levels illustrates that considering locally dominant strategies does not appreciably simplify the strategy space that needs to be taken into account in large-scale assessments of forest properties and ecosystem functions^{42,43}."

L269ff: "There are, however, large areas in eastern USA and Central Europe that do not tend towards either strongly anisohydric or isohydric, indicating that tree species are ordered on a continuum rather than a dichotomy⁴⁶." – yep and also because more and less isohydric and anisohydric species coexist in the same ecosystems – as well as acquisitive and conservative species. Aggregating on the community level discards information about this very obvious fact.

We have added a sentence to clarify the result (lines 295-298). It reads as:

"This result may also arise from coexistence of species with different isohydric strategies making up similar proportions of the same community, in which case the community weighted means used here may be more indicative of overall community behaviour than a single dominant strategy."

L284: "The link of PC1 with mean summer temperature was not reported before" – if no-one else did a PCA with the exact same variables, it is not surprising that it was not reported before. While it seems tempting to see PC1 as a metric of the spectrum of conservative to acquisitive strategies, in fact a PCA axis is a linear combination of all variables in the analysis (literally just $X \% * \% < \text{corresponding eigenvector} >$), so comparing the PCA results from a PCA with LES traits and hydraulic traits to PCA results from a PCA using for example the classical LES traits (LMA, Amass, N, P, Rmass and LL, as in Wright et al., 2004) is pretty difficult.

We have deleted this sentence. Now we have added a sentence on the low relationship between trait and environmental factors (lines 316-318).

"However, the relationships between PC scores and climate factors were also relatively weak, aligning with the patterns observed in global trait-climate relationships⁴⁸."

L305ff: "Higher spatial resolution work will be necessary to identify and distinguish which factor causes the one water stress strategy to prevail over another." – while I agree that high-resolution data never hurt, I believe that in this case a better coverage of species with actual measurements of hydraulic traits would be much more beneficial.

We have added this description on actual trait measurements (lines 340-342).

“Additionally, a larger coverage of species with observations of functional traits, in particular hydraulic traits, will be beneficial for evaluating the local dominant strategies of forest communities.”

L317ff: “Moreover, identifying the main functional traits in response to water stress is crucial, which may lead to apparent discrepancies in exploration of trait trade-offs and main functional strategies⁷.” – This sentence seems off – why would identifying the main functional traits lead to discrepancies?

We have revised this sentence (lines 363-364). It reads as:

“Identifying how functional traits respond to water stress is crucial, which may lead to apparent discrepancies in exploration of trait trade-offs and main functional strategies.”

L321: “availability⁵¹ The strategies” – a full stop is missing before “The”.

Added.

L329: “regular cutting might be expected” – I believe a comma is missing before “might”.

Added.

L345: “surprisingly the relationships with climate are relatively weak” – is this really so surprising? To me, it seems in line with general patterns in trait-environment relationships (for instance see Anderegg 2023).

Anderegg, L. D. L. Why can’t we predict traits from the environment? *New Phytologist* 237, 1998–2004 (2023).

We have deleted the “surprisingly” in the sentence.

Tables:

Table 1:

1. I would reformat the units in the second column using negative exponents rather than fraction lines (division is left-associative unless stated otherwise, so technically it is correct, but almost all formatting guidelines ask for parentheses to avoid ambiguities in such cases).

We have revised the units in table 1 as you suggested.

2. For Ks, MPa got auto-corrected to Mpa – please fix.

We have revised it.

3. “Leaf mass area” – some people seem to call LMA this way but I find “leaf mass per area” much more logical because “leaf mass area” leaves the impression that the area ends up in the numerator.

4. “Slope for the embolism vulnerability curve between P50-P88” – this is a very niche definition for a vulnerability curve slope – normally, the used models estimate the local slope (i.e. first derivative) of the curve at one of the critical water potentials (mostly commonly P50). Is there a rationale behind using such an uncommon definition? Doesn’t this complicate comparison with other studies?

We have adopted this version to easily relate between P50 and P88, with the latter linked to mortality thresholds (Meinzer & McCulloh, 2013), which makes it a particularly relevant trait for modellers. We take the reviewer’s point that this is not such a common definition, but also note that whilst there are numerical differences between the two approaches, the basic character of the results would not be affected by using the other definition of slope (which in any case requires to make an assumption about the shape of the curve between the two points).

2) Btw. if you really calculate slope like this (and also in the other case) the unit of the slope is % MPa⁻¹ (assuming the curve to be described in terms of PLC).

We have changed the unit of slope to % MPa⁻¹.

5. “. A less negative TLP indicates more isohydric.” – “more isohydric behaviour”?

We have changed it to “more isohydric behaviour”.

Table S2: check for consistent capitalization (see e.g. No. 5 blackman et al.).

We have revised it.

Figures: Figure 1 and Figure 2 differ in naming from Fig. 3 and Fig. 4 – please be consistent.

We have revised them.

Figure 2: This figure seems a bit stretched horizontally – does it have 1:1 aspect ratio or did it get messed up in Word? (The same is true for Fig. S6, but not so much for S2).

We have modified the subplots in figure 2. Now it only has two subplots. Moreover, we also modified the figure in supporting file.

Figure 4: I feel that you would get a better linearity in Fig. 4 b and e if log-transforming the aridity index – this transformation would be well justified as the AI is a ratio, which is more easily interpretable on a logarithmic scale. In either case it makes sense to reflect about the biological significance of a statistically significant relationship that only explains 1% of the variance in a PCA axis.

We have tried logging the AI relationship, which results in higher R² values (both R²=0.06). Please see the figure below. But divergence strategies at low aridity is less visually clear when using logged values. Therefore, we prefer to show the original aridity index values and PCA axis relationship.

Figure caption: Effect of climatic factors on the dominance of functional strategies of forest communities (PC1 in upper panels and PC2 in lower panels). a and d, Result of the relative importance analysis of the individual climate variable for the strategies along PC1 or PC2. The bar plot and error bar show the bootstrapped estimates of the relative importance of each climatic variable and the confidence interval respectively. b and e, the linear regressions with the aridity index (logged values), as hypothesized in the introduction. c and f, the regressions with the most important climatic variable for the PC1 (mean summer temperature: Summer_T) and PC2 (mean annual temperature: MAT) separately.

Figure S7: This figure seems stretched vertically – or is this a weird projection? In either case the maps in the main text look better.

We have modified it. Please see the supporting file (Fig. S13).

Reference:

- Choat, B., Jansen, S., Brodribb, T. J., Cochard, H., Delzon, S., Bhaskar, R., ... & Zanne, A. E. (2012). Global convergence in the vulnerability of forests to drought. *Nature*, 491(7426), 752-755.
- Choat, B., Brodribb, T. J., Brodersen, C. R., Duursma, R. A., López, R., & Medlyn, B. E. (2018). Triggers of tree mortality under drought. *Nature*, 558(7711), 531-539.
- Fu, X., & Meinzer, F. C. (2019). Metrics and proxies for stringency of regulation of plant water status (iso/anisohydry): a global data set reveals coordination and trade-offs among water transport traits. *Tree Physiology*, 39(1), 122-134.
- Liu, H., Gleason, S. M., Hao, G., Hua, L., He, P., Goldstein, G., & Ye, Q. (2019). Hydraulic traits are coordinated with maximum plant height at the global scale. *Science Advances*, 5(2), eaav1332.

Meinzer, F. C., & McCulloh, K. A. (2013). Xylem recovery from drought-induced embolism: where is the hydraulic point of no return?. *Tree physiology*, 33(4), 331-334.

Olson, D. M., Dinerstein, E., Wikramanayake, E. D., Burgess, N. D., Powell, G. V., Underwood, E. C., ... & Kassem, K. R. (2001). Terrestrial Ecoregions of the World: A New Map of Life on Earth: A new global map of terrestrial ecoregions provides an innovative tool for conserving biodiversity. *BioScience*, 51(11), 933-938.

Sitch, S., Smith, B., Prentice, I. C., Arneth, A., Bondeau, A., Cramer, W., ... & Venevsky, S. (2003). Evaluation of ecosystem dynamics, plant geography and terrestrial carbon cycling in the LPJ dynamic global vegetation model. *Global change biology*, 9(2), 161-185.

REVIEWER COMMENTS

Reviewer #1 (Remarks to the Author):

Second Review of the manuscript entitled Mapping multi-dimensional variability in water stress strategies across temperate forests by Liu et al.

General statement and summary: Thank you for the detailed answers to my points raised. I enjoyed reading the manuscript again. The authors thoroughly incorporated the suggested changes, and I hope and think the manuscript could profit from these.

In detail, the authors improved their manuscript in terms of clarity, analysis, and ecological embedding.

Overall, I think this is a well-written paper, with a sound analysis and interesting results that give rise to further questions.

Minor:

Only in a couple of parts, I missed some clarity or a simpler language (but I am not a native speaker).

Clarity. In few places small changes could increase clarity.

- Thank you for integrating changes into (earlier lines 126). With re-reading, I would again take the chance to clarify the sentence more. I am sure the authors can do a better job than me. Possibly I have difficulties understanding the purpose of this sentence: Lines 133 – 134: "When functional traits at the species-level are assembled into communities, the dominant strategy depends on the balance between niche complementarity and resource competition" I read two slightly different things in there. You are likely describing (a) the assembly process of species into a community - based on their plant traits, and not (b) the way how the dominant strategy arises methodologically from the aggregation of trait data.

For (a) Could you clarify not traits are assembled but species based on their traits? Possibly very briefly define "dominant strategy" in this context (unless I missed it).

- Rather introduce the models before referring to them Line 143: "These models [...]"

- Suggestion to change and simplify. Lines 135 -136, e.g. (only a suggestion, I am not an English native speaker) "As With reduced water stresswater resources become less limiting, competition may tend to favour more acquisitive strategies, resulting in a convergence towards a limited range of water-demanding strategies 15,16,26».

- Logic: Clarify if these strategies are (a) related to species' plasticity, or whether they are (2) present due to an assembly processes. Lines 137-138: "In water-limited environments, a wide range of water stress strategies can develop partitioned environmentally according to particular combinations of conditions that favour certain strategies7,27,28.» E.g. (only a suggestion, I am not an English native speaker) In water-limited environments, a wide range of water stress strategies may survive/be present. These strategies have shown to be ecologically partitioned according to specific combinations of environmental conditions (add reference).

- Check this sentence again Lines 151-154: "We considered eight continuous functional traits of woody plants (shrubs and trees) related to potential forest functional strategies to water stress (Table 1), concentrating on acquisitive-conservative, structural, stomatal and water storage strategies, as the information on rooting traits is very limited. «

- I commented on this before, I am wondering if one could harmonize this to either question or description of analytical steps. Lines 162 – 166: (I am sure the authors can make a better job than me, in order to illustrate what I mean with extracting the question, see this example:) "We (a) reduced the water stress traits at community-level by means of a principal component analysis (PCA), (b) compared it to the one at the species-level and (c) related these strategies to geographic and climatic space of temperate forest communities (Fig. 1c, d).»

Reviewer #2 (Remarks to the Author):

General comments

In the revised version of the manuscript by Liu et al., I appreciate the thorough effort directed at answering all reviewer comments in great detail. I feel that the authors in general did a great job at considering reviewer suggestions, in many cases accepting the suggested changes, while explaining their motives clearly in places where they decided against the reviewer input.

As stated in the last round of reviews, I feel that ecological strategies (i.e., sets of traits/collections of traits in syndromes that are jointly selected for, cf. Reich et al. 2003) are a concept that is not meaningful on community level, because evolutionary selection does not act on communities. For this reason, I am very glad that the authors agreed to downweigh the focus on "community-level strategies" and "community-level trade-offs". Their choice of talking of "locally dominant strategies" instead is certainly an improvement, as in this phrasing it is acknowledged that strategies reflect processes that do not act on community level. Indeed, ecological strategies are now clearly defined in the manuscript sensu Westoby (1998) as "the set of adaptation[s] or behaviours that allow a species to maintain a population", i.e., defining strategies as properties of species, not communities.

However, I feel that the paper still could benefit from being more precise about the levels of aggregation the analysis is based upon. In the following, I will briefly expand on the issue I am having with drawing inference on ecological strategies based on community level data, and then list a couple of minor issues that I feel should be addressed before publication. Pending these suggested changes, I feel the paper now is a good candidate for publication in Nature Communications.

Specific comments

Major comments

Apart from the species-level analysis shown in the supplementary material, the main focus of inference in the paper is on community-level data. However, there are many instances where conclusions are drawn about processes acting at the individual or species level based on community-level data, which constitutes a logical fallacy, the so-called ecological fallacy (Robinson 1950, see Pollet et al. 2015 for a friendly low-level explanation directed at biologists, albeit in a very different context).

In my opinion, the drivers of association between traits on species level (specifically evolutionary selection and allocation trade-offs) are very different from the processes acting on community level (community assembly processes like environmental filtering). While the main patterns in trait association on species level will likely remain visible on community level averages, there is no guarantee that this is always the case. Indeed, when there is more than one viable strategy to deal with environmental constraints and these strategies are rather different, the community average of traits may not be very meaningful at all. As an example, in tropical dry forest ecosystems you find species with a high wood density, low stemwood capacitance, rather embolism resistant wood, and small, hard, evergreen leaves, i.e. species that have a water stress tolerance strategy. These species coexist with stemwater-storing species with low wood density, high capacitance, low embolism resistance and large, thin, drought-deciduous leaves, i.e., species that have a drought avoidance strategy. Averaging over all these traits may result in an intermediate trait combination that is not present on the site and might not even be viable under the local conditions. This hopefully illustrates that CWMs do not necessarily describe a "locally dominant" strategy.

While I think that with an appropriate caveat it can be permissible to interpret CWMs as an approximation to locally dominant strategies, I feel it problematic to use community-level data to jump back to conclusions on the individual or species level, and personally would probably rather speak of trait associations on community level rather than strategies. I have the impression that currently, there are still many places in the paper that either use community level data to infer about species-level strategies, or, even more problematic, imply that strategies act at or are measurable at the community level, for example (list not exhaustive):

L188: "Variation in strategy at the community level"

L188f: "dominant community functional strategies"

L255: "The strategy space at community-level"

L257f: "The consistency of the strategy space across both species and community levels"

L272ff: "Geographic variation of functional strategies. [...] Acquisitive strategies [...] were mainly distributed in [...], whilst conservative strategies [...] were distributed in[...]."

L305ff: "Climatic factors for variation of functional strategies" – in this section, all the regressions with PCA scores derived from community level trait averages are interpreted in terms of ecological strategies.

L310f: "dominant community-level hydraulic strategy"

L334f: "forest communities can follow a broad range of water stress strategies"

L394f: "consistency in trait coordination and trade-offs at both community-level and species-level"

L507f: "functional strategies at species-level and community-level"

In many cases, rewording the findings in a way that does not conflate inference on different aggregation levels is easy, and actually closer to what actually happened during analysis. For instance, instead of saying that "Acquisitive strategies [...] where mainly distributed in central and eastern USA and Northern France", you could say that ... that "Trait combinations consistent with an acquisitive strategy [...] where more prevalent in plant communities in central and eastern USA and Northern France". Analogously, I would recommend avoiding conclusions on strategies (which make sense on individual or species level) based on community level aggregates elsewhere in the paper and rephrase it in a more careful, hedging way that acknowledges that you talk about community level data.

I feel that one of the motivations of the paper was to map "strategies", but in my opinion by being more honest about the limitations of such an approach and reporting in a more nuanced, matter-of-factly way the paper actually will gain in substance.

Pollet, T. V., Stulp, G., Henzi, S. P. & Barrett, L. Taking the aggravation out of data aggregation: A conceptual guide to dealing with statistical issues related to the pooling of individual-level observational data: Data Aggregation and the Ecological Fallacy. *Am. J. Primatol.* 77, 727–740 (2015).

Robinson, W. S. Ecological Correlations and the Behavior of Individuals. *American Sociological Review* 15, 351–357 (1950).

Minor comments

L72ff: "We found that the range of water stress strategies which dominated at community-level were consistent with those available at species-level." – thank you for the modification, this sentence (and other instances of similar changes) read much more clearly now.

L99f: "Ecology strategies describe the set of adaptation or behaviours" – "Ecological strategies...", "...of adaptations and behaviours"

L154ff: "The choice of these traits is driven particularly by their relevance for informing process-based modelling of forest form and function" – thanks for adding this, it gives a lot of weight to the study to know the motives behind the choice of traits!

109: "regions, are currently lacking" – the comma seems misplaced here.

L171: "when water is limiting;" – Introduction ends with a semicolon – something went wrong here.

L197ff: "...showing consistency between acquisitive-conservative strategies at the leaf-level and the wider plant structure, as previously reported at species level. This consistency reflects a whole-plant functional coherence..." – I am having problems here because of the fact that these statements are based on community data, which do not permit inference on the aggregation levels where strategies matter (see major comment).

L222ff: "We relate this axis to the isohydricity strategy (Table 1), with higher PC2 values associated with a more anisohydric strategy (more negative leaf turgor loss and higher wood density)" – I would write "we relate this axis to the isohydr(icit)y spectrum", because you use "anisohydric strategy" below and hence a reader would assume that a more isohydric strategy would be placed on the other side of the axis.

L224ff: "A more negative leaf turgor loss point and higher wood density contributed to the positive direction of this axis, while a steeper embolism vulnerability [...] contributed to the negative direction" – this sounds a bit convoluted. I guess you could write "Leaf turgor loss point and vulnerability curve slope were negatively and wood density positively associated with this axis", "Communities with a more negative average leaf TLP and a higher average wood density scored higher on this axis, while communities with a lower average VC slope had lower scores" or something like that.

L228f: "is likely driven by the predominance of conifers at more negative P50" – probably you'd better say "the often-higher embolism resistance of conifers" or (maybe closer to the truth) "the high proportion of conifers among highly embolism resistant species" but in either case this statement requires a reference, because at least in temperate regions I am not sure whether this is actually true.

L262ff: "Our sensitivity tests indicate that the coordination principles of forest communities reported above were robust with respect to the methods used to gap fill traits after species matching" – sensitivity analyses generally are a good idea, but at least for the imputation methods, have you considered to use the same imputation on a separate dataset not used in your analysis (say, other species from the same ecosystems that are available in the database") to check their performance? You could impute the data for species where you actually know the true values and check the average random and systematic errors, variance explained by imputed values etc. If 80% of the data for some of the most important traits were imputed in one of my analyses, I would for sure be interested in how well the imputation actually works.

L293ff: "There are, however, large areas in eastern USA and Central Europe that do not tend towards either strongly anisohydric or isohydric, consistent with tree species being ordered on a continuum rather than a dichotomy" – avoid ecological fallacies! This is based on community level data, so no conclusions can be drawn about species. As implied in the next sentence, each location likely may

contain both more isohydric and more anisohydric species. Averaging over their traits may or may not result in a reasonable estimate of the average trait combination at that site.

L366ff: "Furthermore, scaling up individual or species-level traits to the community level may only capture the dominant strategies." – to be honest it is not even guaranteed that this method captures dominant strategies (see major comment).

L378: "was not include" – should be "was not included"

L385: "to separate natural biogeography from human managed decisions" – "from human management decisions"?

L394: "Our results demonstrate consistency in trait coordination and trade-offs at both community-level and species-level" – please re-think the community-level trade-offs.

L397: "the identified geographical patterns of functional strategy" – I would be careful with such claims in the conclusion because they seem an overreach (compare major comment).

L402ff: "We found that xylem embolism and [...] collapsed into a single axis of variation [...]. But the community-level xylem cavitation curve [...] loaded onto an orthogonal axis [...]." – P50 and the slope of the "community level xylem cavitation curve" (what does that even mean?) are estimated from the same vulnerability curve – they are the two parameters that describe the relationship between drought-induced loss of xylem-conductance and water potential. If they load on two different axes, xylem embolism does for sure not "collapse into a single axis of variation" with other traits.

L407f: "Our results suggest that dominant water stress strategies are more closely related to water demand than water availability." – I would not overinterpret this, because the selection for water stress strategies is likely driven by the extremes in temperature and water availability rather than average values, so I would not be too surprised if there are only weak patterns with AI and mean summer or annual temperatures.

L412ff: "Whilst to some extent, this filter may be determined by small-scale heterogeneity in edaphic and climatic conditions, it also indicates that dominant water stress strategy may be influenced by the trade-offs among multiple functional traits across a much wider range of environmental conditions." – or maybe there are simply many ways to skin a cat. If different species at the same site rely on different solutions to the same problem (surviving under drought) that require different adaptations visible in the species-level trait values, this information will simply be lost in the CWMs.

L501ff: "All eight functional traits were selected a priori before the PCA..." – again, thanks for clarification, this is a really fundamental point for this type of analysis.

L507f: "The PCA was conducted to explore the main axes of functional strategies at species-level and community-level" – note that here you still speak of strategies at community level. I would really recommend to stick something less divisive, like "main axes of trait variation".

L530f: "We used the Procrustes analysis using package (version 2.6-4) 74 to check the similarity of PCA tests" – this is based on R package vegan, right? Is the package name missing here?

Dear reviewers,

Thank you very much for giving us another opportunity to revise our manuscript “Mapping multi-dimensional variability in water stress strategies across temperate forests”. We have revised the manuscript based on your suggestions and comments. Please see the point-by-point responses below.

Best regards,

Daijun Liu on behalf of all authors

Responses to the two reviewers

#####

REVIEWER COMMENTS

Reviewer #1 (Remarks to the Author):

Second Review of the manuscript entitled Mapping multi-dimensional variability in water stress strategies across temperate forests by Liu et al.

General statement and summary: Thank you for the detailed answers to my points raised. I enjoyed reading the manuscript again. The authors thoroughly incorporated the suggested changes, and I hope and think the manuscript could profit from these.

In detail, the authors improved their manuscript in terms of clarity, analysis, and ecological embedding.

Overall, I think this is a well-written paper, with a sound analysis and interesting results that give rise to further questions.

Thank you very much for your positive comment.

Minor:

Only in a couple of parts, I missed some clarity or a simpler language (but I am not a native speaker).

Clarity. In few places small changes could increase clarity.

We have revised them. Please see the responses below.

- Thank you for integrating changes into (earlier lines 126). With re-reading, I would again take the chance to clarify the sentence more. I am sure the authors can do a better job than me. Possibly I have difficulties understanding the purpose of this sentence: Lines 133 –134: "When functional traits at the species-level are assembled into communities, the dominant strategy depends on the balance between niche complementarity and resource competition" I read two slightly different things in there. You are likely describing (a) the assembly process of species into a community - based on their plant traits, and not (b) the way how the dominant strategy arises methodologically from the aggregation of trait data. For (a) Could you clarify not traits are assembled but species based on their traits? Possibly very briefly define "dominant strategy" in this context (unless I missed it).

We have removed the first sentence of the paragraph (we conclude that it is unnecessary after the rephrase) and replaced the second sentence with the following (Lines 131-133):

“When species assemble into communities, the strategy that comes to dominate depends on niche complementarity and resource competition^{23–25}.”

- Rather introduce the models before referring to them Line 143: "These models [...]"

We have rephrased this sentence (Lines 139-142):

“Understanding trait associations is, however, fundamental to being able to predict the likely response of the forest community to ongoing environmental changes, and to underlie quantitative predictions by the vegetation models which are used to make projections of future forest function ².”

- Suggestion to change and simplify. Lines 135 -136, e.g. (only a suggestion, I am not an English native speaker) "As With reduced water stress water resources become less limiting, competition may tend to favour more acquisitive strategies, resulting in a convergence towards a limited range of water-demanding strategies 15,16,26».

Thank you very much. We prefer to keep this sentence because it is nice to refer to acquisitive strategies, which we have defined earlier in the argument. But we have revised it as (Lines 133-135):

“As water resources become less limiting, competition may tend to favour more acquisitive strategies, resulting in a convergence towards a limited range of water-demanding strategies ^{9,10,17,26}.”

- Logic: Clarify if these strategies are (a) related to species' plasticity, or whether they are (2) present due to an assembly processes. Lines 137-138: "In water-limited environments, a wide range of water stress strategies can develop partitioned environmentally according to particular combinations of conditions that favour certain strategies^{7,27,28}.» E.g. (only a suggestion, I am not an English native speaker) In water-limited environments, a wide range of water stress strategies may survive/be present. These strategies have shown to be ecologically partitioned according to specific combinations of environmental conditions (add reference).

We have revised it accordingly. Please see the main text (Lines 135-137).

“In water-limited environments, a wide range of water stress strategies may be presented. These strategies have been shown to be ecologically partitioned according to specific combinations of environmental conditions ^{11,27,28}.”

- Check this sentence again Lines 151-154: "We considered eight continuous functional traits of woody plants (shrubs and trees) related to potential forest functional strategies to water stress (Table 1), concentrating on acquisitive-conservative, structural, stomatal and water storage strategies, as the information on rooting traits is very limited. «

We have revised this sentence, and it reads as (Lines 150-153):

“We considered eight continuous functional traits of woody plants (shrubs and trees) related to potential functional strategies for dealing with water stress (Table 1). We concentrated on acquisitive-conservative, structural, stomatal and water storage strategies, as the information on rooting traits is very limited.”

- I commented on this before, I am wondering if one could harmonize this to either question

or description of analytical steps. Lines 162 – 166: (I am sure the authors can make a better job than me, in order to illustrate what I mean with extracting the question, see this example:) "We (a) reduced the water stress traits at community-level by means of a principal component analysis (PCA), (b) compared it to the one at the species-level and (c) related these strategies to geographic and climatic space of temperate forest communities (Fig. 1c, d).»

We have accepted your suggestion to revise it (Lines 164-168).

“We first assessed the axes of variation of community-level trait associations with respect to water stress traits based on principal component analysis (PCA), then compared them to the axes of variation of plant strategies at the species level. Finally, we related the community-level associations to the geographic and climatic space of temperate forest communities (Fig. 1c, d).”

#####

Reviewer #2 (Remarks to the Author):

General comments

In the revised version of the manuscript by Liu et al., I appreciate the thorough effort directed at answering all reviewer comments in great detail. I feel that the authors in general did a great job at considering reviewer suggestions, in many cases accepting the suggested changes, while explaining their motives clearly in places where they decided against the reviewer input.

As stated in the last round of reviews, I feel that ecological strategies (i.e., sets of traits/collections of traits in syndromes that are jointly selected for, cf. Reich et al. 2003) are a concept that is not meaningful on community level, because evolutionary selection does not act on communities. For this reason, I am very glad that the authors agreed to downweigh the focus on "community-level strategies" and "community-level trade-offs". Their choice of talking of "locally dominant strategies" instead is certainly an improvement, as in this phrasing it is acknowledged that strategies reflect processes that do not act on community level. Indeed, ecological strategies are now clearly defined in the manuscript sensu Westoby (1998) as "the set of adaptation[s] or behaviours that allow a species to maintain a population", i.e., defining strategies as properties of species, not communities.

However, I feel that the paper still could benefit from being more precise about the levels of aggregation the analysis is based upon. In the following, I will briefly expand on the issue I am having with drawing inference on ecological strategies based on community level data, and then list a couple of minor issues that I feel should be addressed before publication. Pending these suggested changes, I feel the paper now is a good candidate for publication in Nature Communications.

Thank you very much for your suggestions.

Specific comments

Major comments

Apart from the species-level analysis shown in the supplementary material, the main focus of inference in the paper is on community-level data. However, there are many instances where

conclusions are drawn about processes acting at the individual or species level based on community-level data, which constitutes a logical fallacy, the so-called ecological fallacy (Robinson 1950, see Pollet et al. 2015 for a friendly low-level explanation directed at biologists, albeit in a very different context).

In my opinion, the drivers of association between traits on species level (specifically evolutionary selection and allocation trade-offs) are very different from the processes acting on community level (community assembly processes like environmental filtering). While the main patterns in trait association on species level will likely remain visible on community level averages, there is no guarantee that this is always the case. Indeed, when there is more than one viable strategy to deal with environmental constraints and these strategies are rather different, the community average of traits may not be very meaningful at all. As an example, in tropical dry forest ecosystems you find species with a high wood density, low stemwood capacitance, rather embolism resistant wood, and small, hard, evergreen leaves, i.e. species that have a water stress tolerance strategy. These species coexist with stemwater-storing species with low wood density, high capacitance, low embolism resistance and large, thin, drought-deciduous leaves, i.e., species that have a drought avoidance strategy. Averaging over all these traits may result in an intermediate trait combination that is not present on the site and might not even be viable under the local conditions. This hopefully illustrates that CWMs do not necessarily describe a "locally dominant" strategy.

While I think that with an appropriate caveat it can be permissible to interpret CWMs as an approximation to locally dominant strategies, I feel it problematic to use community-level data to jump back to conclusions on the individual or species level, and personally would probably rather speak of trait associations on community level rather than strategies. I have the impression that currently, there are still many places in the paper that either use community level data to infer about species-level strategies, or, even more problematic, imply that strategies act at or are measurable at the community level, for example (list not exhaustive):

Thank you for your thorough checks and pedagogic explanations – it is really much appreciated! We have accepted your comments. We have revised all the points as you suggested and have rechecked the description of functional strategies on the community in the manuscript. We have broadly moved to a terminology of “trait associations”, making inferences based on these regarding the dominant strategy.

L188: "Variation in strategy at the community level"

Revised it to “Variation in trait associations on the community level”.

L188f: "dominant community functional strategies"

Revised it to “Trait associations”

L255: "The strategy space at community-level"

Revised it to “functional trait space at community level”.

L257f: "The consistency of the strategy space across both species and community levels"

We have revised this sentence. Now the issue is not exiting anymore.

L272ff: "Geographic variation of functional strategies. [...] Acquisitive strategies [...] were mainly distributed in [...], whilst conservative strategies [...] were distributed in[...]."

Revised all of them (Lines 272-277).

“Geographic variation of trait associations. We found distinct geographic patterns of the trait association variation across the regions (Fig. 3). Trait associations consistent with an underlying acquisitive strategy (or higher PC1 scores) were mainly distributed in central and eastern USA and Northern France. Whilst associations linked to a conservative strategy (or lower PC1 scores) were distributed in the western USA, boreal regions and the Mediterranean Basin (Fig. 3).”

L305ff: "Climatic factors for variation of functional strategies" – in this section, all the regressions with PCA scores derived from community level trait averages are interpreted in terms of ecological strategies.

We have revised all the descriptions on functional strategies at community level.

L310f: "dominant community-level hydraulic strategy"

Revised.

L334f: "forest communities can follow a broad range of water stress strategies"

We have revised this sentence and it reads as:

“This supports Hypothesis 3 that the trait associations of forest communities diverge when water is limiting, reflecting an underlying broad range of species-specific water stress strategies to survive in these conditions.”

L394f: "consistency in trait coordination and trade-offs at both community-level and species-level"

We have deleted the “and trades-offs”.

L507f: "functional strategies at species-level and community-level"

We have revised it.

“The PCA was conducted to explore the main axes of trait variation at species level and community level using the *FactoMineR* package in R (version 1.34)⁶⁹.”

In many cases, rewording the findings in a way that does not conflate inference on different aggregation levels is easy, and actually closer to what actually happened during analysis. For instance, instead of saying that "Acquisitive strategies [...] were mainly distributed in central and eastern USA and Northern France", you could say that ... that "Trait combinations consistent with an acquisitive strategy [...] were more prevalent in plant communities in central and eastern USA and Northern France". Analogously, I would recommend avoiding conclusions on strategies (which make sense on individual or species level) based on

community level aggregates elsewhere in the paper and rephrase it in a more careful, hedging way that acknowledges that you talk about community level data.

We have accepted your suggestion. Now we have revised them in whole manuscript. “Trait associations consistent with an underlying acquisitive strategy (or higher PC1 scores) were mainly distributed in central and eastern USA and Northern France. Whilst associations linked to a conservative strategy (or lower PC1 scores) were distributed in the western USA, boreal regions and the Mediterranean Basin (Fig. 3).”

I feel that one of the motivations of the paper was to map "strategies", but in my opinion by being more honest about the limitations of such an approach and reporting in a more nuanced, matter-of-factly way the paper actually will gain in substance.

We have deleted the description on “mapping functional strategies” and worked to make the text more nuanced as suggested.

Pollet, T. V., Stulp, G., Henzi, S. P. & Barrett, L. Taking the aggravation out of data aggregation: A conceptual guide to dealing with statistical issues related to the pooling of individual-level observational data: Data Aggregation and the Ecological Fallacy. *Am. J. Primatol.* 77, 727–740 (2015).

Robinson, W. S. Ecological Correlations and the Behavior of Individuals. *American Sociological Review* 15, 351–357 (1950).

Minor comments

L72ff: "We found that the range of water stress strategies which dominated at community-level were consistent with those available at species-level." – thank you for the modification, this sentence (and other instances of similar changes) read much more clearly now.

Thank you very much for your positive comment.

L99f: "Ecology strategies describe the set of adaptation or behaviours"
– "Ecological strategies...", "...of adaptations and behaviours"

We have revised them accordingly.

L154ff: "The choice of these traits is driven particularly by their relevance for informing process-based modelling of forest form and function" – thanks for adding this, it gives a lot of weight to the study to know the motives behind the choice of traits!

Thank you.

109: "regions, are currently lacking" – the comma seems misplaced here.

Revised it.

L171: "when water is limiting;" – Introduction ends with a semicolon – something went wrong here.

Revised it.

L197ff: "...showing consistency between acquisitive-conservative strategies at the leaf-level and the wider plant structure, as previously reported at species level. This consistency reflects a whole-plant functional coherence..." – I am having problems here because of the fact that these statements are based on community data, which do not permit inference on the aggregation levels where strategies matter (see major comment).

We have revised the text to relate the associations at community level to literature on strategies at the species level, avoiding making fallacious inferences.

L222ff: "We relate this axis to the isohydricity strategy (Table 1), with higher PC2 values associated with a more anisohydric strategy (more negative leaf turgor loss and higher wood density)" – I would write "we relate this axis to the isohydr(icit)y spectrum", because you use "anisohydric strategy" below and hence a reader would assume that a more isohydric strategy would be placed on the other side of the axis.

We have revised it based on your suggestion (Lines 219-220).

"We relate this axis to the isohydricity spectrum (Table 1), with higher PC2 values associated with a more anisohydric strategy (more negative leaf turgor loss and higher wood density)."

L224ff: "A more negative leaf turgor loss point and higher wood density contributed to the positive direction of this axis, while a steeper embolism vulnerability [...] contributed to the negative direction" – this sounds a bit convoluted. I guess you could write "Leaf turgor loss point and vulnerability curve slope were negatively and wood density positively associated with this axis", "Communities with a more negative average leaf TLP and a higher average wood density scored higher on this axis, while communities with a lower average VC slope had lower scores" or something like that.

We have accepted the second suggestion and revised this sentence. Now it reads as (Lines 221-223):

"Communities with a more negative average leaf turgor loss point and a higher average wood density scored higher on this axis, while communities with a lower average slope of embolism vulnerability curve had lower scores (Fig. 2a)."

L228f: "is likely driven by the predominance of conifers at more negative P50" – probably you'd better say "the often-higher embolism resistance of conifers" or (maybe closer to the truth) "the high proportion of conifers among highly embolism resistant species" but in either case this statement requires a reference, because at least in temperate regions I am not sure whether this is actually true.

We have revised this sentence and now it reads as (Lines 223-226):

"More robust xylem (i.e., more negative P50) was associated with higher wood density, but the correlation was relatively weak and P50 and wood density largely loaded orthogonally on the PCA, consistent with wood density being a complex trait which is associated with many ecological processes^{10,35,39}."

L262ff: "Our sensitivity tests indicate that the coordination principles of forest communities reported above were robust with respect to the methods used to gap fill traits after species

matching" – sensitivity analyses generally are a good idea, but at least for the imputation methods, have you considered to use the same imputation on a separate dataset not used in your analysis (say, other species from the same ecosystems that are available in the database") to check their performance? You could impute the data for species where you actually know the true values and check the average random and systematic errors, variance explained by imputed values etc. If 80% of the data for some of the most important traits were imputed in one of my analyses, I would for sure be interested in how well the imputation actually works.

We thank you for your suggestion on the sensitivity test. Now we have tested for the eight traits separately for the woody species used in our analysis. We randomly selected 20% of species which have the trait values and then checked the correlations with the trait values of these selected species predicted by the gap-filling method - the median of genus and family and phylogenetic relationship. The correlation between the original values and predicted ones for all of the eight traits had significances ($R^2 = 0.43 \sim 0.83$; all p values < 0.001). We have added the sensitivity result in the Result and Discussion (Line 260-262) and Methods and Materials (Lines 527-531).

“We tested the gap-filling method used for traits across species by predicting 20% of known values for each trait, finding strong and significant correlations in all cases (all p values < 0.001, $R^2 = 0.43-0.83$; Fig. S5).”

“In order to test the effectiveness of the gap-filling method, for species for which we had observations for a particular trait, we randomly selected 20% of these species and predicted their values from the remaining 80%, following the gap-filling method described above - the median of genus and family and phylogenetic relationship (default, as described above).”

L293ff: "There are, however, large areas in eastern USA and Central Europe that do not tend towards either strongly anisohydric or isohydric, consistent with tree species being ordered on a continuum rather than a dichotomy" – avoid ecological fallacies! This is based on community level data, so no conclusions can be drawn about species. As implied in the next sentence, each location likely may contain both more isohydric and more anisohydric species. Averaging over their traits may or may not result in a reasonable estimate of the average trait combination at that site.

We thank you on this statement. We have deleted the strategy at species level. Now it reads as (Line 293-295):

“There are, however, large areas in eastern USA and Central Europe that do not tend towards either strongly anisohydric or isohydric trait associations at the community level.”

L366ff: "Furthermore, scaling up individual or species-level traits to the community level may only capture the dominant strategies." – to be honest it is not even guaranteed that this method captures dominant strategies (see major comment).

We have revised this sentence (Lines 378-379).

“Scaling up individual or species-level traits to the community level may only capture the average of trait values, which approximate locally strategies.”

L378: "was not include" – should be "was not included"

Revised it.

L385: "to separate natural biogeography from human managed decisions" – "from human management decisions"?

Revised it.

L394: "Our results demonstrate consistency in trait coordination and trade-offs at both community-level and species-level" – please re-think the community-level trade-offs.

We have deleted the trade-offs.

L397: "the identified geographical patterns of functional strategy" – I would be careful with such claims in the conclusion because they seem an overreach (compare major comment).

Revised it to "The identified geographical patterns of trait associations".

L402ff: "We found that xylem embolism and [...] collapsed into a single axis of variation [...]. But the community-level xylem cavitation curve [...] loaded onto an orthogonal axis [...]." – P50 and the slope of the "community level xylem cavitation curve" (what does that even mean?) are estimated from the same vulnerability curve – they are the two parameters that describe the relationship between drought-induced loss of xylem-conductance and water potential. If they load on two different axes, xylem embolism does for sure not "collapse into a single axis of variation" with other traits.

We apologise for this error we made. We have revised this sentence and it reads as (Lines 404-408):

"We found that traits relating to xylem embolism resistance, sapwood conductivity and tree structural strategies collapsed into a single axis of variation, loading into the widely reported acquisitive-conservative spectrum (or fast-slow continuum). But the xylem cavitation slope, leaf turgor loss point and wood density loaded onto an orthogonal axis, associated with the spectrum of isohydricity and, tentatively, water storage."

L407f: "Our results suggest that dominant water stress strategies are more closely related to water demand than water availability." – I would not overinterpret this, because the selection for water stress strategies is likely driven by the extremes in temperature and water availability rather than average values, so I would not be too surprised if there are only weak patterns with AI and mean summer or annual temperatures.

We appreciate your comment on this point. We found that water stress strategies had stronger power with the temperature variability than water availability. There is a sentence which states the weak correlation (Lines 411-414). We thus believe it is fine to conclude it here.

"But the relationships with climate are relatively weak, and the proliferation of different strategies in more arid environments, along the limited explanatory power of climate

variables, raises the question of what factors exert the primary filter governing which water stress strategy comes to dominate a forest community.”

L412ff: "Whilst to some extent, this filter may be determined by small-scale heterogeneity in edaphic and climatic conditions, it also indicates that dominant water stress strategy may be influenced by the trade-offs among multiple functional traits across a much wider range of environmental conditions." – or maybe there are simply many ways to skin a cat. If different species at the same site rely on different solutions to the same problem (surviving under drought) that require different adaptations visible in the species-level trait values, this information will simply be lost in the CWMs.

We thank you this point. We fully agree that CWMs may not include the different adaptations for the species. We have stated this limitation in the “Future challenges” section (Lines 378-382). We thus prefer not to show this statement again in the “Conclusion” section.

“Scaling up individual or species-level traits to the community level may only capture the average of trait values, which approximate locally strategies. The intraspecific trait variation in response to environmental conditions was not included according to the method of community-weighted mean traits.”

L501ff: "All eight functional traits were selected a priori before the PCA..." – again, thanks for clarification, this is a really fundamental point for this type of analysis.

Thank you.

L507f: "The PCA was conducted to explore the main axes of functional strategies at species-level and community-level" – note that here you still speak of strategies at community level. I would really recommend to stick something less divisive, like "main axes of trait variation".

We have accepted your suggestion to change it as “main axes of trait variation” (Line 505)

L530f: "We used the Procrustes analysis using package (version 2.6-4) 74 to check the similarity of PCA tests" – this is based on R package vegan, right? Is the package name missing here?

We have added the missing package name as you suggested (Line 534).

REVIEWER COMMENTS

Reviewer #1 (Remarks to the Author):

Specific addition on the gap-filling approach.

I am thankful for raising the point about gap-filling. Overall, the precision of gap-filled data appears to be poor, (but see the comment on the method for evaluating the quality of imputations). The method for gap-filling is valid, which is also supported by similar results of trait-trait correlations, independently of the approach used (Figure S6). For using these data for trait-trait correlations, the results are still likely solid. Based on the Procrustes analysis the trait-trait correlations appear to hold, but see my recommendation on re-doing the test, based on traits, not individual samples.

In order to understand the quality of gap-filling I followed two approaches. (1) Evaluating the overall precision by means of RMSE and comparison to other errors in literature. (2) Evaluating the gap-filled data's usefulness by means of detecting imputed biases from gap-filling for this specific application to evaluate the validity of the claims posed in the study.

(1) Evaluating the overall precision by means of RMSE and comparison to other errors in literature.

a. RMSE (root mean squared error, Schrod et al. 2015) and NRMSE (Poyatos et al 2018, Moreno-Martinez et al. 2018) is the common descriptor of model fit in imputation or gap-filling.

In order to compare the imputation fit of Liu et al., I ran a simple analysis with 1000 random normally distributed samples for a , representing observed data. While b represents imputed/predicted data ($b = a * rnorm(1000, sd = 2.08)$). In my example (10 repetitions, 8 traits), the Pearson correlation coefficient was 69%, being thus similar to some results in Figure S5. This resulted in an RMSE of 2.3.

Figure 1: Randomly generated data with Pearson correlation coefficient of 0.69, RMSE of 2.3. $a_1 =$ "observed", $b_1 =$ "predicted"

b. An RMSE of 2.3 appears to be rather high when comparing to < 1 - the ones evaluated in Joswig et al. 2023 in GEB, Figure 2, which, however are zlog transformed. The analyses refer RMSEs drawn from gap filling with BHPMF of a completely observed data set (1140 individuals x 6 traits, and in supplement 611 individuals x 5 traits with similar results) for which randomly values were deleted and thereafter imputed.

c. Method for evaluating the model fit. Liu et al. evaluated the gap-filling for 20% randomly selected values. Usually, one would predict all samples from 80% of the observations, i.e. split the complete data set into 5 random chunks and predict each one from the rest. Then repeat this approach 10 times to retrieve 10 times an RMSE and r^2 . Doing this only once and only for 20% may not be representative. It did not become clear to me, if all data was predicted, and how many times. Finally the evaluation of gap-filling as in Figure S2 is not be representative.

(2) Liu et al used taxa mean, i.e. genus, family and taxon mean to fill species trait data where missing in addition also the gap-filling method funspace. (" For the species-level functional traits that were missing, we used the median trait values of the same genus instead. And if the trait was still missing, we used the median values of the family to fill the gaps 63. However, there were still some species for which the trait data were missing (3%-25%). We imputed the remaining trait values for the species using the funspace package in R (version 0.1.1) 64, which is coupled with phylogenetic information to impute missing trait data 65.").

Generally, this is a common approach, as trait values of individuals within species tend to be overall more similar than across species, the same goes for species within genera, and genera within families. Although I am not familiar with the funspace package, I am convinced of the validity of this approach judging from the paper describing it.

Taxonomy or phylogeny are useful for gap-filling, also BHPMF makes use of it, one gap-filling algorithm used for traits (Schrod et al. 2015 GEB). Consequently, similar biases as from BHPMF and

mean taxa imputation can be expected. The gap-filled data from Liu et al. are likely reduced in intra-taxa variability, and thus to be strongly taxonomically biased, but may still be useful for trait-trait relationships.

a. Liu et al. claim in the abstract: «We found that the trait associations at community level was consistent with that available at species level.» . Because of the gap-filling this claim may be slightly weakened by the fact the data sets (if I understand correctly) are the same. It would be useful to provide the percentage of shared data. I saw reviewer 2 commented on this aspect earlier and hope she/he can contribute to this aspect.

b. Liu et al. claim: "Traits associated with acquisitive-conservative strategies formed one dimension of variation, while leaf turgor loss point, associated with stomatal water regulation strategy, loaded along a second dimension." (Abstract) And indeed, despite the bias and the beforementioned high gap-filling RMSE, trait-trait relationships may still hold. In my last review I overlooked the requested Procrustes analysis is based on the individual samples (here species), and not on the traits themselves. Although it seems from the analysis of the individual samples, the trait-trait correlations might be in fact similar as well, I would strongly recommend to do the Procrustes analysis as well as the significance test for traits (in R this would be `procrustes(PCA_outvarcoord[,1:2])`, plus `protest()`\$signif).

c. I used the small analysis with 8 random traits without induced trait-trait correlations to evaluate the influence of rather high RMSE (see 1.c and figure 1) on trait-trait correlations. In 5 out of the 10 repetitions Procrustes would find significance similarities (when based on individual samples, not traits, the ratio was approximately the same). The R-script is added below. Weakness of this approach is of course the missing taxonomic hierarchy, trait-trait correlations, and missing any observations. As a consequence, the trait-trait relationships may still hold, but also may not. I expect them to still hold, but again, one should verify with the Procrustes test mentioned above.

References:

- Joswig, J. S., Kattge, J., Kraemer, G., Mahecha, M. D., Rüger, N., Schaepman, M. E., Schrodte, F., & Schuman, M. C. (2023). Imputing missing data in plant traits: A guide to improve gap-filling. *Global Ecology and Biogeography*, 32, 1395–1408. <https://doi.org/10.1111/geb.13695>
- Moreno-Martínez, Á., Camps-Valls, G., Kattge, J., Robinson, N., Reichstein, M., van Bodegom, P., Kramer, K., Cornelissen, J. H. C., Reich, P., Bahn, M., Niinemets, Ü., Peñuelas, J., Craine, J. M., Cerabolini, B. E. L., Minden, V., Laughlin, D. C., Sack, L., Allred, B., Baraloto, C., ... Running, S. W. (2018). A methodology to derive global maps of leaf traits using remote sensing and climate data. *Remote Sensing of Environment*, 218, 69– 88. <https://doi.org/10.1016/j.rse.2018.09.006> ISSN 00344257
- Poyatos, R., Sus, O., Badiella, L., Mencuccini, M., & Martínez-Vilalta, J. (2018). Gap-filling a spatially explicit plant trait database: Comparing imputation methods and different levels of environmental information. *Biogeosciences*, 15(9), 2601–2617. <https://doi.org/10.5194/bg-15-2601-2018> ISSN 1868-9746

R-script:

```
require(FactoMineR)
install.packages("vegan")
library(vegan)

# produce input matrices
res_cor <- matrix(NA,nrow=10,ncol=8)
res_rmse <- matrix(NA,nrow=10,ncol=8)
res_protest <- rep(NA,10)

for(i in 1:10){ # repetitions
#-----
# produce random data sets = traits
```

```

#-----
a1 <- rnorm(1000) # observed
b1 <- a1*2+rnorm(1000,sd = 2.08) # predicted
res_cor[i,1] <- cor(a1,b1) # input correlation
res_rmse[i,1] <- sqrt(mean((b1 - a1)^2)) # input rmse

# same for all 8 sample traits
a2 <- rnorm(1000)
b2 <- a2*2+rnorm(1000,sd = 2.08)
res_cor[i,2] <- cor(a2,b2)
res_rmse[i,2] <- sqrt(mean((b2 - a2)^2))

a3 <- rnorm(1000)
b3 <- a3*2+rnorm(1000,sd = 2.08)
res_cor[i,3] <- cor(a3,b3)
res_rmse[i,3] <- sqrt(mean((b3 - a3)^2))

a4 <- rnorm(1000)
b4 <- a4*2+rnorm(1000,sd = 2.08)
res_cor[i,4] <- cor(a4,b4)
res_rmse[i,4] <- sqrt(mean((b4 - a4)^2))

a5 <- rnorm(1000)
b5 <- a5*2+rnorm(1000,sd = 2.08)
res_cor[i,5] <- cor(a5,b5)
res_rmse[i,5] <- sqrt(mean((b5 - a5)^2))

a6 <- rnorm(1000)
b6 <- a6*2+rnorm(1000,sd = 2.08)
res_cor[i,6] <- cor(a6,b6)
res_rmse[i,6] <- sqrt(mean((b6 - a6)^2))

a7 <- rnorm(1000)
b7 <- a7*2+rnorm(1000,sd = 2.08)
res_cor[i,7] <- cor(a7,b7)
res_rmse[i,7] <- sqrt(mean((b7 - a7)^2))

a8 <- rnorm(1000)
b8 <- a8*2+rnorm(1000,sd = 2.08)
res_cor[i,8] <- cor(a8,b8)
res_rmse[i,8] <- sqrt(mean((b8 - a8)^2))

#-----
# check trait-trait correlations

plot(a1,b1,pch=16)
abline(0,1)

#-----
# prepare for PCA and procrustes
#-----
# "Observed" data set
dat <- as.data.frame(cbind(a1,a2,a3,a4,a5,a6,a7,a8))
pca1 <- PCA(dat)# PCA obs

```

```

# "Predicted" data set
dat2 <- as.data.frame(cbind(b1,b2,b3,b4,b5,b6,b7,b8))
pca2 <- PCA(dat2)# PCA pred

# run procrustes
proc <- procrustes(pca1$ind$coord[,1:2], pca2$ind$coord[,1:2], scale=TRUE)## S3 method for class
'procrustes':
smry <- summary(proc)
smry$ss
plot(proc, kind=1)

residuals(proc) # check these

#-----
# significance test
#-----
prot <- protest(X = pca1$var$coord[,1:2], Y = pca2$var$coord[,1:2], scores = "sites", permutations
= 999)
res_protest[i] <- prot$signif

}

mean(res_cor)
res_cor
res_rmse
mean(res_rmse)
res_protest
sum(res_protest<.05)

mean(res_protest[,1])

```

Reviewer #2 (Remarks to the Author):

After the last round of reviews of the manuscript by Liu et al., I feel that the authors now addressed all reviewer comments in a thorough and convincing fashion. Specifically, I highly appreciate how much effort was spent in this version to accommodate my comments regarding the concept of strategies on species vs. community level. I feel that conceptual clarity is very important in this point, and hope that my input has contributed to improving the overall quality of the manuscript without being too much of a nuisance to the authors. I just finished thoroughly reading through both the rebuttal letter and the manuscript itself and came to the conclusion that the manuscript has reached a satisfactory level of quality and from my point of view is ready for publication now.

Dear reviewer #1,

Thank you very much for giving us an opportunity to revise our manuscript “Mapping multi-dimensional variability in water stress strategies across temperate forests”. We have responded to the gap-filling method based on your suggestion. Please see the responses below.

Best regards,

Daijun Liu on behalf of all authors

Responses to the reviewer #1

#####

REVIEWER COMMENTS

Reviewer #1 (Remarks to the Author):

(you can find the same text attached as a Word file)
Specific addition on the gap-filling approach.

I am thankful for raising the point about gap-filling. Overall, the precision of gap-filled data appears to be poor, (but see the comment on the method for evaluating the quality of imputations). The method for gap-filling is valid, which is also supported by similar results of trait-trait correlations, independently of the approach used (Figure S6). For using these data for trait-trait correlations, the results are still likely solid. Based on the Procrustes analysis the trait-trait correlations appear to hold, but see my recommendation on re-doing the test, based on traits, not individual samples.

Thank you very much for your comment. Please see the responses below.

In order to understand the quality of gap-filling I followed two approaches. (1) Evaluating the overall precision by means of RMSE and comparison to other errors in literature. (2) Evaluating the gap-filled data's usefulness by means of detecting imputed biases from gap-filling for this specific application to evaluate the validity of the claims posed in the study. (1) Evaluating the overall precision by means of RMSE and comparison to other errors in literature. a. RMSE (root mean squared error, Schrodt et al. 2015) and NRMSE (Poyatos et al 2018, Moreno-Martinez et al. 2018) is the common descriptor of model fit in imputation or gap-filling. In order to compare the imputation fit of Liu et al., I ran a simple analysis with 1000 random normally distributed samples for a, representing observed data. While b represents imputed/predicted data ($b=a*\text{rnorm}(1000, \text{sd}=2.08)$). In my example (10 repetitions, 8 traits), the Pearson correlation coefficient was 69%, being thus similar to some results in Figure S5. This resulted in an RMSE of 2.3.

Figure 1: Randomly generated data with Pearson correlation coefficient of 0.69, RMSE of 2.3. a1= “observed”, b1=”predicted”

b. An RMSE of 2.3 appears to be rather high when comparing to < 1 - the ones evaluated in Joswig et al. 2023 in GEB, Figure 2, which, however, are zlog transformed. The analyses refer RMSEs drawn from gap filling with BHPMF of a completely observed data set (1140 individuals x 6 traits, and in supplement 611 individuals x 5 traits with similar results) for which randomly values were deleted and thereafter imputed.

c. Method for evaluating the model fit. Liu et al. evaluated the gap-filling for 20% randomly selected values. Usually, one would predict all samples from 80% of the observations, i.e. split the complete data set into 5 random chunks and predict each one from the rest. Then repeat this approach 10 times to retrieve 10 times an RMSE and r^2 . Doing this only once and only for 20% may not be representative. It did not become clear to me, if all data was predicted, and how many times. Finally the evaluation of gap-filling as in Figure S2 is not be representative.

Thank you very much for giving an example to explain the gap-filling method. We agree that the two suggested approaches: 1) Evaluating the overall precision by means of RMSE and comparison to other errors in literature; 2) Evaluating the gap-filled data’s usefulness by means of detecting imputed biases from gap-filling for this specific application to evaluate the validity of the claims posed in the study. We have also noticed that the sensitivity test by evaluating the gap-filling for only 20% randomly selected values may be not representative. Thus, we randomly selected for seven different proportions (0.2, 0.3, 0.4, 0.5, 0.6, 0.7, 0.8) of species for which we had observations of trait values and then predicted the trait values for these species from the remaining species for which we had trait values, based on the gap-filling trait method a - the median of genus and family and phylogenetic relationship. Then we randomly run 100 times to obtain the R^2 and RMSE values. Lastly, we evaluate the mean values of R^2 and RMSE. And the results have been shown in Figure S5 in the Supplementary information.

Accordingly, we have revised the sensitivity test description in the Methods and Materials and Results sections (Lines 529-539 and Lines 260-264; respectively).

“In order to test the effectiveness of the gap-filling method, for species for which we had observations for a particular trait, we randomly selected seven different proportions (from 0.2 to 0.8, increasing each 0.1) of the species which we have observations of trait values (missingness proportions) and predicted their values from the remaining of the species for which have the trait values, following the gap-filling method described above – i.e. the median of genus and family and phylogenetic relationship (default, as described above). Then we randomly run 100 times to obtain the correlation coefficient (R^2) and root mean squared error (RMSE) values of z-transformed predicted versus observation trait values as indicator of overall prediction accuracy for trait gap-filling^{73,74}. Lastly, we evaluated the mean R^2 and RMSE values. Here, we did not use very low proportions of missingness (< 0.2) in order to avoid the observation bias from low number observations.”

“We tested the gap-filling method used for each trait across species by validating different proportions of missingness (0.2-0.8 for the species which have observation trait values), finding strong correlation coefficients (mean $R^2 = 0.62-0.94$; Fig. S5a) and low root mean squared error (RMSE) (mean RMSE < 1 ; 0.3-0.86; Fig. S5b) for all comparisons. Generally, we found the differences to be relatively small compared to the trait values from the observations.”

(2) Liu et al used taxa mean, i.e. genus, family and taxon mean to fill species trait data where missing in addition also the gap-filling method funspace. (“ For the species-level functional traits that were missing, we used the median trait values of the same genus instead. And if the trait was still missing, we used the median values of the family to fill the gaps⁶³. However, there were still some species for which the trait data were missing (3%-25%). We imputed the remaining trait values for the species using the funspace package in R (version 0.1.1)⁶⁴, which is coupled with phylogenetic information to impute missing trait data⁶⁵.”).

Generally, this is a common approach, as trait values of individuals within species tend to be overall more similar than across species, the same goes for species within genera, and genera within families. Although I am not familiar with the funspace package, I am convinced of the validity of this approach judging from the paper describing it. Taxonomy or phylogeny are useful for gap-filling, also BHPMF makes use of it, one gap-filling algorithm used for traits (Schrodtt et al. 2015 GEB). Consequently, similar biases as from BHPMF and mean taxa imputation can be expected. The gap-filled data from Liu et al. are likely reduced in intra-taxon variability, and thus to be strongly taxonomically biased, but may still be useful for trait-trait relationships.

Thank you very much for your comment.

a. Liu et al. claim in the abstract: «We found that the trait associations at community level was consistent with that available at species level.» . Because of the gap-filling this claim may be slightly weakened by the fact the data sets (if I understand correctly) are the same. It would be useful to provide the percentage of shared data. I saw reviewer 2 commented on this aspect earlier and hope she/he can contribute to this aspect.

Indeed, the trait associations at community and species level do use the same trait data. The point of this aspect of the analysis was to see if the trait vs trait relationships that manifest at species level also manifest at community level after the filtering which occurs in community assembly. It is entirely plausible that some trait associations, whilst are well represented at the

species level, are not well represented at the community level because the species driving them are relatively rare. Our hypothesis, “the strategic variation at species level will be reflected in the variation of trait associations at community level”, is therefore most consistently tested using the same data and not independent datasets in which the test could potentially be confounded by differences emerging in the species-level relationships.

b. Liu et al. claim: “Traits associated with acquisitive-conservative strategies formed one dimension of variation, while leaf turgor loss point, associated with stomatal water regulation strategy, loaded along a second dimension.” (Abstract) And indeed, despite the bias and the beforementioned high gap-filling RMSE, trait-trait relationships may still hold. In my last review I overlooked the requested Procrustes analysis is based on the individual samples (here species), and not on the traits themselves. Although it seems from the analysis of the individual samples, the trait-trait correlations might be in fact similar as well, I would strongly recommend to do the Procrustes analysis as well as the significance test for traits (in R this would be `procrustes(PCA_outvarcoord[,1:2])`, plus `protest()`\$signif).

We thank you for your recommendation. Indeed, we have tested the Procrustes analysis and the significance test for the eight traits with different imputation methods. The results indicate that the different gap-filling methods for the traits were similar for the comparisons (all p values <0.001). Please see the Figure S8 in supporting information.

c. I used the small analysis with 8 random traits without induced trait-trait correlations to evaluate the influence of rather high RMSE (see 1.c and figure 1) on trait-trait correlations. In 5 out of the 10 repetitions Procrustes would find significance similarities (when based on individual samples, not traits, the ratio was approximately the same). The R-script is added below. Weakness of this approach is of course the missing taxonomic hierarchy, trait-trait correlations, and missing any observations. As a consequence, the trait-trait relationships may still hold, but also may not. I expect them to still hold, but again, one should verify with the Procrustes test mentioned above.

As suggested, we have carried out the Procrustes test on the trait-trait correlations, finding that they were similar for all gap-filling approaches (please see response to previous comment). We then evaluated the root mean squared error (RMSE) of z-transformed predicted versus observation trait values, finding that the correlation was strong ($R^2=0.62 - 0.94$), and all the RMSE values were much less than 1 (‘1’ refers to 1 standard deviation) and most of them were around 0.5 (please also see our first response above).

References:

- Joswig, J. S., Kattge, J., Kraemer, G., Mahecha, M. D., Rüger, N., Schaepman, M. E., Schrod, F., & Schuman, M. C. (2023). Imputing missing data in plant traits: A guide to improve gap-filling. *Global Ecology and Biogeography*, 32, 1395–1408. <https://doi.org/10.1111/geb.13695>
- Moreno-Martínez, Á., Camps-Valls, G., Kattge, J., Robinson, N., Reichstein, M., van Bodegom, P., Kramer, K., Cornelissen, J. H. C., Reich, P., Bahn, M., Niinemets, Ü., Peñuelas, J., Craine, J. M., Cerabolini, B. E. L., Minden, V., Laughlin, D. C., Sack, L., Allred, B., Baraloto, C., ... Running, S. W. (2018). A methodology to derive global maps of leaf traits using remote sensing and climate data. *Remote Sensing of Environment*, 218, 69–88. <https://doi.org/10.1016/j.rse.2018.09.006> ISSN 00344257
- Poyatos, R., Sus, O., Badiella, L., Mencuccini, M., & Martínez-Vilalta, J. (2018). Gap-filling a spatially explicit plant trait database: Comparing imputation methods and different

R-script:

```
require(FactoMineR)
install.packages("vegan")
library(vegan)

# produce input matrices
res_cor <- matrix(NA,nrow=10,ncol=8)
res_rmse <- matrix(NA,nrow=10,ncol=8)
res_protest <- rep(NA,10)

for(i in 1:10){ # repetitions
#-----
# produce random data sets = traits
#-----
a1 <- rnorm(1000) # observed
b1 <- a1*2+rnorm(1000,sd = 2.08) # predicted
res_cor[i,1] <- cor(a1,b1) # input correlation
res_rmse[i,1] <- sqrt(mean((b1 - a1)^2)) # input rmse
# same for all 8 sample traits
a2 <- rnorm(1000)
b2 <- a2*2+rnorm(1000,sd = 2.08)
res_cor[i,2] <- cor(a2,b2)
res_rmse[i,2] <- sqrt(mean((b2 - a2)^2))
a3 <- rnorm(1000)
b3 <- a3*2+rnorm(1000,sd = 2.08)
res_cor[i,3] <- cor(a3,b3)
res_rmse[i,3] <- sqrt(mean((b3 - a3)^2))
a4 <- rnorm(1000)
b4 <- a4*2+rnorm(1000,sd = 2.08)
res_cor[i,4] <- cor(a4,b4)
res_rmse[i,4] <- sqrt(mean((b4 - a4)^2))
a5 <- rnorm(1000)
b5 <- a5*2+rnorm(1000,sd = 2.08)
res_cor[i,5] <- cor(a5,b5)
res_rmse[i,5] <- sqrt(mean((b5 - a5)^2))
a6 <- rnorm(1000)
b6 <- a6*2+rnorm(1000,sd = 2.08)
res_cor[i,6] <- cor(a6,b6)
res_rmse[i,6] <- sqrt(mean((b6 - a6)^2))
a7 <- rnorm(1000)
b7 <- a7*2+rnorm(1000,sd = 2.08)
res_cor[i,7] <- cor(a7,b7)
res_rmse[i,7] <- sqrt(mean((b7 - a7)^2))
a8 <- rnorm(1000)
b8 <- a8*2+rnorm(1000,sd = 2.08)
```

```

res_cor[i,8] <- cor(a8,b8)
res_rmse[i,8] <- sqrt(mean((b8 - a8)^2))
#-----
# check trait-trait correlations
plot(a1,b1,pch=16)
abline(0,1)
#-----
# prepare for PCA and procrustes
#-----
# "Observed" data set
dat <- as.data.frame(cbind(a1,a2,a3,a4,a5,a6,a7,a8))
pca1 <- PCA(dat)# PCA obs
# "Predicted" data set
dat2 <- as.data.frame(cbind(b1,b2,b3,b4,b5,b6,b7,b8))
pca2 <- PCA(dat2)# PCA pred
# run procrustes
proc <- procrustes(pca1$ind$coord[,1:2], pca2$ind$coord[,1:2], scale=TRUE)## S3 method
for class 'procrustes':
smry <- summary(proc)
smry$ss
plot(proc, kind=1)
residuals(proc) # check these
#-----
# significance test
#-----
prot <- protest(X = pca1$var$coord[,1:2], Y = pca2$var$coord[,1:2], scores = "sites",
permutations = 999)
res_protest[i] <- prot$signif
}
mean(res_cor)
res_cor
res_rmse
mean(res_rmse)
res_protest
sum(res_protest<.05)
mean(res_protest[,1])

```

We thank you for your comment and providing an example for the validation of gap-filling methods.